# Systematic identification of structure-specific protein–protein interactions

Aleš Holfeld[1], Dina Schuster [1,2,3], Fabian Sesterhenn[1], Alison K Gillingham [4], Patrick Stalder[1], Walther Haenseler [1,5], Inigo Barrio-Hernandez [6,7], Dhiman Ghosh [8], Jane Vowles[9], Sally A Cowley [9], Luise Nagel [10], Basavraj Khanppnavar [2,3], Tetiana Serdiuk[1], Pedro Beltrao [1], Volodymyr M Korkhov [2,3], Sean Munro [4], Roland Riek[8], Natalie de Souza [1,11] & Paola Picotti [1]✉

## Abstract

**The physical interactome of a protein can be altered upon perturbation, modulating cell physiology and contributing to disease. Identifying interactome differences of normal and disease states of proteins could help understand disease mechanisms, but current methods do not pinpoint structure-specific PPIs and interaction interfaces proteome-wide. We used limited proteolysis–mass spectrometry (LiP–MS) to screen for structure-specific PPIs by probing for protease susceptibility changes of proteins in cellular extracts upon treatment with specific structural states of a protein. We first demonstrated that LiP–MS detects well-characterized PPIs, including antibody–target protein interactions and interactions with membrane proteins, and that it pinpoints interfaces, including epitopes. We then applied the approach to study conformation-specific interactors of the Parkinson's disease hallmark protein alpha-synuclein (aSyn). We identified known interactors of aSyn monomer and amyloid fibrils and provide a resource of novel putative conformation-specific aSyn interactors for validation in further studies. We also used our approach on GDP- and GTP-bound forms of two Rab GTPases, showing detection of differential candidate interactors of conformationally similar proteins. This approach is applicable to screen for structure-specific interactomes of any protein, including posttranslationally modified and unmodified, or metabolite-bound and unbound protein states.**

**Keywords** Limited Proteolysis; Mass Spectrometry; Structural Proteomics; Protein–protein Interactions; Structure-Specific Interactions
**Subject Category** Proteomics

## Introduction

Many cellular processes are governed by proteins assembled into complexes; thus, protein–protein interactions (PPIs) have multiple essential roles in cells. The physical interactome of a given protein (i.e., the set of proteins with which it interacts) is not static. The organization of the interactome can be altered due to numerous molecular events that occur in response to environmental stimuli, stress, time, and disease state (Goh et al, 2007). These molecular events include not only genetic variations (Carter et al, 2013; Ferlini and Fini, 2015; Auton et al, 2015) but also covalent (Mann and Jensen, 2003; Pan and Chen, 2022; Xu et al, 2018; Jensen, 2004; Khoury et al, 2011; Duan and Walther, 2015) and noncovalent modifications (Schmidt and Robinson, 2014; Gillingham et al, 2019) that can lead to structural alterations of a given protein. Thus, a protein of interest may associate with different sets of protein partners under normal compared to disease conditions.

Abnormal alterations in PPIs have the potential to modulate physiological processes and contribute to disease phenotypes (Sahni et al, 2015; Thompson et al, 2020; Calabrese et al, 2022). For example, in neurodegenerative diseases such as Parkinson's disease (PD), dementia with Lewy bodies (DLB), multiple system atrophy (MSA), Alzheimer's disease (AD), and Huntington's disease, disease-associated proteins (e.g., alpha-synuclein (aSyn), amyloid-β, tau, or huntingtin) aggregate into β-sheet-rich structures that are thought to be toxic to cells (Soto, 2003; Goedert, 2015; Bates, 2003; Taylor et al, 2002; Ross and Poirier, 2004). However, it remains enigmatic how protein aggregation affects cell physiology. One hypothesis is that aggregation-prone proteins, such as aSyn, may undergo changes in their interactomes while transitioning from the monomeric to the aggregated state (Leitão et al, 2021; van Diggelen et al, 2020; Lassen et al, 2016; Betzer et al, 2015). Such interactome changes could profoundly affect cellular physiology and could play a role in the onset of various diseases. Thus, a systematic assessment of structure-specific interactomes could

[1]Institute of Molecular Systems Biology, Department of Biology, ETH Zurich, Zurich, Switzerland.[2]Institute of Molecular Biology and Biophysics, Department of Biology, ETH Zurich, Zurich, Switzerland.[3]Laboratory of Biomolecular Research, Division of Biology and Chemistry, Paul Scherrer Institute, Villigen, Switzerland.[4]MRC Laboratory of Molecular Biology, Cambridge, UK.[5]University Research Priority Program AdaBD (Adaptive Brain Circuits in Development and Learning), University of Zurich, Zurich, Switzerland.[6]European Molecular Biology Laboratory, European Bioinformatics Institute, Wellcome Genome Campus, Hinxton, Cambridge, UK.[7]Open Targets, Wellcome Genome Campus, Hinxton, Cambridge, UK.[8]Laboratory of Physical Chemistry, ETH Zurich, Zurich, Switzerland.[9]James and Lillian Martin Centre for Stem Cell Research, Sir William Dunn School of Pathology, University of Oxford, Oxford, UK.[10]Cluster of Excellence Cellular Stress Responses in Aging-associated Diseases (CECAD), University of Cologne, Cologne, Germany.[11]Department of Quantitative Biomedicine, University of Zurich, Zurich, Switzerland. ✉E-mail: picotti@imsb.biol.ethz.ch

help elucidate pathological cellular processes and unravel disease mechanisms.

Multiple methods have been developed to study PPIs (Meyerkord and Fu, 2015), but all have limitations for the study of structure-specific interactomes. Affinity purification coupled to mass spectrometry (AP–MS) relies on the purification of a bait protein of interest from a cellular extract, together with its interaction partners (Dunham et al, 2012; Collins and Choudhary, 2008; Morris et al, 2014; Meyer and Selbach, 2015; Chang, 2006). These experiments typically only detect stable interactions, and engineered affinity tags may alter protein structures and interaction sites. Furthermore, structural changes in proteins may alter interactions with specific antibodies, thus affecting the capability to detect structure-specific interactomes, and structure-specific antibodies are not available for most proteins (Kumar et al, 2020). Proximity labeling approaches, such as BioID (Roux et al, 2012) or APEX (Martell et al, 2012), identify interacting proteins by fusing one or more baits with an enzyme that covalently labels proximal proteins (Go et al, 2021; Trinkle-Mulcahy and Poterszman, 2019; Han et al, 2018; Xu et al, 2021; Samavarchi-Tehrani et al, 2020). Although these strategies allow the capture of more transient interactions and can be employed in living cells, they also identify bystander proteins that are near the bait but do not interact with it. The interactome can also be profiled in an untargeted manner using co-fractionation techniques coupled with MS (Kirkwood et al, 2013; Bludau et al, 2021; Heusel et al, 2019; Bludau et al, 2020; Heusel et al, 2020), in which proteins are separated according to size/shape (size exclusion chromatography; SEC) or charge (ion exchange chromatography), and interactions are inferred based on co-fractionation patterns (Scott et al, 2015; Hu et al, 2019; Fossati et al, 2021; Stacey et al, 2017). SEC–MS has identified thousands of putative PPIs (Heusel et al, 2019; Bludau et al, 2021, 2020; Heusel et al, 2020; Kristensen et al, 2012; Liu et al, 2008), examined protein complex dynamics (Kristensen et al, 2012), and improved the detection of variations in protein complexes associated with specific proteoforms (Kirkwood et al, 2013). However, these methods are not easily scalable and do not report directly on physical interactions, which can lead to false positive assignments. Furthermore, different structural states of proteins may be insufficiently separated in the chromatographic step, and studying the PPIs of aggregated proteins can be hindered by the elution of aggregates in the void volume together with both interacting and non-interacting proteins. Thermal proteome profiling has also been used to monitor protein complex dynamics in situ (Tan et al, 2018; Becher et al, 2018; Mateus et al, 2018), but does not identify interaction sites and has not been used to probe structure-specific interactions. Finally, in crosslinking coupled to MS (XL–MS) (Iacobucci et al, 2020; Wheat et al, 2021; Leitner et al, 2010; Liu et al, 2017; Holding, 2015; Leitner et al, 2020; Chavez and Bruce, 2019; Liu and Heck, 2015; Leitner et al, 2016), covalent links are formed between proximal amino acid residues to probe PPIs as well as three-dimensional structures and interaction interfaces; however, due to the difficulty of identifying crosslinked peptides, XL–MS yields only modest coverage of the interactome in complex biological samples.

In this study, we report an approach to screen for PPIs in complex proteomes based on limited proteolysis–mass spectrometry (LiP–MS) (Schopper et al, 2017; Feng et al, 2014; Malinovska et al, 2022), our previously developed structural proteomics method

that relies on the brief application of a sequence-unspecific protease, proteinase K, to a cellular extract under native conditions followed by trypsin digestion. These steps generate structure-specific proteolytic fragments that can be measured with MS. We have previously shown that LiP–MS detects protein structural changes (Feng et al, 2014), metabolite– and drug–protein interactions (Holfeld et al, 2023; Piazza et al, 2018, 2020), and other functional events within complex cellular extracts with peptide-level resolution (Cappelletti et al, 2021). We postulated that LiP–MS would detect PPIs since physical interaction between two proteins should alter their protease susceptibility either at the interaction interface itself or in other protein regions that change structurally upon interaction (Konno, 1987; Wilson, 1991; de Pereda and Andreu, 1996; Digiacomo et al, 2017). Thus, adding a protein to a cellular extract should result in altered protease susceptibility of its cellular interactors. These changes in proteolytic patterns could then be detected by quantitative MS analysis to identify interactors of the target protein. Importantly, adding distinct structural states of a protein to the cell extract should enable comparison of their interactomes and thus identification of structure-specific interactions.

Here, we demonstrate that LiP–MS can be applied to the systematic screening for PPIs in complex cellular environments and to detect candidate structure-specific interactomes. We first benchmark the approach by testing its ability to detect known interactions between the respiratory syncytial virus F glycoprotein and its site-specific antibodies, and show that it identified several known antigenic sites including one site directly in a eukaryotic cellular environment. Therefore, the approach enables the identification of protein–protein interaction interfaces. We further show that the method detected known interactions between adenylyl cyclase 8 and calmodulin, as proof-of principle that it can be applied to study interactors of integral membrane proteins. Finally, we applied LiP–MS to screen for structure-specific interactomes of GTP- and GDP-bound forms of Rab GTPases, and of aSyn, a protein involved in PD, for which the mechanisms of toxicity are still largely unknown. We identify known interactors of aSyn monomer and amyloid fibrils, as well as several novel candidate interactors of all bait proteins that will require validation by orthogonal methods. The detection of structure-specific interactors of disease-associated protein structural states should provide novel molecular insights into disease mechanisms and suggest new therapeutic targets.

## Results

### Protein–protein interactions can be detected by LiP–MS

We tested the feasibility of identifying PPIs using the LiP–MS workflow (Fig. 1A) by probing well-characterized interactions in vitro. First, we investigated interactions between respiratory syncytial virus F (RSVF) glycoprotein and several site-specific neutralizing monoclonal antibodies against this target. The RSVF glycoprotein is a class I fusion protein that undergoes a conformational change from a metastable prefusion state to a stable postfusion state during viral entry. We incubated the purified recombinant RSVF glycoprotein stabilized in its prefusion or postfusion state with each of five purified antibodies specific for

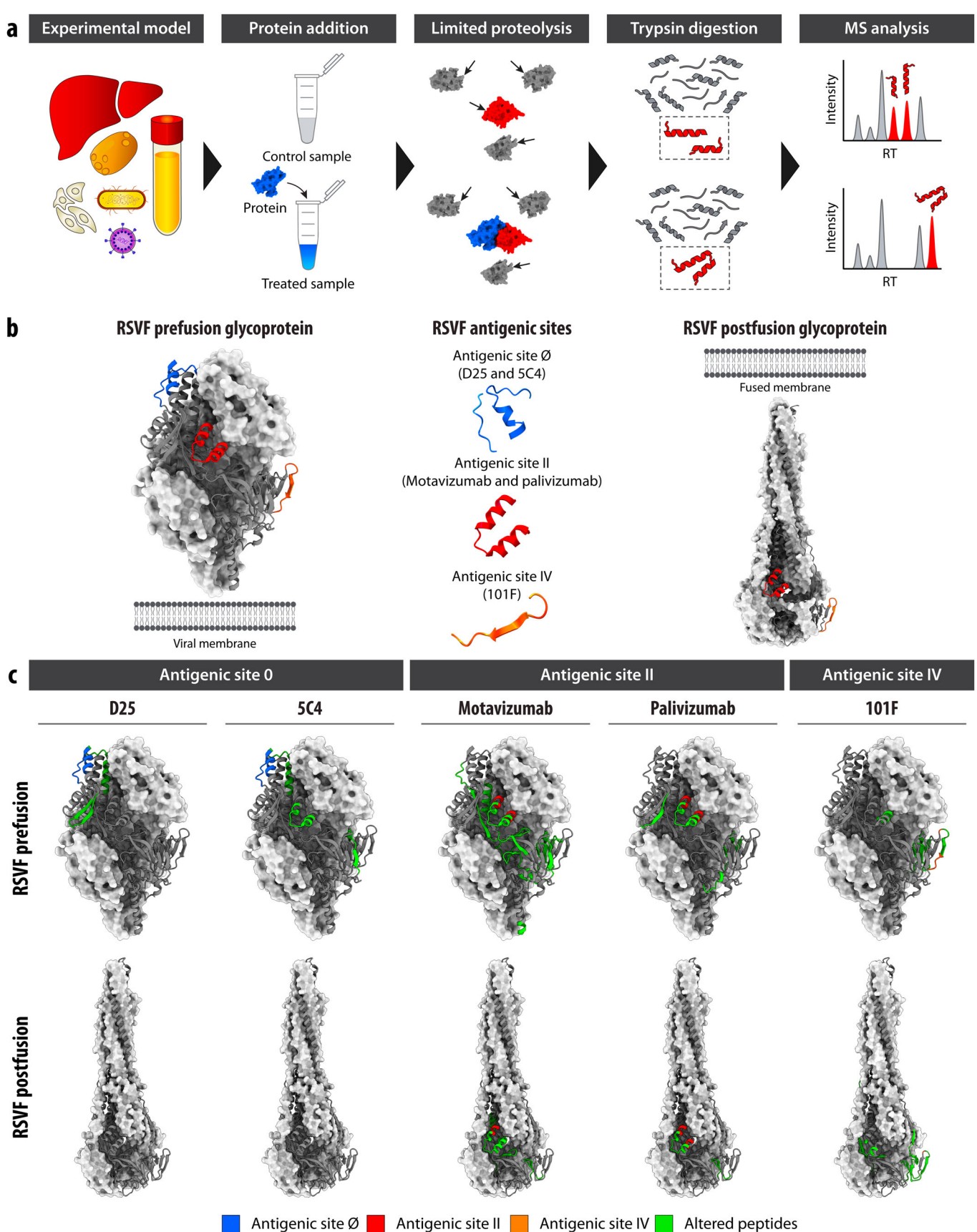

**a**

| Experimental model | Protein addition | Limited proteolysis | Trypsin digestion | MS analysis |

Control sample

Protein

Treated sample

**b**

**RSVF prefusion glycoprotein**

Viral membrane

**RSVF antigenic sites**

Antigenic site Ø
(D25 and 5C4)

Antigenic site II
(Motavizumab and palivizumab)

Antigenic site IV
(101F)

**RSVF postfusion glycoprotein**

Fused membrane

**c**

| Antigenic site 0 | | Antigenic site II | | Antigenic site IV |
| D25 | 5C4 | Motavizumab | Palivizumab | 101F |

RSVF prefusion

RSVF postfusion

■ Antigenic site Ø   ■ Antigenic site II   ■ Antigenic site IV   ■ Altered peptides

◄ **Figure 1. LiP–MS detects protein–protein interactions in purified systems.**

(A) Schematic of LiP–MS workflow. Proteins are extracted from an experimental model, such as tissues, human cells, bacteria, yeast, viruses, or biofluids, under native-like conditions. The extract is then exposed to a protein of interest (treated) or not exposed (control) and subjected to limited proteolysis with proteinase K. Under LiP conditions, proteinase K cleaves solvent-exposed, accessible, and flexible regions, thus generating protein fragments that may differ between the treated and control samples for an interactor of the spiked-in protein. These protein fragments are digested by trypsin under denaturing conditions to produce peptides that are measurable by bottom-up proteomics. By comparing differential peptides between the treated and control samples, interactors of the protein of interest can be identified. (B) Structures of preRSVF (left, PDB: 4JHW) (McLellan et al, 2013) and postRSVF (right, PDB: 3RRR) (McLellan et al, 2011). Known antigenic sites are shown both on the protein structure and in isolation (middle). Blue indicates antigenic site Ø, targeted by antibodies D25 and 5C4. Red indicates antigenic site II, targeted by palivizumab and motavizumab. Orange indicates antigenic site IV, targeted by 101 F. (C) Visualization of structurally altered peptides in green (|log$_2$ FC|>1, moderated $t$-test, $q$ value <0.01, for all comparisons, $n = 4$ replicates each for control and treated samples), on one of the subunit of trimeric preRSVF (upper panel) and postRSVF (lower panel) protein structures upon addition of the indicated antibodies. Antigenic sites are colored as in panel (B).

one of the three antigenic sites Ø, II, or IV (Fig. 1B) or with an unspecific human IgG antibody (referred to as control). We applied the LiP–MS workflow and identified antibody-induced protease susceptibility changes of RSVF based on LiP peptide intensities that were significantly different (log$_2$-fold change, |log$_2$ FC| >1; $q$ value <0.01, moderated $t$-test) between a sample incubated with each site-specific antibody versus control; MS analysis was performed using label-free data-independent acquisition (DIA). We then mapped the significantly altered peptides in preRSVF (PDB: 4JHW) (c) and postRSVF (PDB: 3RRR) (McLellan et al, 2011) onto the three-dimensional structures of the relevant protein conformation (Fig. 1C).

Antigenic site Ø is situated at the trimer apex of preRSVF and consists of a kinked α helix (17 residues) and a disordered loop (7 residues) (McLellan et al, 2013). Antibodies targeting this site (D25 and 5C4) are known to be specific for the prefusion conformation of the RSVF glycoprotein. Consistent with this, incubation with the D25 antibody resulted in four significantly changed LiP peptides on the preRSVF protein (out of 412 detected peptides) (Dataset EV1). No significant changes were observed for peptides of the postfusion protein (out of 427 detected peptides). Furthermore, one of the four altered peptides in the prefusion protein mapped directly to the antigenic site Ø, and the other three altered peptides were situated near the antigenic site Ø (Fig. 1C; we define direct mapping as the identified peptide containing the sequence of the known antigenic site), confirming the binding of D25 to the expected RSVF region. Similarly, the addition of the 5C4 antibody yielded seven significantly changed peptides on the prefusion form of RSVF (out of 447 detected peptides), one of which mapped to the antigenic site Ø, whereas we detected no significant changes for the postfusion protein (out of 408 detected peptides). Although both 5C4 and D25 bind to the antigenic site Ø, these antibodies are known to differ in their vertical and horizontal angles of approach (Tian et al, 2017), which may explain the detection of altered LiP peptides at the antigenic site II.

LiP peptides that change in intensity upon the addition of an antibody may be either fully tryptic (FT) with two tryptic ends or semi-tryptic (ST) where only one end is generated by trypsin cleavage. When located at an antigenic site, ST and FT peptides should, in principle, show different quantitative behavior upon the addition of an antibody. ST peptides that map to an antigenic site on RSVF should decrease in abundance when the antibody is added, since binding reduces susceptibility to proteinase K (PK). In contrast, FT peptides at antigenic sites should show the opposite behavior (i.e., an increase in abundance when the binder is added). The type of peptide (FT or ST) and the

direction of the intensity change are indicated for all peptides (Dataset EV1).

The highly conserved antigenic site II is found on both prefusion and postfusion conformations of RSVF glycoprotein and is recognized by the antibodies palivizumab (Synagis®) and motavizumab (MEDI-524, Numax). For the RSVF prefusion glycoprotein, we identified eight peptides with altered abundances relative to the control due to palivizumab and 31 due to motavizumab binding (out of 453 and 412 detected peptides, respectively) (Dataset EV1). Seven of these peptides showed changes for both antibodies and mapped at or near the known antigenic site II. For the postfusion conformation, we detected five significantly changed peptides upon palivizumab binding (out of 432 detected peptides), four of which were also detected for preRSVF and which again mapped directly to the antigenic site II. For motavizumab, we found 12 differential peptides (out of 414 detected peptides) compared to the control, which were likewise situated either directly at or near the antigenic site II (Fig. 1C; Appendix Fig. S1C,E). The greater number of significantly altered peptides for motavizumab could be because it is an enhanced potency antibody developed from palivizumab and, as such, binds to the target protein with much higher affinity. This is supported by our finding that relative abundance changes of altered RSVF peptides were larger when motavizumab was bound. This observation is also in good agreement with the recent report that small molecules that bind with higher affinity result in higher occupancy and, thus, larger fold changes in LiP peptide abundances (Piazza et al, 2018, 2020).

Antigenic site IV on the RSVF glycoprotein involves an irregular six-residue bulged β-strand epitope and is the major target of 101 F antibody in both prefusion and postfusion forms (McLellan et al, 2010). We observed eight peptides (out of 426 detected peptides) on preRSVF and 11 peptides (out of 428 detected peptides) on postRSVF that changed proteolytic patterns upon 101 F binding (Dataset EV1). As expected, all peptides mapped either at or near antigenic site IV (Fig. 1C). In summary, our data show that LiP–MS detects target protein–antibody interaction interfaces for several well-characterized target protein–antibody pairs under defined, purified conditions. Given that the conformation-specific antibodies D25 and 5C4 specifically caused protease susceptibility changes in the prefusion and not the postfusion form of RSVF, these data also provide a first indication that the approach may be useful for detecting structure-specific interactions.

Since our goal was to systematically identify PPIs within a native cellular environment, we further analyzed the interactions between postRSVF and motavizumab in a complex extract of HEK293T cells. We note that spiked-in RSVF was the most abundant protein in these

lysates (Appendix Fig. S1A). We identified 14 peptides ($|\log_2$ FC| >0.75; moderated $t$-test, $q$ value <0.01; Dataset EV2) that significantly changed in LiP intensity upon the addition of 3 µg motavizumab to the lysate, corresponding to seven proteins (out of 69,263 detected peptides, corresponding to 4582 proteins). Of the 14 changing peptides, eight mapped to the postRSVF glycoprotein, with two out of eight peptides overlapping with peptides detected in vitro. Further, a score measuring effect strength (the $|\log_2$ FC| divided by the adjusted $p$ value) indicated a larger effect for RSVF peptides (median score of 3651) relative to all peptides (median of 141) (Dataset EV2). Structural changes detected in the other six proteins could have resulted from direct interactions of RSVF or motavizumab with these proteins in the lysate or from indirect structural effects. Similarly, contaminant proteins present in the preparations of postRSVF or the antibody could interact with proteins in the lysate or otherwise cause indirect structural changes.

Next, we performed a dose-response experiment to better distinguish between true and false positive hits, as we have previously done to identify small-molecule–protein interactions (Piazza et al, 2020; Holfeld et al, 2023). We exposed the HEK293T cellular extract, supplemented with 1 µg of postRSVF, to five concentrations of motavizumab, and identified peptides that showed high correlation ($r$) to a sigmoidal trend of the peptide-intensity response profile. Of the 14 peptides identified in the single-dose experiment, the intensity responses of eight peptides were inversely proportional to the amount of motavizumab with a high correlation ($r > 0.85$; Fig. 2A; Appendix Fig. S1B,D). Importantly, all eight were postRSVF peptides, and all were mapped onto or near the antigenic site II (Fig. 2B,C), indicating that a dosage series of the target protein can help identify true positive direct interactors in a complex lysate. All of these eight peptides were ST and are thus expected to decrease in abundance with increasing amounts of motavizumab.

An analysis of all peptides that map to known antigenic sites on RSVF and show intensity changes upon spike-in of any one of the five tested antibodies (D25, 5C4, Motavizumab, Palivizumab, 101 F) showed that all 11 such peptides (three FT peptides, eight ST peptides) indicate increased protease protection upon addition of the antibody. In contrast, analysis of all RSVF peptides with altered intensity upon antibody addition (i.e., including peptides at epitopes, but also those merely near epitopes, or more distant) showed a more complex picture. Out of the 57 peptides we detected in all experiments, 63% (6 FT and 30 ST peptides) indicate increased protection, and 37% (10 FT and 11 ST peptides) indicate decreased protection upon antibody addition. Note that FT peptides are generally more reliable in analyses of protease susceptibility since ST peptides can also result from secondary cleavages (i.e., additional cleavage of a previously generated ST peptide). Nevertheless, these data indicate that some of the protease susceptibility changes we detect are unlikely to represent a footprinting effect, i.e., a masking of the antigenic site by the antibody, but may be due to other structural changes that occur as a secondary consequence of binding.

Overall, our data show that in situ LiP–MS analysis enables the identification of PPIs and pinpoints interaction interfaces in complex biological matrices.

## LiP–MS detects protein–protein interactions with integral membrane proteins

Integral membrane proteins (IMPs) represent a biologically interesting set of proteins as they constitute a large proportion of

therapeutic targets in drug discovery, but they are challenging to study using proteomics. In a second proof-of-principle investigation of a known interaction pair, we asked whether LiP–MS enables the identification of PPIs of membrane proteins. We tested the applicability of LiP–MS to detect the interaction of calmodulin (CaM) with membrane-integral adenylyl cyclase type 8 (AC8) (Fig. 3). CaM is an intracellular $Ca^{2+}$-binding protein that is known to interact with CaM-binding domains (CaMBDs) in the N-terminus and in the C-terminal cytoplasmic regulatory subdomain (AC8-C2b) of AC8 (Fig. 3A) (Gu and Cooper, 1999; Herbst et al, 2013). We applied the LiP–MS workflow to crude membrane preparations from HEK293S GnTI- cells engineered to overexpress bovine AC8 fused at its C terminus to TwinStrep-YFP, incubated with a six-dose concentration series of bovine CaM. The coverage of membrane-annotated proteins was better in the crude membrane preparations than in standard cell lysates from which membranes had been removed (Fig. 3B; Dataset EV3), and we also identified more membrane-annotated proteins overall ($n = 3037$ in membrane preparations versus 1506 in whole lysates). We also observed good sequence coverage of our target AC8 (220 peptides covering 58.5% of the AC8 sequence) in the crude membrane preparation, although we did not detect peptides from the transmembrane domains (Fig. 3C), as expected in any bottom-up proteomics experiment, due to their hydrophobicity.

Upon addition of CaM to the membrane preparations, 279 peptides were significantly altered (of 91,847 peptides detected, corresponding to 5185 proteins) relative to the no-treatment control ($r > 0.85$, $|\log_2$ FC| >0.75, moderated $t$-test, $q$ value <0.01; Dataset EV3). These peptides mapped to 163 proteins. Amongst these, 16 peptides with high correlation to sigmoidal profiles ($r > 0.85$; Appendix Fig. S2) originated from AC8 and mapped exactly to the N-terminal AC8-CaMBD and the C-terminal AC8-CaMBD (Fig. 3C,E). These data confirmed that LiP–MS detects CaM binding and pinpoints known binding sites. The changing peptides on AC8 could not all be detected in whole cellular lysates (Fig. 3D), again pointing to improved coverage of membrane proteins in membrane preparations.

We then examined the larger set of proteins that underwent structural alterations upon CaM addition to the crude membrane preparation. We searched for canonical CaM-binding motifs within the sequences of all proteins for which we detected structural alterations upon CaM addition and showed that 85 of the 279 significantly altered peptides (corresponding to 56 proteins, of which 37 are membrane-annotated proteins) are predicted to contain CaM-binding motifs (Mruk et al, 2014); however, we found no enrichment for CaM-binding motifs in the full set of candidate interactors. Overall, our data demonstrate that the LiP–MS pipeline detects protein interactors of IMPs with soluble proteins and enables the identification of interaction interfaces in situ in detergent-free crude membranes.

## Differential interactomes of alpha-synuclein monomer and amyloid fibrils

Having established that LiP–MS can detect known protein–protein interactions, we next applied it in a discovery context. Since any stable structural form of a protein can be used as a spiked-in bait, we assessed the ability of the approach to characterize differential interactors of two different structural forms of a protein. We first

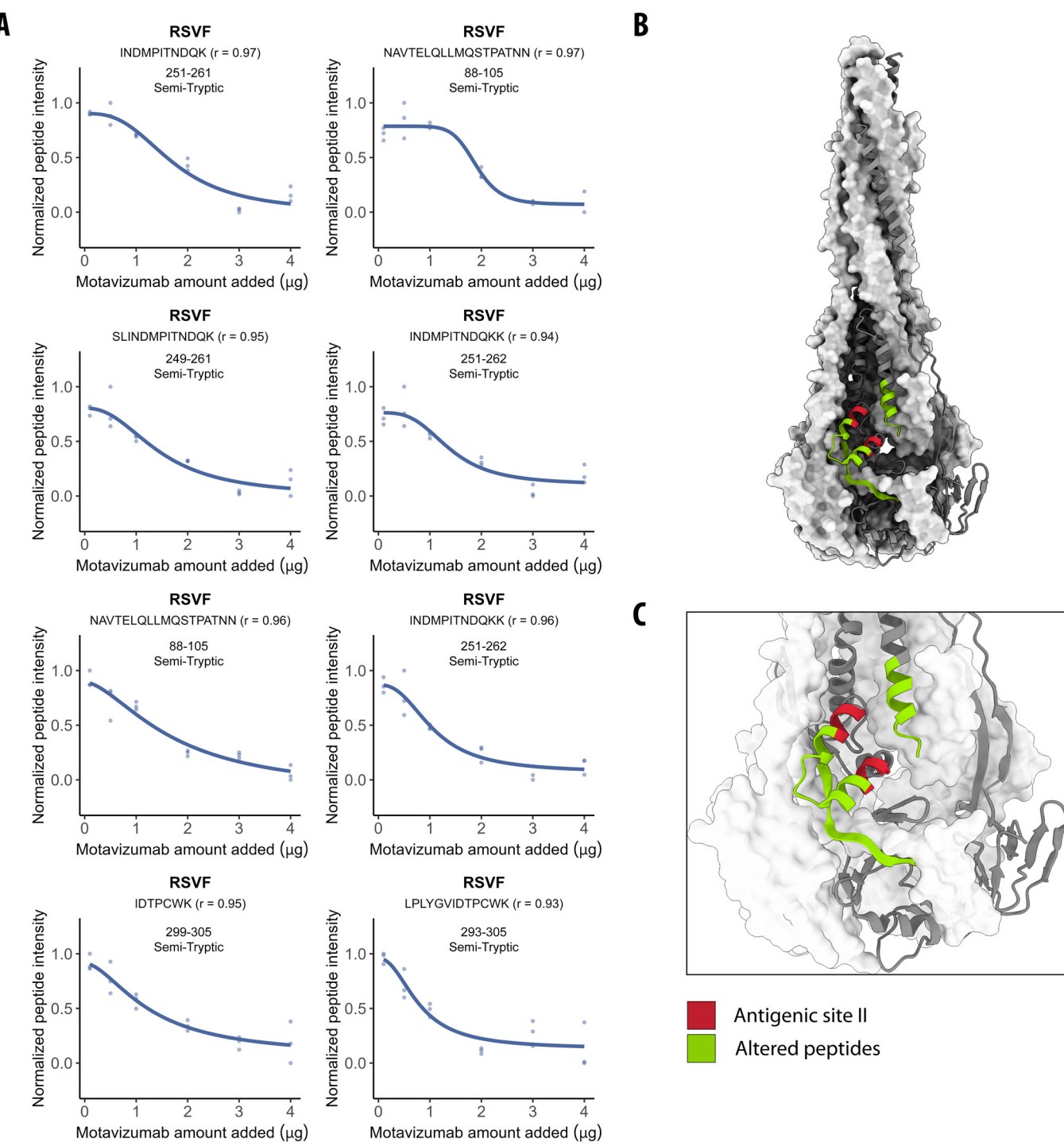

**Figure 2. LiP–MS detects protein–protein interactions in complex proteomes.**

(**A**) Dose-response curves of eight LiP peptides with indicated amino acid positions originating from postRSVF show relative peptide intensities proportional to the amount of motavizumab spiked into HEK293T cellular extracts ($n = 3$ replicates each). Pearson's coefficient ($r$) to a sigmoidal trend of the peptide-intensity response profile is indicated. These peptides correspond to the altered peptides (green) in panels (**B**) and (**C**). (**B**) The structure of postRSVF (PDB: 3RRR) (McLellan et al, 2011) with peptides altered in the dose-response analysis ($r > 0.85$; |log$_2$ FC|>0.75; moderated $t$-test, $q$ value <0.01) indicated in green and antigenic site II in red. (**C**) Zoom of the altered peptides on the structure of postRSVF (PDB: 3RRR) (McLellan et al, 2011) with colors as in panel (**B**).

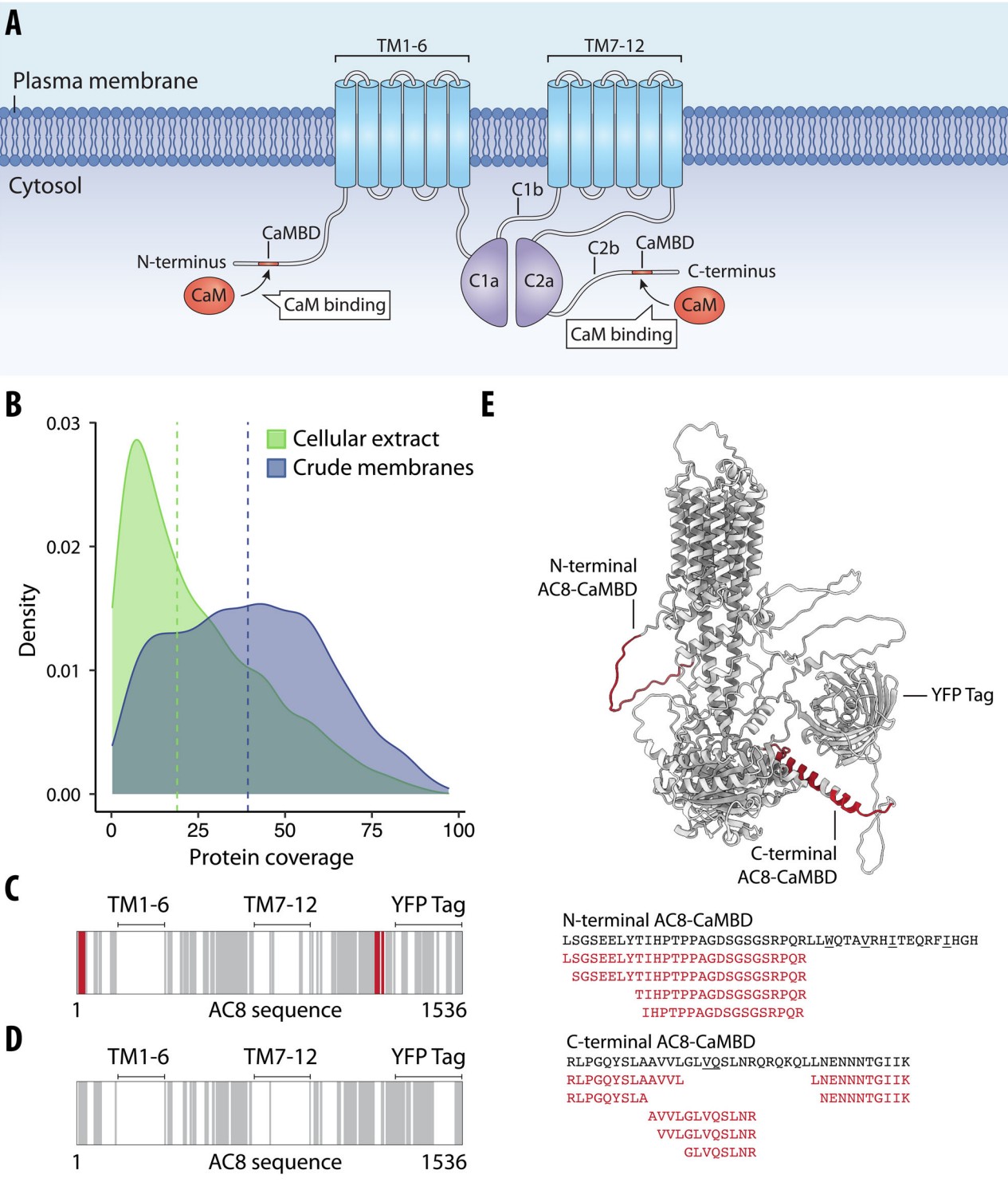

**Figure 3. LiP–MS detects interactors of integral membrane proteins in crude membranes.**

(A) Schematic of AC8 with the CaMBD in the N-terminus, transmembrane domains 1–6 and 7–12 (TM1–6 and TM7–12), and catalytic domains C1a, C1b, C2a, and C2b indicated. (B) Distribution of protein coverage for membrane-annotated proteins identified in crude membrane preparations of HEK293S GnTI- cells (blue) and in HEK293T cellular extracts (green). Blue and green vertical lines indicate calculated median coverages of 29.6% and 17.6%, respectively. (C, D) Protein sequence coverage of bovine AC8-YFP in LiP–MS in crude membranes (C) and in cellular extracts (D) is visualized. The barcodes depict peptides along the AC8-YFP sequence. Gray represents detected peptides, white represents non-detected regions, and red represents peptides that were significantly altered upon CaM addition ($r > 0.85$, |log2 FC|>1, moderated $t$-test, $q$ value <0.01). (E) AlphaFold2 (Varadi et al, 2022; Jumper et al, 2021) predicted structure of AC8 (including the tag domain) with peptides altered upon CaM addition, highlighted in red. Amino acid sequences comprising the CaM-binding motifs of AC8 are depicted in black. Hydrophobic residues of the CaM-binding motif are underlined. The significantly altered peptides upon CaM addition are shown in red.

tested the Parkinsons's disease (PD)-associated protein alpha-synuclein (aSyn). In PD, aSyn aggregates into fibrillar structures in neuronal cells, but mechanisms of toxicity remain unclear (Wong and Krainc, 2017). One hypothesis is that, upon aggregation, aSyn undergoes changes in its interactome that underlie disease development (Leitão et al, 2021; Lassen et al, 2016; Betzer et al, 2015; van Diggelen et al, 2020). We thus applied our LiP–MS approach to assess whether monomeric and aggregated, fibrillar structural states of aSyn have different cellular interactomes. We generated a cellular extract of cortical neurons differentiated from an SNCA-knockout (KO) induced pluripotent stem cell (iPSC) line (Appendix Fig. S3), to avoid possible effects of endogenous aSyn on the analysis (Fernandes et al, 2016; Zambon et al, 2019; Haenseler et al, 2017a). We purified acetylated aSyn monomer, which is considered to be the physiologically relevant form (Burré et al, 2013; Fauvet et al, 2012; Runfola et al, 2020), and generated aSyn amyloid fibrils in vitro, ensuring the conformations of our preparations using SEC, TEM, native-PAGE, and ThT fluorescence as quality control steps (Appendix Fig. S4). Subsequently, we spiked increasing amounts of aSyn monomer or fibrils into lysates of SNCA-KO iPSC-derived neurons. We then performed LiP–MS experiments in a dose-dependent manner to identify the resulting protein structural alterations across the proteome and thus putative interactors of the monomeric and amyloid fibril conformational states of the protein.

We identified 68 and 242 significantly changing peptides upon addition of aSyn monomer and amyloid fibrils to the cellular extracts, respectively ($r > 0.85$, $|\log_2 \text{FC}| > 0.75$, moderated $t$-test, $q$ value <0.01; Dataset EV4) (out of 90,416 and 85,084 detected peptides, corresponding to 5435 and 5536 proteins) (Fig. 4A). A total of 64 proteins showed structural changes upon spike-in of aSyn monomer and 225 proteins upon spike-in of aSyn fibrils, indicating a higher apparent binding to aSyn fibrils compared to monomer. Several putative aSyn interacting proteins displayed monomer-specific ($n = 50$; Dataset EV4) and fibril-specific changes ($n = 211$; Dataset EV4) (Fig. 4B). In general, 14 putative interactors were found to be conformation-unspecific (CANX, CCT2, EEF1A1, FARSB, PAF1, PEBP1, PIN1, RBM8A, RPS27A, SEC13, SMC3, SPTAN1, VPS52, and YWHAB), nine of which are known vesicular proteins, thus supporting evidence that aSyn localizes with vesicles (Ebanks et al, 2020). Five of the 14 proteins were previously reported to bind aSyn in the STRING database (CANX, EEF1A1, PAF1, PIN1, and RPS27A).

First, we analyzed proteins ($n = 64$; Dataset EV4) that showed structural changes upon treatment with aSyn monomer. This set of proteins was significantly enriched ($p$ value <0.01, Fisher's exact test) for known interactors of aSyn, containing ten proteins that were previously classified as physical interactors of aSyn in the STRING database (interaction score >150) (CALM1, CANX, EEF1A1, ILF3, MAP1B, PAF1, PIN1, RPS27A, VIM, and YWHAZ) (Fig. 4C), out of the 237 such STRING interactor proteins that we detected experimentally. Interestingly, the interaction between CALM1 and aSyn was reported to be monomer-specific (Lee et al, 2002), consistent with our data. In addition, we identified structural changes in several proteins, such as AGRN, SYNJ1, MAP1B, and YWHAZ, which have been implicated in PD based on disease-gene associations mined from literature, and in peroxiredoxin-1 (PRDX1), which has been linked to neurodegenerative processes (Hallacli et al, 2022; Szeliga, 2020). A functional enrichment (GO) analysis of the putative interactors of

aSyn monomer showed enrichment for RNA-binding, protein-binding, and protein-specific domain-binding molecular functions ($q$ value <0.01, Benjamini–Hochberg FDR, minimum hypergeometric test; SimRel functional similarity, size = 0.7) (Fig. 4E), consistent with the known interaction of aSyn with proteins involved in mRNA translation (Hallacli et al, 2022; Chung et al, 2017). Putative interactors were also enriched for extracellular organelles, cell junction, and vesicles (GO cellular components), consistent with the known localization of aSyn to presynaptic terminals, its interaction with synaptic vesicles, and its roles in exocytosis, endocytosis, and vesicle recycling (Huang et al, 2019). Finally, putative interactors were enriched (GO biological processes) for the establishment of protein localization to the mitochondrial membrane, peptide biosynthetic process, and cellular localization, supporting prior evidence for aSyn monomer involvement in mitochondrial bioenergetics (Ludtmann et al, 2016) and membrane transport (Huang et al, 2019).

Next, we examined proteins ($n = 225$; Dataset EV4) that were structurally altered in neuronal lysates upon spike-in of aSyn fibrils. This set was again enriched in previously defined aSyn interactors (STRING; interaction score >150), including 25 such proteins (CALR, CANX, CCT3, DNMT1, DYNC1H1, DYNLL1, EEF1A1, EFTUD2, FKBP1A, GAPDH, HSP90AA1, HSP90AB1, HSPA1A, HSPA8, HSPD1, MAP2K2, PAF1, PIN1, PPP3CA, PREP, RAB3A, RPS27A, RPS3, SMU1, and SNRNP200) ($p$ value <0.01, Fisher's exact test) (Fig. 4D) out of the 235 STRING-defined interactors we detected experimentally. As for monomeric aSyn, we identified proteins previously reported to interact exclusively with aggregated forms of aSyn: the mitochondrially localized protein superoxide dismutase 2 (SOD2), which interacts specifically with fibrillar aSyn, and the vesicular ras-related protein Rab-3A (RAB3A) which preferentially interacts with oligomeric and aggregated aSyn (Chen et al, 2013; Tan et al, 2022; Huang et al, 2018). Among other proteins structurally altered upon the addition of aSyn fibrils, we also identified the amyloidogenic protein gelsolin (GSN), a component of PD-associated intraneuronal inclusions of which aggregated aSyn is a major component (Welander et al, 2011), as well as the actin-binding protein cofilin-1 (CFL1), which is known to co-aggregate with aSyn fibrils and is implicated in pathogenicity in PD (Tan et al, 2022; Yan et al, 2022). Notably, we further identified four components of chaperonin-containing T-complex (CCT2, CCT3, CCT6A, and CCT8) that play a central role in protein folding, degradation, aggregation, as well as in the inhibition of aSyn aggregation (Ghozlan et al, 2022; Sot et al, 2017; Grantham, 2020).

We compared our data with previous proteomic analyses of the content of Lewy Bodies (LBs) either in a neuronal aSyn fibril seeding model (Mahul-Mellier et al, 2020) or in postmortem patient brains (Petyuk et al, 2021; Xia et al, 2008). We found an enrichment of LB-associated proteins in the set of putative aSyn fibril interactors. In total, 49 and 38 of our fibril-dependent structurally altered proteins (out of 225 detected proteins) were previously identified as components of neuronal LBs after 14 and 21 days of cell treatment with aSyn fibrils, respectively, with a significant enrichment over all detected proteins that overlapped with the previous study (734/422 respectively) ($p$ value <0.01, Fisher's exact test; Appendix Fig. S5A,B). When we compared our data with a previous study that investigated LBs purified from postmortem brain tissues of patients diagnosed with the LB variant of AD (Xia et al, 2008), we found that 14 of the proteins with

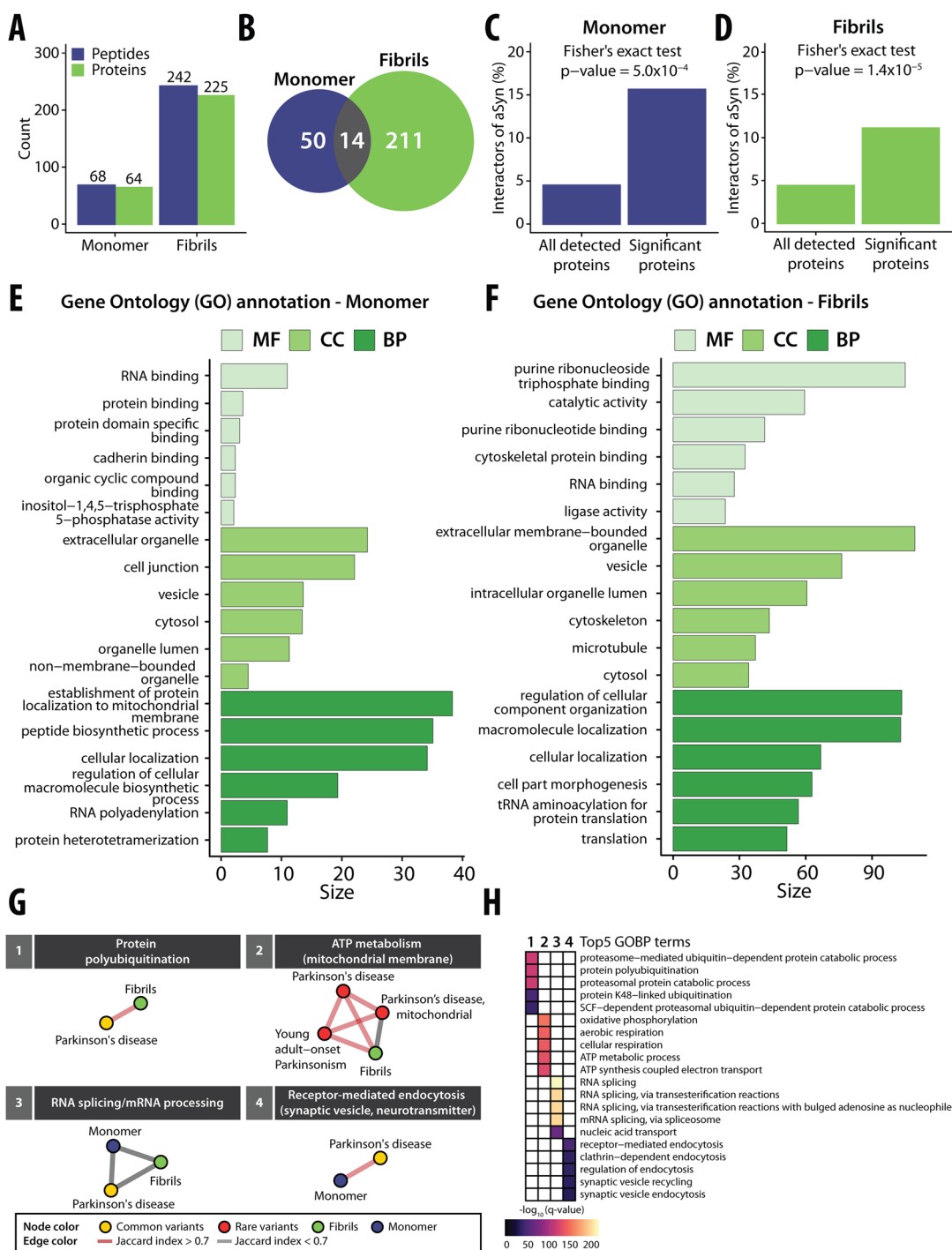

**Figure 4. A systematic investigation of structure-specific interactors of the amyloidogenic protein aSyn and gene module analysis of Parkinson's disease.**

(A) Barplot with the numbers of altered LiP peptides (blue) and corresponding structurally altered proteins (green) for aSyn monomer (left) and amyloid fibrils (right). (B) Venn diagram with the numbers of structurally altered proteins for aSyn monomer (blue) and for amyloid fibrils (green). The overlap of structurally altered proteins identified for both aSyn monomer and amyloid fibrils is indicated in gray. (C) The plots show the fraction of known aSyn interactors (based on the STRING database) in structurally altered proteins (right) versus all detected proteins (left) upon spike-in of aSyn monomer into an iPSC-derived cortical neuron extract. The $p$ value assessing enrichment (Fisher's exact test) is shown. (D) Enrichment plot as in (C) upon spike-in of aSyn fibrils. (E, F) Functional enrichment analyses of structurally altered proteins upon spike-in of aSyn monomer (E) or fibril (F), based on the indicated ontologies (molecular function in light green, cellular component in green, biological process in dark green); the plots show the size (i.e., a score calculated based on $q$ value) of the top6 significant gene ontology terms upon removal of redundant terms ($q$ value <0.01, Benjamini–Hochberg FDR, minimum hypergeometric test, SimRel functional similarity, size = 0.7). (G) Identified modules with enriched GOBP ($q$ value <0.05, Fisher's exact test, one-sided) that are linked to either common (yellow node) or rare variants (red node) of PD genes for aSyn monomer (blue) or fibrils (green). The thickness of lines represents the Jaccard index (red for Jaccard index >0.7, gray for Jaccard index <0.7). (H) Heatmap showing the top4 GOBP terms within each module as indicated in (B). The gradient color indicates the significance based on the results of a GOBP enrichment test (purple = low significance, yellow = high significance).

structural changes (p values <0.05 and an absolute fold-change |FC| >0.5) overlapped with the previously defined LB proteins, of the 67 overlapping proteins detected overall, with a significant enrichment (p value <0.01, Fisher's exact test) (Appendix Fig. S5C). This set of proteins included six known aSyn interactors (CALR, CANX, DYNC1H1, GAPDH, HSP90AB1, and RPS3), as well as several mitochondrial proteins (e.g., ACO2, ATP5PB, IDH2, NDUFS1, and RPS3), calcium-binding proteins of the calreticulin protein family (CALR, CANX) (Davidi et al, 2020) and gelsolin (GSN). When we assessed the overlap of our data with proteins previously identified in LBs from PD cases with dopaminergic neuronal loss (NL) (Petyuk et al, 2021), we found that 17 out of 156 overlapping proteins detected overall showed structural changes upon fibril spike-in, with a significant enrichment (p value <0.01, Fisher's exact test) (Appendix Fig. S5D). Within this set of 17 proteins, we detected SEC31A (SEC31 Homolog A, COPII coat complex component), which is involved in ER to Golgi transport and macroautophagy (Antoniou et al, 2022) and was shown to exhibit altered protein levels in neurons. In particular, we also identified peptidyl-prolyl cis-trans isomerase FKBP1A, which was observed to promote the aggregation of aSyn and cause abnormal, nonlinear, hydrophobic aggregation of aSyn (Caminati and Procacci, 2020).

Functional enrichment analysis (q value <0.01, Benjamini–Hochberg FDR, minimum hypergeometric test, SimRel functional similarity, size = 0.7) on the set of putative fibril interactors showed enrichment in purine ribonucleoside triphosphate binding, including ATP- and GTP-binding proteins, proteins with catalytic activity, and cytoskeletal protein binding (molecular functions), in extracellular membrane-bounded organelles, vesicles, microtubules, or cytoskeleton (cellular components) and in the regulation of cellular component organization and macromolecule localization (biological processes) (Fig. 4F). Several of these terms correspond to processes and pathways known to be modulated in neurodegenerative diseases, including PD (Wong and Krainc, 2017; Lassen et al, 2016).

Next, we asked whether the proteins we identified as structurally altered upon spike-in of aSyn could be functionally linked to Parkinson's disease-associated genes that have either been directly implicated in disease or have been identified in genome-wide association studies (GWAS). While our candidate interactors did not include such disease genes, it is well known that GWAS will miss many important genes due to a lack of association power. We, therefore, asked whether the candidate interactors are part of pathways or complexes that contain PD disease genes. For this analysis, we used a comprehensive interaction network consisting of physical or functional interaction data and performed network-based expansion (the personalized PageRank algorithm) followed by walktrap clustering for monomer- and fibril-interacting proteins and known PD-associated genes, enabling the identification of gene modules and shared biological processes that overlap with PD-associated genes, as described previously (Barrio-Hernandez et al, 2021). Within our set of structurally altered proteins, we found four significant modules (Fig. 4G) enriched for genes involved in biological processes that are associated with common or rare variants of PD-associated genes (q value <0.05, Benjamini–Hochberg FDR, Fisher's exact test, one-sided) (Fig. 4H), specifically protein polyubiquitination (module 1), ATP metabolism (module 2), RNA splicing/mRNA processing (module 3), and receptor-mediated endocytosis (module 4). We only identified

modules corresponding to protein polyubiquitination and ATP metabolism upon the addition of fibrillar but not monomeric aSyn, indicating that these processes may be impaired in PD via interactions with aSyn amyloid fibrils. On the contrary, endocytotic processes, such as synaptic vesicle recycling and clathrin-dependent endocytosis, were specifically enriched for aSyn monomer. In summary, these results show that our method can identify aSyn conformation-specific structurally altered proteins that are functionally linked to cell biological processes relevant to PD.

Due to the pathological relevance of aSyn amyloid fibrils, we further probed putative fibril-specific interactions using an orthogonal approach. We spiked either monomeric or fibrillar aSyn, or a PBS control, into SNCA-KO iPSC extracts and used quantitative MS to identify proteins that co-precipitate with the insoluble aSyn fibrils upon ultracentrifugation. As expected, aSyn monomer was predominantly recovered in the soluble fraction and Syn fibrils in the pellet. We identified 574 proteins (|FC| >1.5, moderated t-test, q value <0.05) that were either enriched in the pellet exclusively upon fibril spike-in or depleted from the supernatant upon spike-in of aSyn monomer or PBS control. These proteins may be direct binders of aSyn fibrils and were indeed enriched in the set of proteins with fibril-dependent structural changes identified by LiP–MS (p value <0.01, Fisher's exact test) (Appendix Fig. S5E). The overlap between putative aSyn binding proteins in the two datasets is relatively small (53 proteins; ~24% of the identified LiP hits), as expected due to the experimental differences between LiP–MS and ultracentrifugation. With LiP–MS, we probe protein structural changes that can result from direct interactions of aSyn fibrils with other proteins in the lysate irrespective of their binding affinity, or from other indirect structural effects. In contrast, low-affinity interactions may be disrupted during ultracentrifugation. Consistent with this, analysis of the LiP–MS dose-response curves for the candidate aSyn fibril interactors yielded an average half-maximal response concentration of 1.7 μg for the 53 hits detected with both techniques and of 2.7 μg for those detected only with LiP–MS (Appendix Fig. S5F); this may indicate a higher affinity of the former set (but see Discussion). These candidate binders identified by both assays are likely high-confidence interactors and are provided (Dataset EV4). An independent investigation of the set of proteins co-immunoprecipitated from SH-SY5Y lysates with monomeric aSyn showed that 7 out of the 64 candidate interactors identified by LiP–MS showed evidence for physical interaction with monomeric aSyn (Dataset EV4).

Taken together, LiP–MS identified several known as well as novel putative interactors of aSyn. Although some of these have previously been shown to exclusively interact with aSyn monomer or fibrils, in most previous studies, it is not clear if proteins are interactors of the monomeric or the aggregated protein, and aSyn conformation-dependent interactions were typically probed only for a few proteins (van Diggelen et al, 2020; Betzer et al, 2015; Leitão et al, 2021). In comparison, our results demonstrate that we can systematically compare which interactions occur with either of the structural states of aSyn in situ and, most importantly, directly in complex cellular extracts without prior labeling and purification. Our study thus provides a dataset of putative interactors for the monomeric and fibrillar states of aSyn, including interaction interfaces, that will be a rich resource for future follow-up studies.

## Differential interactomes of Rab GTPases

To assess whether our approach could detect candidate interactors of different protein conformations that are expected to be quite similar, we compared the effects on the proteome of GTP- and GDP-bound forms of two Rab GTPases, Rab2A and Rab5A (Fig. 5A). These two Rabs are both key regulators of membrane traffic and both bind many different effectors in a GTP-dependent manner and so recruit them to membranes (Homma et al, 2021; Hutagalung and Novick, 2011). We spiked purified forms of each of these proteins, in each case engineered to carry mutations known to lock them in the GDP- or the GTP-bound form (Stenmark et al, 1994; Li and Stahl, 1993) into HEK293T cell lysates, and then used our dose-response LiP–MS workflow to identify proteins showing protease susceptibility changes in response. We identified several significantly changing LiP peptides and proteins in response to the addition of Rab2A-GTP, Rab2A-GDP, Rab5A-GTP, and Rab5A-GDP (Fig. 5B,C). The set of proteins with protease susceptibility changes was different, or the same protein had different response strengths, for the GTP- and GDP-bound forms of both Rabs (Fig. 5D–F), indicating differences in the candidate interactors of the two conformational states of these proteins. This corroborates the capability of our approach to detect differential interactomes of alternative protein conformations.

Most known effectors of Rab2A-GTP and Rab5A-GTP are low-abundance proteins and were not detected mass spectrometrically in these experiments. The 5 known effectors we detected did not show protease susceptibility changes upon Rab2A or Rab5A spike-in, possibly because of low sequence coverage (average of 36.6%). For each form of each Rab protein, our candidate interactors were significantly enriched in proteins previously identified as potential interactors in a proximity labeling approach (MitoID, based on BioID proximity labeling of mitochondrially-targeted proteins) (Gillingham et al, 2019) (Appendix Fig. S6). However, these enriched hits were previously scored as having low specificity for a given Rab relative to multiple ($n = 11$) Rab GTPases in the context of MitoID, suggesting that this overlap between the two approaches may be driven by high-abundance proteins. Despite this, the set of our candidate interactors that overlapped with MitoID hits differed between the Rab2A and Rab5A (Fig. 5G–I). In summary, these data show that our approach identifies differential interactomes also of proteins that differ only slightly in conformation, for which developing specific antibodies could be challenging. However, the results also suggest that the approach penalizes the detection of low-abundance interactors since it is less likely, in this case, to detect the interaction site.

## Discussion

Interactomics studies are challenging for proteins that adopt multiple structural states within the cell. Here, we present a structural proteomics approach that enables screening for structure-specific interactomes of any protein that can be purified in a defined state or controllably switched between states. We validated the ability of LiP–MS to map known PPIs between purified proteins as well as between proteins in complex biological matrices, and applied the approach to screen for differential interactomes for the monomeric and fibrillar states of the

PD-associated protein aSyn, and for different nucleotide-bound forms of Rab GTPases. We have shown that LiP–MS identifies altered proteolytic patterns upon protein–protein binding and that its peptide-level resolution enables the identification of PPI interfaces in situ. Knowing interaction interfaces is useful for the structural characterization of protein complexes, the introduction of mutations to disrupt interactions, and the potential development of drugs that target specific PPIs of interest. We emphasize that our approach is a first screening step for the identification of structure-specific interactomes and that candidate interactors require orthogonal validation.

In a complex background such as a cell lysate, spiking in multiple doses of a bait protein may help differentiate directly from secondary targets or high from low-affinity interactions and thus is likely to improve the identification of true positives. The use of dose-response curves for this purpose is derived from our LiP–Quant approach (Piazza et al, 2020), which we previously showed could help prioritize direct interactors of small molecules. In contrast to LiP–Quant, we do not use measures of non-specific background or other ranking criteria for the identification of protein–protein interactors; hits are instead assessed solely by how well the data fit a sigmoidal dose-response curve. We also derive half-maximal response concentrations from these curves, which should reflect relative in situ binding affinities and may allow the ranking of PPIs on this basis. For protein–small molecule interactions, we have previously shown that LiP–MS data can be used in this way, namely that dose-response curves can yield quantitative binding parameters that allow ranking of hits based on them and prioritization of targets for further investigation (Piazza et al, 2020). However, applicability to protein–protein interactions requires further testing against ground-truth data sets. Despite the increase in experimental and instrumental resources, multiple-dose experiments are therefore valuable in certain contexts. Note, however, that dose-response curves are not strictly needed for using LiP–MS to identify candidate protein interactors; a single-dose experiment may also be used for this purpose, especially if secondary screens are available for the identification of direct targets.

Our demonstration that LiP–MS detects interactions between site-specific antibodies and conformers of RSVF has implications in the field of antibody development beyond the validation of our method for the detection of PPIs. In experiments with purified proteins, our method pinpointed the locations of the three well-characterized antigenic sites on prefusion and postfusion RSVF conformers. We detected no interactions between antigenic site Ø-specific antibodies and postRSVF. This was expected since this antigenic site is known to be solvent-inaccessible in the postfusion conformation. This ability to accurately detect site- and conformation-specific interactions between antibodies and target proteins should be useful for the identification of structure-specific antigenic sites as well as the characterization of novel antibodies, in particular when high-resolution structures are difficult to obtain. Importantly, the dose-response experiment in which a site-specific antibody was added to whole cell extracts was able to identify the specific target interaction, interaction interface, and relative protein binding parameters as previously demonstrated for small molecules (Piazza et al, 2020). This type of experiment is expected to become a tool for screening for off-target binders in both basic and pharmaceutical research.

                                                                 

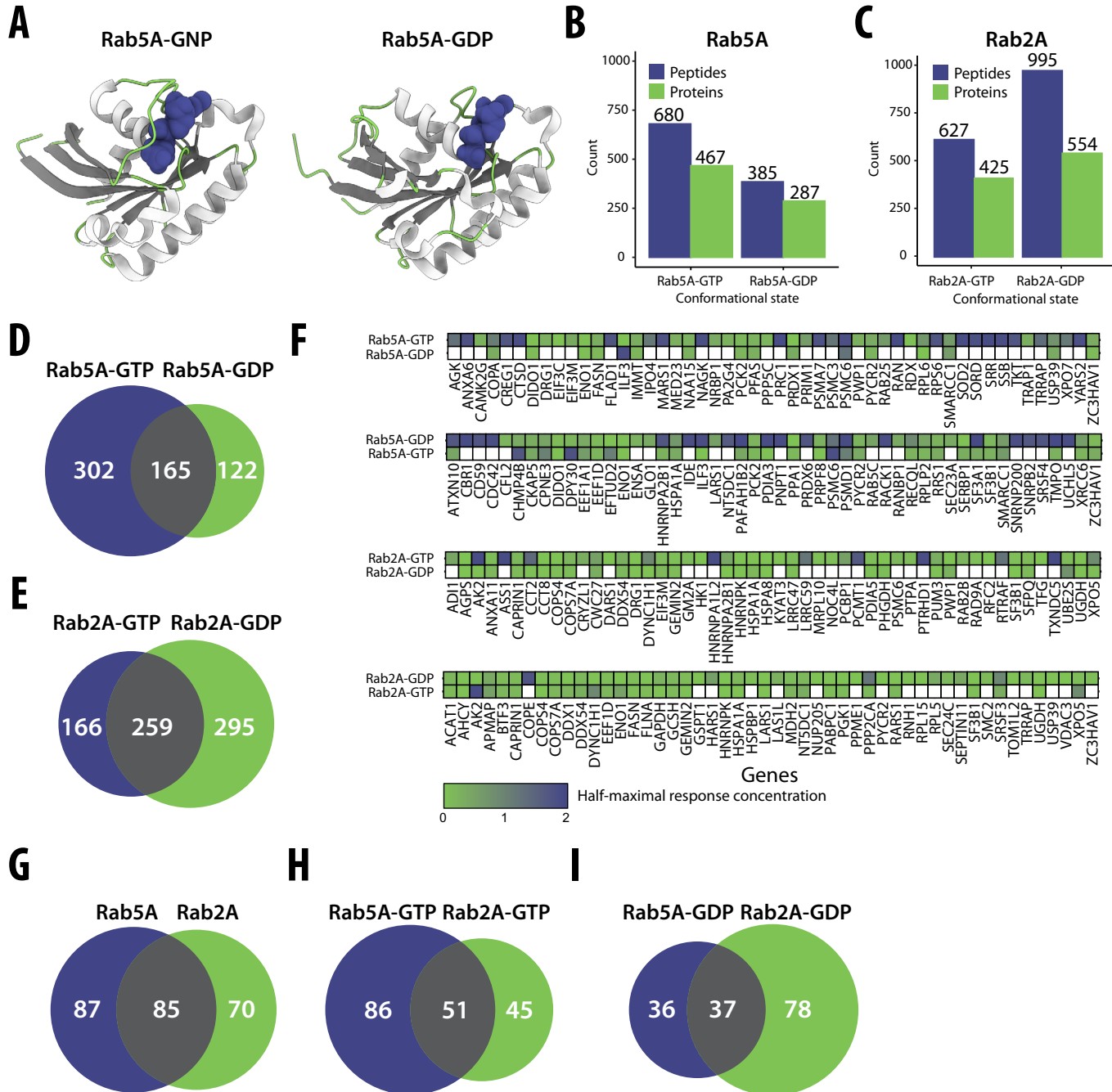

**Figure 5. A systematic investigation of structure-specific interactors of Rab5A and Rab2A GTPases.**

(A) Crystal structures of Rab5A bound to guanosine-5′-[(β,γ)-imido]triphosphate (GNP) (PDB: 1N6H) and guanosine diphosphate (GDP) (PDB: 1TU4), colored according to their secondary structural elements, are shown to underscore the conformational differences between the GNP- and GDP-bound states of Rab5A. Note that the GNP-bound structure mimics the GTP-bound form. Coil regions are highlighted in green, β-sheets in gray, and α-helices in white. The bound GNP and GDP molecules are depicted in blue, demonstrating the nucleotide interaction sites. (B, C) Barplot with numbers of significantly changing peptides (blue) and corresponding proteins (green) for Rab5A (B) or Rab2A (C) in their GTP- (left) or GDP-bound (right) forms. (D, E) Venn diagrams with the overlap of proteins identified as significantly changing upon the addition of GTP and GDP-bound forms of Rab5A (D) or Rab2A (E). (F) The plots illustrate the extent of overlap between the two nucleotide-bound forms of a bait Rab protein for the top hits in our interaction screen. In each case, the upper row shows the top 50 hits for the indicated bait protein (e.g., Rab5A-GTP) based on correlation to a sigmoidal fit of the dose-response curve (Methods); the lower row shows the results for each candidate interactor for the other form of the protein (e.g., Rab5A-GDP). Hits are arranged alphabetically; color indicates the half-maximal response concentration, and white shows non-interactors. (G–I) Venn diagrams showing interactors identified by both MitoID and LiP–MS, compared between both forms of Rab5A and Rab2A (G), GTP-bound forms (H), and GDP-bound forms (I) of the two proteins.

Furthermore, we demonstrated that protein interaction measurements can be performed on crude membrane suspensions using LiP in combination with label-free quantitative MS. Importantly, we showed that protein sequence coverage of membrane proteins is significantly improved when utilizing crude membrane suspensions compared to standard cellular extracts, from which membrane proteins are largely removed. Our approach enabled studying PPIs that are difficult to resolve by other structural proteomics methods such as cryo-EM or X-ray crystallography. As such, our method can be used to better understand the participation of unstructured and flexible regions in PPIs, thus gaining deeper insights, for example, into the regulation and mechanism of action of ACs or other proteins with such domains. We were able to detect known interaction interfaces between AC8 and CaM, including their relative binding parameters in situ, and our system-wide analysis identified other putative CaM-binding proteins that contain canonical CaM-binding motifs. These novel putative interactors should be characterized in future studies. Our proof-of-principle study demonstrates that LiP, in combination with label-free quantitative MS, will be very valuable for the analysis of IMPs, as these targets are typically difficult to purify and immunoprecipitate.

Using the LiP–MS structural readout, we have probed the interactome of monomeric and fibrillar forms of aSyn, which undergoes protein misfolding and aggregation events and localizes in brain deposits, called LBs, in individuals with the disease. Previous studies have suggested that the interactome of aSyn can be conformation-dependent; however, conformation-specific interactions were demonstrated for only a small subset of proteins (Betzer et al, 2015; Lassen et al, 2016; Leitão et al, 2021; van Diggelen et al, 2020). Our work extends these prior studies to a proteome-wide scale, shows that aSyn monomers and fibrils likely interact with different sets of cellular proteins, and provides the rich resource of an extensive putative interactome of these aSyn conformations in situ.

In the course of our study, we identified a number of previously described interactors of aSyn, some of which are known to specifically interact with the monomeric form (CALM1) or fibrillar form (e.g., HSP90AA1, HSP90AB1, HSPA1A, HSPA8, and RAB3A) of the protein. However, other known interactors of aSyn (e.g., PINK1, PARK2, and LRRK2) were not detectable in our analysis, primarily due to their low expression levels in neurons, which is in accordance with previous literature (Lee et al, 2020). This reflects a general caveat of our approach, which relies on MS detection and sufficient sequence coverage of proteins in order to be able to investigate them as potential interactors.

We observed many more structurally altered proteins upon the addition of aSyn fibrils than upon the addition of monomer, consistent with the known lower binding preference of the aSyn monomer (Leitão et al, 2021; Betzer et al, 2015; van Diggelen et al, 2020). Notably, we have identified novel aSyn monomer-specific candidate interactor proteins involved in RNA binding and protein binding. We further identified putative interactors localized in extracellular organelles, cell junctions, vesicles, as well as mitochondria. In contrast, we detected a set of fibril-specific putative interactors, including ATP- and GTP-binding proteins, proteins with catalytic activity, and cytoskeletal protein binding. These proteins localize in extracellular vesicles, vesicles, or microtubules, some of which were previously unknown interactors and would

require follow-up to confirm direct aSyn conformation-specific interaction. Interestingly, a subgroup of our potential aSyn fibril interactors are components of LBs or implicated in their formation (Xia et al, 2008; Petyuk et al, 2021; Mahul-Mellier et al, 2020). Although LB formation is a complex process, it is possible that this set of proteins interacts with fibrillar aSyn also in LBs in vivo.

The network-based propagation and clustering analysis demonstrate disease-relevant links to common or rare variants of PD, such as involvement of protein polyubiquitination, ATP metabolism, RNA splicing/mRNA processing, and receptor-mediated endocytosis. One interesting aspect that emerged from the analysis of PD-associated genes was that fibril specificity was observed for intracellular energy metabolism and protein polyubiquitination. In this regard, defective mitochondrial functions that lead to increased oxidative stress have been demonstrated to play a central role in PD pathogenesis (Hattori and Mizuno, 2015). In particular, deficiencies of mitochondrial respiratory chain complex I may lead to the degeneration of neurons in PD by reducing the synthesis of ATP. In our analysis, we revealed interactions of aSyn fibrils with three proteins (NDUFB5, NDUFS1, and NDUFV2) that are part of the mitochondrial respiratory chain complex I. Furthermore, two proteins (ATP5PB and UQCRFS1) that showed structural alterations are part of the inner mitochondrial membrane protein complex. It is suggestive that, contrary to physiological, monomeric aSyn, pathogenic aSyn fibrils preferentially interact with mitochondria and cause mitochondrial respiration defects, as proposed by previous studies (Wang et al, 2019).

Growing evidence strongly implicates a direct role of the ubiquitin-proteasome system (UPS) in the pathogenesis of PD (Lim and Tan, 2007). The UPS is a type of intracellular protein degradation machinery, and its disruption in the presence of aSyn fibrils could lead to dysfunction of associated protein quality control mechanisms. We identified associations of three structurally altered proteins (CAND1, RCHY1, and UBE2O) with a common variant of the *PRKN* gene linked to PD. This gene encodes for the Parkin protein that functions as an E3 ubiquitin ligase. Interestingly, aggregated aSyn has been shown to selectively interact with the 19S cap and concomitantly inhibit the function of the 26S proteasome (Snyder et al, 2003).

In regard to endocytotic processes, we observed associations with genes involved in synaptic vesicle recycling and clathrin-dependent endocytosis. Through our analysis, we identified structurally altered proteins, specifically SYNJ1 and PACS1, that were found in the module of genes associated with common variants linked to PD. Previous studies provide evidence that an excess of aSyn monomer impairs clathrin-mediated synaptic vesicle endocytosis, as indicated by a loss of synaptic vesicles (Medeiros et al, 2018).

In our study, we utilized recombinantly expressed aSyn and amyloid fibrils generated in vitro. While previous studies have employed in vitro-generated aSyn amyloid fibrils to study aSyn aggregation (Narkiewicz et al, 2014; Viennet et al, 2018; Afitska et al, 2020), it is important to note that the structures of amyloid fibrils produced in the test tube may differ from those found in vivo fibrils in the brains of individuals with PD. Recent studies have generated brain-derived aSyn fibrils by amplification from brain extracts (Strohäker et al, 2019). These could be used in our approach to studying aSyn interactors in PD, but also in other synucleinopathies, such as DLB or MSA. It should be noted that

aSyn fibrils from different synucleinopathies are thought to adopt distinct conformations, known as amyloid strains. Our approach could thus enable the analysis of strain-specific interactomes for different strains of aSyn fibrils linked to various clinical phenotypes. Similarly, although we chose to use lysates of human SNCA-KO iPSC-derived cortical neurons in our experiments in order to reduce the potential background from endogenous aSyn, our approach could be applied to extracts of any other cellular or organismal model of PD.

A key advantage of our approach, as we have shown in this study, is that PPIs can be detected by adding a purified protein, in different conformations if desired, directly into complex extracts without the need for prior labeling. However, a few limitations should be noted. Cell lysis can lead to artifacts due to disruption of subcellular organization. Further, the method relies on the purity of proteins introduced into cellular extracts as contaminants could cause structural changes in the extract through direct binding or other indirect effects. In addition, although we have demonstrated that our approach identifies PPI interfaces directly, it may also detect conformational changes in other parts of the protein that occur due to protein binding or indirect effects triggered by binding to the target protein. Therefore, putative interactors must be confirmed using orthogonal methods.

Collectively, our work demonstrates that our LiP–MS approach successfully identifies PPIs in situ and detects interaction interfaces of different classes of proteins, including antibodies, membrane proteins, structured proteins, proteins with unstructured, intrinsically disordered regions, and disease-associated, amyloidogenic proteins, which remain difficult to study in classical interactomics experiments. LiP–MS further allows for the profiling of differential interactomes of different structural states of proteins in situ. As we demonstrate for aSyn, our method can be applied generally to study the interactomes of disease-relevant proteins that undergo structural alterations, and could thus help identify novel targets in drug discovery.

# Methods

### Reagents and tools table

| Reagent/resource | Reference or source | Catalog number |
|---|---|---|
| **Experimental models** | | |
| HEK293T cells (*H. sapiens*) | Aleš Holfeld | N/A |
| HEK293S GnTI- cell (*H. sapiens*) | Dina Schuster | CVCL_A785 |
| SFC856-03-04 human induced pluripotent stem cells (iPSCs) (*H. sapiens*) | Haenseler et al, 2017b | N/A |
| SFC856-03-04 SNCA-/- 12E knockout iPSCs (*H. sapiens*) | Haenseler et al, 2017b | N/A |
| **Antibodies** | | |
| Motavizumab | Sesterhenn et al, 2020 | N/A |
| D25 | Sesterhenn et al, 2020 | N/A |
| 5C4 | Sesterhenn et al, 2020 | N/A |

| Reagent/resource | Reference or source | Catalog number |
|---|---|---|
| Palivizumab | Sesterhenn et al, 2020 | N/A |
| 101 F | Sesterhenn et al, 2020 | N/A |
| InVivoMAb human IgG1 isotype control | Bio X Cell | #BE0297 |
| Rabbit anti-Alpha-synuclein antibody (MJFR1) | Abcam | 138501 |
| **Recombinant proteins** | | |
| Respiratory syncytial virus fusion (RSVF) glycoprotein | Sesterhenn et al, 2020 | N/A |
| Calmodulin | Merck | P1431 |
| Ras-related protein Rab5A (GTP- and GDP-bound form) | Alison K Gillingham | N/A |
| Ras-related protein Rab2A (GTP- and GDP-bound form) | Alison K Gillingham | N/A |
| Alpha-synuclein monomer | Dhiman Ghosh | N/A |
| Alpha-synuclein amyloid fibrils | Patterson et al, 2019 | N/A |
| **Chemicals, enzymes and other reagents** | | |
| Lysyl endopeptidase (Lys-C) | IGZ Instruments | 129-02541 |
| Trypsin | Promega | V5113 |
| TCEP (tris(2-carboxyethyl) phosphine hydrochloride) | Pierce | 20490 |
| Iodoacetamide | Sigma-Aldrich | I1149 |
| Ammonium bicarbonate | Sigma-Aldrich | 9830 |
| Formic acid (FA) | Sigma-Aldrich/Merck | 64-18-6 |
| HEPES (4-(2-hydroxyethyl) piperazine-1-ethanesulfonic acid, *N*-(2-Hydroxyethyl)piperazine-*N* ¢-(2-ethanesulfonic acid) | Sigma-Aldrich | H4034 |
| Sodium deoxycholate | Sigma-Aldrich | D6750 |
| Proteinase K (PK) from engyodontium album | Sigma-Aldrich | P2308 |
| HRM calibration kit | Biognosys AG | Ki-3002-2 |
| Magnesium chloride hexahydrate | Fluka | 63072 |
| Acetonitrile (ACN), LC-MS grade | ROTISOLV | AE70.1 |
| Potassium chloride | Merck | K41042236-032 |
| Thioflavin T (ThT) | Sigma-Aldrich | T3516 |
| Roche cOmplete EDTA free inhibitor cocktail | Roche | 11873580001 |
| DNAse I | Merck | 10104159001 |
| **Software** | | |
| Spectronaut 15 | https://biognosys.com/ | N/A |
| SpectroMine 4 | https://biognosys.com/ | N/A |
| R (4.2.2) | https://www.r-project.org/ | N/A |
| Rstudio (2022.07.2) | https://posit.co/download/rstudio-desktop/ | N/A |

| Reagent/resource | Reference or source | Catalog number |
|---|---|---|
| STRING (version 12.0) | https://string-db.org/ | N/A |
| AlphaFold2 | Varadi et al, 2022; Jumper et al, 2021 | N/A |
| UCSF ChimeraX (1.3rc202111292147) | https://www.rbvi.ucsf.edu/chimerax/ | N/A |
| **Other** | | |
| BCA protein assay | Pierce | 23228 |
| NativePAGE™ Sample Prep Kit | Invitrogen | BN2008 |
| NativePAGE™ 4 to 16%, Bis-Tris, 1.0 mm, Mini Protein Gels, 10-well | Invitrogen | BN1002BOX |
| NuPAGE™ MES SDS Buffer Kit (for Bis-Tris Gels) | Invitrogen | NP0060 |
| NuPAGE™ 4 to 12%, Bis-Tris, 1.0 mm, Mini Protein Gel, 10-well | Invitrogen | NP0321BOX |
| C18 MACROspin plate | The Nest Group | SMM SS18V |
| CLARIOstar Plus plate reader | BMG Labtech | N/A |
| ThermoMixer | Eppendorf | 460-0223 |
| Column oven | Sonation lab solutions | PRSO-V2 |
| Orbitrap Eclipse™ Tribrid™ Mass Spectrometer | Thermo Scientific | FSN04-10000 |
| Orbitrap Fusion™ Lumos™ Tribrid™ Mass Spectrometer | Thermo Scientific | FETD2-10001 |
| Orbitrap Exploris™ 480 Mass Spectrometer | Thermo Scientific | BRE725533 |
| EASY-nLC™ 1200 | Thermo Scientific | LC140 |
| Waters nanoACQUITY | Waters Corporation | 176016000 |
| HT7700 TEM | Hitachi | TEM HT7700 |

## Preparation of experimental models and subject details

### HEK293T cells

HEK293T cells were cultured in Dulbecco's Modified Eagle's Medium (DMEM) (Thermo Fisher) supplemented with 10% fetal bovine serum (FBS) and 1% penicillin/streptomycin. The HEK293T cells were passaged prior to confluency by detachment with 0.25% trypsin, followed by two consecutive washing steps in LiP buffer (100 mM HEPES pH 7.5, 150 mM KCl, 1 mM MgCl$_2$). To store the pellets, the HEK293T cells were transferred to 1.5-mL Eppendorf tubes and centrifuged at $1000 \times g$ at 4 °C for 5 min. LiP buffer was removed and the pellets were snap-frozen and stored at −80 °C until further use.

### HEK293S GnTI- membranes with AC8 overexpression

The full-length DNA construct of bovine AC8 was cloned into a tetracycline-inducible pACAMV-based vector with a C-terminal HRV 3C cleavage site and YFP-Twin-Strep fusion tag. The plasmid was transfected into HEK293S GnTI- cells and a stable clone expressing AC8 was selected for further protein expression.

### CRISPR/Cas9-mediated knockout of SNCA in human induced pluripotent stem cells (iPSCs)

The previously published human iPSC line SFC856-03-04 was used for gene editing (Haenseler et al, 2017b). The iPSC line was derived from a healthy donor (78y female) and is karyotypically normal as assessed by SNP analysis. SNCA has 6 exons, exon 2 is the first coding exon, and exons 3 and 5 can be alternatively spliced out. Exon 2 was chosen as the target for the introduction of a 49 bp deletion (downstream from the translation start site) that would lead to a frameshift and subsequent premature stop codon. The dual guide RNA sequences and double-strand cut strategy is illustrated in Appendix Fig. S3A, and utilized Alt-R CRISPR-Cas9 crRNA/trRNA/Hi-Fi Cas9 ribonucleotide-protein complex (IDT) and Neon electroporation (Thermo Fisher) to deliver the complex to the iPSC. Clones were screened by PCR and those clones harboring the deletion were sequenced across the repair to confirm out-of-frame repair (Appendix Fig. S3B). Clone SFC856-03-04 SNCA-/- 12E was used for the experiments presented here, it was confirmed to still retain the same karyotype as the parent iPSC line by Illumina Ominexpress24 SNP array and pseudokaryogram visualization using Karyostudio (Appendix Fig. S3C). The knockout of SNCA at the protein level was confirmed by differentiation to macrophages (Haenseler et al, 2017b), which express readily measurable levels of alpha-synuclein by flow cytometry (Fig. S3d); antibody MJFR1 (Abcam 138501) was used to detect aSyn, alongside a matched isotype control.

### iPSC culture and differentiation to cortical neurons

The healthy control SFC856-03-04 and the edited SFC856-03-04 SNCA-/- 12E knockout iPSC lines were used in the study presented here. iPSCs were cultured in Essential8 medium (Thermo Fisher) on Geltrex-coated tissue culture plates and passaged as small cell clusters using 0.5 mM EDTA (Thermo Fisher). iPSCs were differentiated into cortical neural progenitor cells (NPCs) with a dual SMAD inhibition protocol (Shi et al, 2012), with minor modifications as described previously (Haenseler et al, 2017a). NPCs were frozen on differentiation day 28 or directly used for further differentiation and experiments. The final plating of cells was on differentiation day 36. To remove proliferating progenitors and astrocytic cells cultures were treated with 2 µM AraC (Sigma-Aldrich) for 3 days. Experiments were performed on differentiation day 56.

### Respiratory syncytial virus fusion (RSVF) glycoprotein

The construct encoding the stabilized prefusion RSVF glycoprotein, known as DS2 (Joyce et al, 2016), corresponds to the sc9-10 DS-Cav1 A149C Y458C S46G E92D S215P K465Q variant. The codon-optimized sequence for mammalian cells was expressed and cloned into the pHCMV-1 vector flanked with two C-terminal Strep-Tag II and one 8x His tag. Expression and purification were performed as described previously (Sesterhenn et al, 2020). The postfusion RSVF glycoprotein was expressed, purified, and provided by Fabian Sesterhenn.

### RSVF site-specific antibodies

All site-specific monoclonal antibodies against RSVF used in this study were expressed, purified, and provided by Fabian Sesterhenn. Prior to LiP experiments, all antibodies were diluted in PBS buffer,

pH 7.0 to a concentration of 1 µg/µL. For LiP titration experiments, a total amount of 0.5, 1.0, 1.5, 2.0, 3.0, and 4.0 µg of motavizumab was spiked into each sample containing 100 µg of total proteins.

### Human IgG1 kappa antibody

The InVivoMAB human IgG1 isotype control antibody (BioXCell) purified from human myeloma serum was diluted in PBS buffer, pH 7.0 to a concentration of 1 µg/µL and used for subsequent LiP experiments with purified RSVF.

### Calmodulin (CaM)

Lyophilized CaM from bovine testes (Sigma-Aldrich) was solubilized in LiP buffer (100 mM HEPES pH 7.4, 150 mM KCl, 1 mM $MgCl_2$). For the CaM titration experiment, CaM was spiked into crude membranes at an amount of 0.01, 0.1, 0.5, 1, 2, and 3 µg per 100 µg of total crude membrane proteins.

### Alpha-synuclein (aSyn) monomer

N-terminally acetylated, human wild-type aSyn was expressed in *E. coli* cells (strain BL21 Star, DE3) transfected with the pRK172 plasmid with the yeast *N*-acetyltransferase complex B (NatB) as described previously (Kumari et al, 2021). Protein purification was performed as described elsewhere (Campioni et al, 2014). Lyophilized aSyn was resuspended in PBS buffer, pH 7.4. Subsequently, spun down with ultracentrifugation at 100,000 × *g* for 30 min before sample preparation to separate oligomeric species from monomeric aSyn. For LiP experiments, monomeric aSyn was spiked into iPSC SNCA-KO cell extracts at an amount of 0.05, 0.1, 0.5, 1.0, 2.0, and 4.0 µg per 50 µg of total proteins.

### aSyn amyloid fibrils

For the formation of mature aSyn amyloid fibrils, Eppendorf LoBind microcentrifuge tubes (1.5 mL) containing 0.75 mL of 5 mg/mL monomeric aSyn in PBS buffer, pH 7.4, 150 mM NaCl were incubated at 37 °C on a thermomixer under agitation at 800 rpm for 1–2 weeks. Amyloid fibrils were sonicated to produce shorter fibrillar structures as described elsewhere (Patterson et al, 2019). To remove low-molecular species, amyloid fibrils were pelleted by ultracentrifugation at 100,000 × *g* for 30 min and diluted in fresh PBS buffer, pH 7.4, to a concentration of 1 µg/µL. Subsequently, 0.05, 0.1, 0.5, 1.0, 2.0, and 4.0 µg of amyloid fibrils were added to each sample containing 50 µg of total proteins of the SNCA-KO iPSC extract.

### Rab2A and Rab5A in GTP- and GDP-bound forms

Samples of purified protein provided by the laboratory of Prof. Sean Munro (MRC Cambridge) were mixed with PBS buffer pH 7.0, resulting in a concentration of 1 µg/µL. To conduct LiP titration experiments in HEK293T cellular extracts, increasing amounts of Rab2A or Rab5A in either the GTP- or GDP-bound state (0.25, 0.5, 1.0, 1.5, and 2.0 µg) was added to each sample containing a total of 100 µg of proteins.

### Ethics and consent

Derivation of the human iPSC line SFC856-03-04 (used as the starting point for SNCA-knockout) used in this study is described elsewhere (Haenseler et al, 2017b). The iPSC line was derived from dermal fibroblasts from healthy donors through the Oxford Parkinson's Disease Centre: participants recruited to this study had signed informed consent, which included the derivation of human iPSC lines from skin biopsies (Ethics Committee that specifically approved this part of the study: for control donors, National Health Service, Health Research Authority, NRES Committee South Central, Berkshire, UK, REC 10/H0505/71, and for SNCA patient REC 07/H0720/161).

### Thioflavin T (ThT) fluorescence assay

The ThT binding assay was performed to confirm the presence of aSyn amyloid fibrils using a 40 µM ThT solution. Aliquots of aSyn monomer and amyloid fibrils were added to the ThT solution in three replicates, and fluorescence emission was measured at 25 °C on a CLARIOstar Plus plate reader (BMG Labtech) with an excitation wavelength of 440 nm. Fluorescence emission was recorded at a wavelength of 484 nm.

### Transmission electron microscopy (TEM)

Samples of amyloid fibrils were examined by TEM with negative staining. A droplet of the sample was placed on carbon film-coated copper grids, dried, and negatively stained with a droplet of 1% (w/v) uranyl acetate. The TEM images of the amyloid fibrils were imaged using a Hitachi HT7700.

### Blue native polyacrylamide gel electrophoresis (BN-PAGE)

BN-PAGE was performed using NativePAGE™ Sample Prep Kit and precasted NativePAGE™ 4 to 16%, Bis-Tris gels (1.0 mm, Mini Protein Gel, 10-well). Samples containing 1, 2, and 3 µg of monomeric aSyn were diluted with NativePAGE™ 4X Sample Buffer. Samples and the NativeMark™ Unstained Protein Standard were loaded into wells filled with 1X NativePAGE™ Dark Blue Cathode buffer, containing Coomassie G-250. Gels were run at 150 V constant in NativePAGE™ Dark Blue Cathode buffer at the Cathode and NativePAGE™ Anode buffer at the Anode for 30 min. NativePAGE™ Dark Blue Cathode buffer was exchanged with NativePAGE™ Light Blue Cathode buffer. The gel was run until completion at 150 V constant. Gels were fixed in a fixed solution (40% methanol, 10% acetic acid) and microwaved for 45 s, followed by shaking on an orbital shaker for 15 min. The gels were then destained in destaining solution (8% acetic acid) and microwaved for 45 s, followed by incubation on the orbital shaker for 15 min. This procedure was repeated multiple times until the gel was completely destained.

### SDS-PAGE

SDS-PAGE was performed using precasted 4–12% NuPAGE™ Bis-Tris gels in NuPAGE™ MES SDS Running Buffer. Laemmli buffer (5x) was added to the samples containing 1, 2, and 3 µg monomeric aSyn. As a marker, we used the PageRuler Plus Prestained protein ladder. The gel was run at 80 V constant for 15 min, followed by 150 V constant until completion. The gels were stained using PageBlue™ Protein Staining Solution, and destaining was achieved by shaking on an orbital shaker in double-deionized water.

## Preparation of cell extracts for MS analysis

### HEK293T and iPSC-derived cortical neurons

All steps throughout sample preparation were performed on ice. Cell pellets were resuspended in 400 μL LiP buffer (100 mM HEPES pH 7.4, 150 mM KCl, 1 mM MgCl$_2$) and lysed using a pellet pestle (Argos Technologies) in ten cycles of 10 s of homogenization and 1-min pause at 4 °C. The lysate was cleared by centrifugation (1000 × g at 4 °C) for 15 min. The supernatant was transferred to a new Eppendorf tube, and the remaining pellet was further resuspended in 200 μL LiP buffer. The lysis step was repeated as described, and supernatants were combined. The total lysate protein concentration was determined with a Pierce BCA Protein Assay Kit (cat #23225) according to the manufacturer's instructions.

### HEK293S GnTI- overexpressing bovine AC8

AC8 overexpressing HEK293S GnTI- cells were lysed in 100 mM HEPES-KOH (pH 7.4), 150 mM KCl, and 1 mM MgCl$_2$ using a dounce homogenizer. The lysate was centrifuged at 1000 × g to remove cell debris. The protein concentration of the lysate was determined with a Pierce BCA Protein Assay kit (cat #23225) and diluted to 2 μg/μL with the addition of final concentrations of 1 mM MnCl$_2$ and 1 mM CaCl$_2$. Four LiP samples and four trypsin control samples were produced, as described below in the AC8-CaM method section.

## Preparation of crude membranes for MS analysis

All steps throughout crude membrane preparation were performed on ice. Pellets of 2 L HEK293S GnTI- cells overexpressing bovine AC8 were resuspended in 50 mL LiP buffer (100 mM HEPES pH 7.4, 150 mM KCl, 1 mM MgCl$_2$) supplemented with one tablet of Roche cOmplete EDTA free inhibitor cocktail and 0.01 mg/mL DNAse I. Cells were lysed using a dounce homogenizer with 20 strokes and centrifuged at 1000 × g at 4 °C for 10 min. The supernatant was split, transferred to two ultracentrifuge tubes containing 25 mL LiP buffer containing protease inhibitors and DNAse I, and spun down (Ti45 rotor, 35,000 rpm at 4 °C) for 40 min. The remaining (membrane-enriched) pellets from the second centrifugation step were combined, resuspended in a total of 15 mL LiP buffer, and further homogenized using a dounce homogenizer with 20 strokes. The total protein concentration of crude membranes was determined with a Pierce BCA Protein Assay Kit (cat #23225) according to the manufacturer's instructions. Crude membranes were stored at –80 °C prior to further use.

## Limited proteolysis in native conditions

Purified RSVF proteins were incubated with site-specific and human isotype control antibodies at a molar ratio of 1:1 (protein:antibody) for 10 min at 25 °C and subjected to limited proteolysis. Proteinase K (PK) from *Tritirachium album* (Sigma-Aldrich) was added simultaneously to all four independent replicates of protein samples per condition (n = 4 for all experiments) with the aid of a multichannel pipette, at an enzyme-to-substrate ratio of 1:100 (w/w) and incubated at 25 °C for 5 min. Proteolytic reactions were stopped by heating samples for 5 min at 99 °C in a heat block. Subsequently, samples were transferred to

Eppendorf tubes containing an equal volume of 10% sodium deoxycholate (Sigma-Aldrich).

For validation experiments with postRSVF and motavizumab in HEK293T cellular extracts, three independent replicate samples (n = 3) per condition containing 100 μg of the HEK293T cellular extract, supplemented with 1 μg of postfusion RSVF, were exposed to a 5-dosage concentration series of motavizumab (0.5, 1.0, 1.5, 2.0, 3.0, and 4.0 μg). A control sample without antibody spiking was included for reference. Upon 10-min incubation at 25 °C, the samples were then subjected to limited proteolysis as described above. Additionally, untreated control and one condition of treated samples in this experiment were subjected to trypsin digestion only to control for potential protein abundance changes.

Crude membranes were aliquoted in equivalent volumes for each of four independent replicates (n = 4) containing 100 μg of proteins and incubated with calmodulin at given concentrations. Similarly, aSyn in its monomeric and fibrillar, aggregated state was added at given concentrations to each of four independent replicates (n = 4) for each condition containing 50 μg of iPSC extract. A reference sample without the addition of CaM was incorporated for comparison. Upon 10-min incubation at 25 °C, the samples were then subjected to limited proteolysis as described above. Additionally, untreated and one condition of treated samples in each experiment were subjected to trypsin digestion only to control protein abundance changes.

For experiments with Rab GTPases, Rab2A or Rab5A in either the GTP- or GDP-bound state was added at multiple concentrations to each of four independent replicates (n = 4), containing a total of 100 μg of proteins. A control sample where no protein was spiked in was incorporated for reference. Upon 10-min incubation at 25 °C, the samples were then subjected to limited proteolysis as described above.

## Trypsin digestion in denaturing conditions

Samples from all experiments were reduced with 5 mM tris(2-carboxyethyl)phosphine hydrochloride for 45 min at 37 °C. Alkylation was carried out in 40 mM iodoacetamide, followed by incubation at RT in the dark for 30 min. Thereafter, samples were diluted in four volumes of 100 mM ammonium bicarbonate and digested with lysyl endopeptidase and trypsin (both at an enzyme-to-substrate ratio of 1:100) at 37 °C for 16 h. Digests were acidified by the addition of formic acid to a final concentration of 2% and sodium deoxycholate precipitate was removed by filtration using a centrifugation filter at 1000 × g for 5 min. Peptides were desalted using a 96-well C18 MACROspin plate with 10–100 μg capacity according to the manufacturer's instructions. After drying, samples were resuspended in 3% acetonitrile (ACN) and 0.1% formic acid. The iRT kit (Biognosys AG, Schlieren, Switzerland) was added to all proteome samples as instructed by the manufacturer.

## Ultracentrifugation assay

The ultracentrifugation assay was employed to separate proteins interacting with amyloid fibrils of aSyn. SNCA-KO iPSC-derived cortical neuron extracts containing 50 μg of protein were incubated in three independent replicates (n = 3) with 2 μg of aSyn monomer and amyloid fibrils for 10 min at 25 °C and centrifuged at 100,000 × g for 30 min at 4 °C. The supernatant was transferred

into a new 1.5-mL Eppendorf tube, and the pellet was washed three times with 200 μL of LiP buffer (100 mM HEPES pH 7.4, 150 mM KCl, 1 mM MgCl$_2$). The pellet was resuspended in LiP buffer by vortexing for 5 min at RT. The protein concentration of the supernatant and the pellet was determined as described above. The samples were further processed with trypsin digestion in denaturing conditions as described above.

## AP–MS of aSyn monomer

A pellet of SH-SY5Y cells was lysed in IP buffer (LiP buffer + 1x complete protease inhibitor, 1xPhosSTOP). The lysate was split in aliquots before spiking in monomeric aSyn (5 μg). The lysate was incubated with aSyn for 1 h at room temperature (with end-to-end rotation). aSyn was immunoprecipitated with anti-αSyn antibody (Abcam MJFR1, 4 μg) for 2 h at 20 °C under constant end-to-end rotation using protein A conjugated magnetic beads. An isotype-specific Ig control was used as a negative control. Beads were collected on a DynaMag-2 magnetic rack and washed 6x with LiP buffer. The proteins were eluted in 60 μl 8 M urea for 30 min at 37 °C (1500 rpm, Eppendorf shaker). Samples were snap-frozen. Classical Tryptic digestion workflow was applied (see Trypsin digestion section).

## Mass spectrometry data acquisition

Peptide digests of purified RSVF and antibodies were analyzed in DIA mode on a Thermo Scientific Orbitrap Fusion Lumos Tribrid mass spectrometer (Thermo Fisher) equipped with a nanoelectrospray ion source and coupled to an Easy-nLC 1200 system (Thermo Fisher). Peptides were separated on a 25 cm × 0.75 μm i.d. analytical column (Thermo Fisher) packed with 1.9 μm C18 beads using a linear gradient from 5 to 30% buffer B (95% acetonitrile in 0.1% formic acid) over 30 min and a flow rate of 300 nL/min under ambient conditions. Full MS1 scans were acquired between 350 and 1400 $m/z$ at a resolution of 120,000. The automatic gain control (AGC) target of $8 \times 10^5$ and a maximum injection time of 100 ms were used. Forty-one variable-width windows (Appendix Table S1) were utilized to measure fragmented precursor ions. DIA-MS2 spectra were acquired at a resolution of 30,000 and an AGC target of $2 \times 10^5$, and an injection time of 54 ms. The normalized collision energy was set to 30.

For RSVF, aSyn, and ultracentrifugation assay proteome samples, peptide digests were analyzed in DDA and DIA modes on a Thermo Scientific Orbitrap Eclipse Tribrid mass spectrometer (Thermo Fisher) equipped with a nanoelectrospray ion source and coupled to an Easy-nLC 1200 system (Thermo Fisher). Peptides were loaded onto a 40 cm × 0.75 μm i.d. analytical column packed in-house with 1.9 μm C18 beads (Dr. Maisch Reprosil-Pur 120) and separated by a 120 min linear gradient at a flow rate of 300 nL/min with increasing buffer B (95% acetonitrile in 0.1% formic acid) from 3 to 30%. For DDA, a full MS1 scan was acquired over a mass range of 350–1400 $m/z$ at a resolution of 120,000 with an AGC target of 200% and an injection time of 100 ms. DDA-MS2 spectra were acquired at a resolution of 30,000 with an AGC target of 200% and an injection time of 54 ms. To maximize parallelization, a duty cycle time was 3 s. For DIA, a full MS1 scan was acquired between 350 and 1100 $m/z$ at a resolution of 120,000 with an AGC target of 200% and an injection time of 100 ms. Forty-one variable-width

windows (Appendix Table S1) were used to measure fragmented precursor ions. DIA-MS2 spectra were acquired at a resolution of 30,000 with an AGC target of 400% and an injection time of 54 ms. The normalized collision energy was set to 30.

For AC8-CaM membrane suspension samples, 1 μg peptide digests were loaded onto a 40 cm × 0.75 μm i.d. column packed in-house with 1.9 μm C18 beads (Dr. Maisch Reprosil-Pur 120) and separated by a 120 min linear gradient at a flow rate of 300 nL/min with increasing buffer B (95% acetonitrile in 0.1% formic acid) from 3 to 30%. All DIA and DDA runs were acquired on a Thermo Scientific Exploris 480 mass spectrometer (Thermo Fisher). For DDA, a full MS1 scan was acquired between 350 and 1100 $m/z$ at a resolution of 120,000 with an AGC target of 200% and an injection time of 100 ms. DDA-MS2 spectra were acquired at a resolution of 30,000 with an AGC target of 200% and an injection time of 54 ms. To maximize parallelization, a duty cycle time was 3 s. For DIA, a full MS1 scan was acquired between 350 and 1100 $m/z$ at a resolution of 120,000 with an AGC target of 100% and an injection time of 100 ms. Forty-one variable-width windows (Appendix Table S1) were used to measure fragmented precursor ions. DIA-MS2 spectra were acquired at a resolution of 30,000 and an AGC target of 2000%. The first mass was fixed at 200 $m/z$ and the normalized collision energy was set to 28. Proteome samples prepared from cell lysate with overexpressed AC8 were acquired on a Thermo Scientific Eclipse Tribrid mass spectrometer (Thermo Fisher) equipped with a nanoelectrospray source, coupled to an Easy-nLC 1200 system (Thermo Fisher), with the same parameters as used for the RSVF experiment.

Peptide digest of Rab GTPases were analyzed in DDA mode on a Thermo Scientific Orbitrap Fusion Lumos Tribrid mass spectrometer (Thermo Fisher) equipped with a nanoelectrospray ion source and coupled to a Waters nanoACQUITY system (Waters Corporation). Peptides were separated on a 40 cm × 0.75 μm i.d. column packed in-house with 1.9 μm C18 beads (Dr. Maisch Reprosil-Pur 120) and separated by a 120 min linear gradient at a flow rate of 300 nL/min with increasing buffer B (95% acetonitrile in 0.1% formic acid) from 3 to 35%. Full MS1 scans were acquired over a mass range of 350–1400 $m/z$ at a resolution of 120,000 with an AGC target of $4 \times 10^5$ and an injection time of 100 ms. DDA-MS2 spectra were acquired at a resolution of 30,000 with an AGC target of $1 \times 10^5$ and an injection time of 54 ms. To maximize parallelization, a duty cycle time was 3 s. The first mass was fixed at 110 $m/z$ and the normalized collision energy was set to 28.

All MS data were acquired following a stringent randomization procedure. This approach ensured that the order of sample analysis was completely randomized, thereby eliminating any potential bias or systematic errors that could arise from a fixed sequence of analysis. No blinding was used in any experiments.

## Mass spectrometry data analysis

Prior to DIA spectra processing, all spectral libraries were generated using the library generation functionality of Spectronaut 15 (Biognosys AG, Schlieren, Switzerland) (Bruderer et al, 2015) using the default settings with minor adaptations. In brief, the DIA and/or DDA files were searched against the human UniProt FASTA database (updated 2020-03-20), the MaxQuant contaminants fasta database (245 entries), and the Biognosys' iRT peptides

FASTA database. Raw files for RSVF and AC8-CaM datasets were additionally searched against the prefusion or postfusion RSVF FASTA databases and the AC8 (including tags) FASTA database (uploaded to the public repository), respectively. For LiP–MS datasets, digestion enzyme specificity was set to Trypsin/P and semi-specific. For trypsin-only treated controls, a digestion enzyme was Trypsin/P with specific cleavage rules. The minimum allowed peptide length was set to 5 amino acids with a maximum of two missed cleavages per peptide. Carbamidomethylation of cysteine was considered a fixed modification, and acetylation (protein N-terminus) and oxidation of methionine as variable modifications. DIA spectra were further processed with Spectronaut using the default settings with a few modifications. In short, dynamic retention time extraction was applied with a correction factor of 1. The identification of peptides and proteins was controlled by the false discovery rate (FDR) of 1%. The machine learning algorithm and $Q$ value calculations were run across the entire experiment. Peptide quantification was carried out on the modified peptide sequence level using precursor ions. Protein quantification included only proteotypic peptides, and global median normalization was applied. For datasets involving Rab GTPases, DDA files were processed using SpectroMine 4 using the default settings. Imputation and normalization were disabled.

## Interpretation of antibody–target protein interactions with purified proteins

Peptide reports provided by Spectronaut were processed using an in-house R script in R Statistical Software (version 4.2.2; R Core Team 2021). Raw abundances of proteotypic RSVF peptides were normalized using variance stabilizing normalization with the *vsn* package (Huber et al, 2002) (version 3.64.0). The normalized $\log_2$-transformed peptide abundances from treated samples with site-specific antibodies against RSVF were compared to control, i.e., anti-Human IgG1 kappa antibody with RSVF. The $\log_2$ FC and the statistical significance (represented by $p$ values adjusted by multiple testing using the Benjamin–Hochberg method) were computed using an empirical Bayes moderated $t$-test provided by the *limma* package (Ritchie et al, 2015) (version 3.52.4). Peptides with at least three measured peptide abundances for each condition, fulfilling the defined criteria ($|\log_2$ FC$| > 1$, $q$ value $< 0.01$), were considered significant. The score for each peptide that shows significant changes upon the addition of antibody was computed by dividing $|\log_2$ FC$|$ of peptide intensity by the adjusted $p$ value. Significant peptides were mapped onto 3D structures of perfusion (PDB: 4JHW) (McLellan et al, 2013) and postfusion (PDB: 3RRR) (McLellan et al, 2011) RSVF.

## Dose-response analysis

Peptide reports generated in Spectronaut or SpectroMine were processed using an in-house R script in R Statistical Software (version 4.2.2; R Core Team 2021). In brief, proteins with at least two peptide precursors were considered. RSVF and AC8-CaM datasets included only proteotypic peptides. For aSyn and Rab GTPases datasets, both proteotypic and non-proteotypic peptides were covered; therefore, non-proteotypic peptides, which are reported, should be taken with caution. Raw peptide abundances were normalized using variance stabilizing normalization with the

*vsn* package (Huber et al, 2002) (version 3.64.0). We used an outlier detection method based on the interquartile range (IQR) to define boundaries outside of the first (Q1) and third (Q3) quartile for peptide abundances within each condition per peptide precursor. Peptide abundances that are more than 1.5 times the IQR below Q1 or more than 1.5 times above Q3 are considered outliers. This approach ensured that potential outliers are removed prior to dose-response analysis. Subsequently, we considered only peptide precursors that were measured in at least 3 replicates, covering at least five conditions. Filtered data, consisting of normalized $\log_2$-transformed peptide abundances, were scaled to a range between 0 and 1 and subjected to dose-response analysis using the *protti* package (Quast et al, 2022) (version 0.5.0) that utilizes the log-logistic model with four parameters (LL.4) from the *drc* package (Ritz et al, 2015) (version 3.0-1). Pearson's correlation coefficient $r$ was used to assess the strength of the sigmoidal trend of dose-response profiles. Only peptides that fulfilled the Benjamini–Hochberg-corrected $p$ values ($q$ values) $< 0.01$ obtained from an analysis of variance (ANOVA) and Pearson's correlation coefficients $r > 0.85$ were considered significant. Unscaled peptide abundances were used for statistical testing of differentially abundant peptides using an empirical Bayes moderated t-test as implemented in the *limma* package (Ritchie et al, 2015) (version 3.52.4). The resulting $p$ values were adjusted by multiple testing using the Benjamin–Hochberg method. The output of the statistical analysis was filtered using the following cutoffs: $q$ value $< 0.01$ and $|\log_2$ FC$| > 0.75$. Note that we used a cutoff of $|\log_2$ FC$| > 0.75$ for experiments in lysates (which were, in most cases, multi-dose experiments) but a cutoff of $|\log_2$ FC$| > 1$ in single-dose experiments with purified samples, because in the former case, meaningful changes are not only detected based on fold change but also on how well the data fit a sigmoidal profile; the analysis can thus tolerate a slightly less stringent FC threshold. In our dose-response analyses, the final list of peptides that fulfilled all the defined criteria represented differentially altered peptides. Every differentially altered peptide has an "EC50" value assigned, which represents the inferred quantity of a protein necessary to observe a half-maximum of the relative peptide intensity change between treated and untreated samples. Furthermore, the score for each differentially altered peptide was computed by dividing $|\log_2$ FC$|$ of peptide intensity by the adjusted $p$ value.

## Proteomic analysis of CRISPR/Cas9-mediated knockout of SNCA in iPSC-derived cortical neurons

To verify the knockout of SNCA, we analyzed the expression levels of aSyn in both healthy control ($n = 4$) and SNCA-KO ($n = 4$) iPSC lines using quantitative DIA–MS. MS2 quantification of aSyn peptides was performed with Spectronaut as described above. Extracted ion chromatographs for peptides of aSyn (Appendix Fig. S3E) were exported from Spectronaut 15.

## Mapping of known interactors

Systematic analysis of known interactors was conducted using the STRING database (https://string-db.org) of physically interacting proteins (Szklarczyk et al, 2021). Proteins with a score of $>150$ were considered known interactors. Fisher's exact test ($p$ value $< 0.01$) was used to determine whether known interactors are enriched

amongst our identified proteins relative to all identified known interactors.

## Identification of Lewy body (LB)-associated proteins

To evaluate whether structurally altered proteins upon spike-in of aSyn are associated with the formation of LBs, we utilized data from previous studies which report on LB-associated proteins in a neuronal aSyn fibril seeding model (Mahul-Mellier et al, 2020) or in postmortem patient brains (Petyuk et al, 2021; Xia et al, 2008) Fisher's exact test ($p$ value <0.01) was used to determine whether LB-associated proteins are enriched amongst our identified structurally altered proteins relative to all identified LB-associated proteins. The web-based functionality g:Orth was used for an orthology search to translate *M. musculus* genes into *H. sapiens* genes using g:Profiler (version e106_eg53_p16_65fcd97, database updated on 18/05/2022) (Raudvere et al, 2019).

## Analysis of calmodulin-binding motifs

Calmodulin (CaM)-binding motifs were assessed using an in-house R script in R Statistical Software (version 4.2.2; R Core Team 2021). Briefly, we concatenated information about known CaM-binding motifs from the Calmodulation database and Meta-analysis predictor website (http://cam.umassmed.edu), which enables the prediction of CaM-binding motifs in protein sequences (Mruk et al, 2014). We predicted the presence of CaM-binding motifs in structurally altered peptides and calculated the number of CaM-binding motifs per protein (without discrimination for transmembrane domains).

## Functional enrichment analysis

Functional enrichment analysis of proteins, based on gene ontology (GO) terms molecular function (MF), biological process (BF), and cellular component (CC), was performed using g:Profiler (version e106_eg53_p16_65fcd97, database updated on 18/05/2022) (Raudvere et al, 2019) with the FDR multiple testing correction method applying a significant threshold of 0.01 ($q$ value <0.01). We further utilized the *rrvgo* package (Supek et al, 2011) (version 1.8.0) to summarize the enriched terms by removing redundant GO terms. The resulting list size was set to 0.7 and the SimRel functional similarity measure for comparing two GO terms with each other was considered.

## Network propagation and clustering analysis

To assess associations between our identified proteins and PD-related traits in the aSyn experiment, we performed network-based expansion and clustering analysis as described previously (Barrio-Hernandez et al, 2021). In brief, to generate a list of starting genes for network expansion, we used all proteins from the LiP experiments and all genes linked to different PD-related traits. We defined a list of Parkinson-related disorders based on the EFO hierarchy, selecting all traits that have the term "Parkinson's disease (EFO:0002508)" as the ancestor and genes with associations based on common or rare variants, leading to the following list: "Parkinson disease, mitochondrial", "Young adult-onset Parkinsonism", "Parkinson's disease" and "Hereditary late-onset Parkinson's disease". To select genes associated with a given trait, we

used the evidence present in the OpenTargets platform (https://www.opentargets.org/). For common variants, we selected all genes with an L2G score (association of an SNP to a given gene from GWAS studies) bigger than 0.5. For rare variants, we used all genes linked to SNPs with a clinical output not considered "benign", according to ClinVar definitions (https://www.ncbi.nlm.nih.gov/clinvar/). The network expansion is performed trait-per-trait, following an approach described previously (Barrio-Hernandez et al, 2021). Briefly, we first mapped all the genes considered as starting signals (LiP experiments or genes associated with genetic evidence of association to a given disease) to a custom version of OpenTargets interactome (compilation of interactions from IntAct, Reactome, Signor, and STRING with score ≥0.75). We applied network propagation using the personalized PageRank algorithm included in the *igraph* package (Csárdi and Nepusz, 2006) (version 1.2.4.2). Those genes with a ranking score bigger than the Q3 (75% of the distribution) are selected for community detection using the walktrap algorithm from the *igraph* package (version 1.2.4.2) via random walks. To define significant communities, we compared the PageRank score, resulting from the network propagation inside and outside the community, using the Kolmogorov–Smirnov test with the Benjamini–Hochberg adjustment. We selected the communities with at least 1 starting hit, no less than ten nodes in total, and with an adjusted $p$ value smaller than 0.05. Significant communities were compared among traits by measuring the nodes' overlap using the Jaccard index. To calculate an enrichment based on GOBP annotation, Fisher's exact test ($p$ value <0.05) was used.

## Data analysis of enriched and depleted proteins

To identify proteins that are either enriched in the pellet with aSyn amyloid fibrils or depleted from the supernatant, $\log_2$-transformed protein abundances provided in the Spectronaut report were used to calculate $\log_2$ FC and $q$ values using an empirical Bayes moderated $t$-test as implemented in the *limma* package (Ritchie et al, 2015) (version 3.52.4). For the assessment of pellet-enriched proteins, pellet samples containing aSyn amyloid fibrils were compared to samples treated with aSyn monomer or untreated samples. For supernatant-depleted proteins, the supernatant recovered after ultracentrifugation was used. Significant proteins ($q$ value <0.01) that changed in abundance either in the pellet or supernatant samples with a fold-change of 1.5 were considered enriched or depleted. Fisher's exact test ($p$ value <0.01) was used to determine whether putative fibril-binding proteins obtained by ultracentrifugation are enriched amongst our identified structurally altered proteins relative to all identified proteins.

## Assessment of half-maximal response concentrations of proteins identified by LiP–MS and ultracentrifugation assay

Half-maximal response concentrations were derived from dose-response curves of peptides identified as significantly changing in the LiP–MS experiments. In each case, the peptide with the lowest half-maximal response concentration was used to represent the protein's half-maximal response concentration. These protein's half-maximal response concentrations were compared to the half-maximal response concentrations of proteins identified by both LiP–MS and ultracentrifugation. The distribution of half-maximal response concentrations was evaluated using the Wilcoxon test, and the obtained $p$ value was used to determine

whether there were statistically significant differences between the half-maximal response concentrations of the two groups.

## Identification and comparison of top hits in Rab-interactor screening

Identification of the 50 top hits for each nucleotide-bound form of each Rab protein was conducted as follows: The candidate interactor list was filtered for a half-maximal response concentration between 0 and 2 µg (i.e., the range over which spiked in the bait protein), considering the peptide with the lowest half-maximal response concentration for proteins where more than one peptide significantly changed. We then selected the top 50 proteins based on the correlation (Pearson's coefficient) of the dose-response curve to a sigmoidal fit. For each form of each Rab (i.e., Rab5A-GTP, Rab5A-GDP, Rab2A-GTP, and Rab2A-GDP), this hit list was then examined in the data for the other nucleotide-bound form, and each pairwise comparison was visualized using heatmaps.

## Prediction of AC8 structure

The 3D structure of bovine AC8 was predicted from its amino acid sequence using AlphaFold2 (Varadi et al, 2022; Jumper et al, 2021). The FASTA file containing the amino acid sequence of AC8 and its tags was submitted to the AlphaFold prediction algorithm. The predicted structure was then used to visualize differential peptides upon CaM treatment.

## 3D analysis of protein structural changes

Significantly altered peptides were mapped onto representative 3D protein structures obtained from the Protein Data Bank (Berman et al, 2000). In RSVF experiments, preRSVF (PDB: 4JHW) (McLellan et al, 2013) and/or postRSVF (PDB: 3RRR) (McLellan et al, 2011) structures were used to identify RSVF regions that changed due to antibody binding. In AC8-CaM experiments, the predicted structure (uploaded to the public repository) was used to detect CaM-binding sites of AC8. All structural alterations were visualized using the molecular visualization program UCSF ChimeraX (1.3rc202111292147) (Goddard et al, 2018; Pettersen et al, 2021).

## Data availability

The mass spectrometry proteomics data have been deposited to the ProteomeXchange Consortium via the PRIDE partner repository (Perez-Riverol et al, 2022). They are available at PXD039481, PXD039520, PXD039784, and PXD048849. The custom R scripts developed and used in this study are available via GitHub at https://gitfront.io/r/PicottiGroup/FeTezEanyUFM/LiP-MS-protein-protein-interactions-lip-data-structural-analysis-protein-protein-interactions/.

## Peer review information

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

## Acknowledgements

The authors thank the Correia laboratory (EPFL Lausanne, Switzerland) for providing samples of purified RSVF and site-specific antibodies for validation experiments. P.P. was supported by the European Research Council (grant agreement no. 866004), the EPIC-XS Consortium (grant agreement no. 823839), a Sinergia grant from the Swiss National Science Foundation (SNSF grant CRSII5_177195), grants from the Synapsis Foundation, Parkinson Schweiz, and the Empiris Foundation, and the EU Horizon 2020 program INFRAIA project EPIC-XS (Project 823839). V.M.K. was supported by a Swiss National Science Foundation grant (no. 184951). W.H. was supported by the Oxford-McGill-Zurich Partnership in Neuroscience and is currently supported by UZH URPP AdaBD. P.B. is supported by the Helmut Horten Stiftung and the ETH Zurich Foundation.

## Author contributions

**Aleš Holfeld**: Data curation; Software; Formal analysis; Validation; Investigation; Visualization; Methodology; Writing—original draft; Writing—review and editing. **Dina Schuster**: Formal analysis; Investigation; Visualization. **Fabian Sesterhenn**: Resources; Writing—original draft. **Alison K Gillingham**: Resources. **Patrick Stalder**: Resources; Investigation. **Walther Haenseler**: Resources. **Inigo Barrio-Hernandez**: Formal analysis. **Dhiman Ghosh**: Resources; Methodology. **Jane Vowles**: Resources. **Sally A Cowley**: Resources. **Luise Nagel**: Formal analysis. **Basavraj Khanppnavar**: Resources. **Tetiana Serdiuk**: Resources; Investigation. **Pedro Beltrao**: Formal analysis; Supervision. **Volodymyr M Korkhov**: Supervision; Investigation. **Sean Munro**: Resources; Supervision. **Roland Riek**: Resources; Supervision. **Natalie de Souza**: Supervision; Writing—original draft; Project administration; Writing—review and editing. **Paola Picotti**: Conceptualization; Supervision; Funding acquisition; Methodology; Project administration; Writing—review and editing.

## Disclosure and competing interests statement

PP is an inventor of a patent licensed by Biognosys AG that covers the LiP–MS method used in this manuscript and a member of the scientific advisory board of Biognosys AG. The remaining authors declare no competing interests. Pedro Beltrao is a member of the Advisory Editorial Board of Molecular Systems Biology. This has no bearing on the editorial consideration of this article for publication.

