## [Peer Review File · Molecular Systems Biology]

Systematic identification of structure-specific protein-protein interactions

Aleš Holfeld, Dina Schuster, Fabian Sesterhenn, Alison Gillingham, Patrick Stalder, Walther Haenseler, Inigo Barrio-Hernandez, Dhiman Ghosh, Jane Vowles, Sally Cowley, Luise Nagel, Basavraj Khanppnavar, Tetiana Serdiuk, Pedro Beltrao, Volodymyr Korkhov, Sean Munro, Roland Riek, Natalie de Souza, and Paola Picotti

Corresponding author(s): Paola Picotti (picotti@imsb.biol.ethz.ch)

Review Timeline:

Submission Date:	1st Feb 23
Editorial Decision:	21st Mar 23
Revision Received:	5th Feb 24
Editorial Decision:	12th Mar 24
Revision Received:	12th Apr 24
Accepted:	15th Apr 24

Editor: Maria Polychronidou

Transaction Report:

21st Mar 2023

Manuscript Number: MSB-2023-11552

Title: Systematic identification of structure-specific protein-protein interactions

Dear Paola,

Thank you again for submitting your work to Molecular Systems Biology. My apologies once again for the slow process. We have now heard back from the four reviewers who agreed to evaluate your study. As you will see below, the reviewers acknowledge that the addressed topic is interesting. They do however raise a series of concerns, which we would ask you to address in a revision.

I think that most of the reviewers' recommendations are rather clear and I therefore see no need to repeat any of the comments listed below. One important point is raised by reviewers #1 and #2, who mention that further analyses would be required to better support the broader applicability of LiP-MS to the identification of PPIs for different structural states. We would strongly encourage the inclusion of such data, as it would significantly enhance the impact of the study. If analyses along those lines are not feasible within the timeline of the major revision, we would recommend carefully editing the text to avoid potential overstatements.

All issues raised by the referees would need to be satisfactorily addressed. Please let me know in case you would like to discuss in further detail any of the issues raised, I would be happy to schedule a call.

On a more editorial level, we would ask you to address the following points:

- Please provide a .doc version of the manuscript text (including legends for the main figures) and individual production quality figure files for the main Figures (one file per figure).
- We have replaced Supplementary Information by the Expanded View (EV format). In this case, all additional figures and Tables can be included in a PDF called Appendix. Appendix figures and Tables should be labeled and called out as: "Appendix Figure S1, Appendix Figure S2... Appendix Table S1..." etc. Each legend should be below the corresponding Figure/Table in the Appendix. Please include a Table of Contents in the beginning of the Appendix. For detailed instructions regarding expanded view please refer to our Author Guidelines: .
- Supplementary Data 1-4 should be provided as Datasets EV1-EV4. Please provide one file per dataset. Each file should include a description of the EV Dataset in a separate tab in the .xls file.
- Please provide a "standfirst text" summarizing the study in one or two sentences (approximately 250 characters), three to four "bullet points" highlighting the main findings and a "synopsis image" (550px width and max 400px height, jpeg format) to highlight the paper on our homepage.
- All Materials and Methods need to be described in the main text. We would ask you to use 'Structured Methods', our new Materials and Methods format, which is mandatory for Methods and Articles with a strong methodological focus. According to this format, the Material and Methods section should include a Reagents and Tools Table (listing key reagents, experimental models, software and relevant equipment and including their sources and relevant identifiers) followed by a Methods and Protocols section in which we encourage the authors to describe their methods using a step-by-step protocol format with bullet points, to facilitate the adoption of the methodologies across labs. More information on how to adhere to this format as well as downloadable templates (.doc or .xls) for the Reagents and Tools Table can be found in our author guidelines: . An example of a Method paper with Structured Methods can be found here: .
- Please include a "Disclosure & Competing Interests Statement".
- Please include a "Data availability" section describing how the data, code etc. have been made available. This section needs to be formatted according to the example below:
The datasets and computer code produced in this study are available in the following databases:
 - Chip-Seq data: Gene Expression Omnibus GSE46748 (<https://www.ncbi.nlm.nih.gov/geo/query/acc.cgi?acc=GSE46748>)
 - Modeling computer scripts: GitHub (<https://github.com/SysBioChalmers/GECKO/releases/tag/v1.0>)
 - [data type]: [full name of the resource] [accession number/identifier] ([doi or URL or identifiers.org/DATABASE:ACCESSION])
- For data quantification: please specify the name of the statistical test used to generate error bars and P values, the number (n) of independent experiments (specify technical or biological replicates) underlying each data point and the test used to calculate p-values in each figure legend. The figure legends should contain a basic description of n, P and the test applied. Graphs must

include a description of the bars and the error bars (s.d., s.e.m.).

- When you resubmit your manuscript, please download our CHECKLIST (<https://bit.ly/EMBOPressAuthorChecklist>) and include the completed form in your submission.

Please note that the Author Checklist will be published alongside the paper as part of the transparent process (<https://www.embopress.org/page/journal/17444292/authorguide#transparentprocess>).

If you feel you can satisfactorily deal with these points and those listed by the referees, you may wish to submit a revised version of your manuscript. Please attach a covering letter giving details of the way in which you have handled each of the points raised by the referees. A revised manuscript will be once again subject to review and you probably understand that we can give you no guarantee at this stage that the eventual outcome will be favorable.

Kind regards,

Maria

Maria Polychronidou, PhD
Senior Editor
Molecular Systems Biology

We realize that it is difficult to revise to a specific deadline. In the interest of protecting the conceptual advance provided by the work, we recommend a revision within 3 months (19th Jun 2023). Please discuss the revision progress ahead of this time with the editor if you require more time to complete the revisions. Use the link below to submit your revision:

IMPORTANT: When you send your revision, we will require the following items:

1. the manuscript text in LaTeX, RTF or MS Word format
2. a letter with a detailed description of the changes made in response to the referees. Please specify clearly the exact places in the text (pages and paragraphs) where each change has been made in response to each specific comment given
3. three to four 'bullet points' highlighting the main findings of your study
4. a short 'blurb' text summarizing in two sentences the study (max. 250 characters)
5. a 'thumbnail image' (550px width and max 400px height, Illustrator, PowerPoint or jpeg format), which can be used as 'visual title' for the synopsis section of your paper.
6. Please include an author contributions statement after the Acknowledgements section (see <https://www.embopress.org/page/journal/17444292/authorguide>)
7. Please complete the CHECKLIST available at (<https://bit.ly/EMBOPressAuthorChecklist>).

Please note that the Author Checklist will be published alongside the paper as part of the transparent process (<https://www.embopress.org/page/journal/17444292/authorguide#transparentprocess>).

See also figure legend guidelines: <https://www.embopress.org/page/journal/17444292/authorguide#figureformat>

9. Please note that corresponding authors are required to supply an ORCID ID for their name upon submission of a revised manuscript (EMBO Press signed a joint statement to encourage ORCID adoption).

(<https://www.embopress.org/page/journal/17444292/authorguide#editorialprocess>)

Currently, our records indicate that there is no ORCID associated with your account.

Please click the link below to provide an ORCID:

Link Not Available

The system will prompt you to fill in your funding and payment information. This will allow Wiley to send you a quote for the article processing charge (APC) in case of acceptance. This quote takes into account any reduction or fee waivers that you may be eligible for. Authors do not need to pay any fees before their manuscript is accepted and transferred to the publisher.

EMBO Press participates in many Publish and Read agreements that allow authors to publish Open Access with reduced/no publication charges. Check your eligibility: <https://authorservices.wiley.com/author-resources/Journal-Authors/open-access/affiliation-policies-payments/index.html>

*** PLEASE NOTE *** As part of the EMBO Press transparent editorial process initiative (see our Editorial at <https://dx.doi.org/10.1038/msb.2010.72>), Molecular Systems Biology publishes online a Review Process File with each accepted manuscripts. This file will be published in conjunction with your paper and will include the anonymous referee reports, your point-by-point response and all pertinent correspondence relating to the manuscript. If you do NOT want this File to be published, please inform the editorial office at msb@embo.org within 14 days upon receipt of the present letter.

Reviewer #1:

In the manuscript titled "Systematic identification of structure-specific protein-protein interactions", the authors applied LiP-MS to identify potential interactors to a given target (bait) protein by probing changes in the digestibility of proteins upon the addition of the bait protein. This is similar to fishing interaction partners using recombinant bait proteins, albeit with the marked difference that in the case of LiP-MS the interaction does not have to survive subsequent biochemical enrichment. Also, in the case of LiP-MS there would be no background from unspecific binding of proteins to beads used during the biochemical enrichment. In consequence, LiP-MS should give much cleaner data. I wonder if the authors might be well advised to demonstrate this by help of a comparison, using some tagged recombinant protein in a classical pulldown and using LiP-MS.

As a benchmark, the authors demonstrated, using purified RSVF and its site-specific antibodies, that LiP-MS could detect interactions between RSVF and antibodies accurately. The structurally altered peptides are located at and near the expected binding sites of respective antibodies. Such interactions were also detected when spiking the RSVF in the background of HEK293 cell extract, where a dose-response experiment was necessary to distinguish true and false positive hits.

A second proof of principle analysis was conducted to show that LiP-MS could detect protein-protein interaction to a membrane protein. Here, LiP-MS was applied to detect a known interaction between calmodulin and a bovine membrane-integral protein CA8 which is over-expressed in the HEK239 cells. Structural changes were also identified for additional proteins from the HEK239 cell extract, a subset of them were predicted to contain calmodulin-binding motifs.

In the end, authors applied LiP-MS to detect, in the extract of SNCA-KO iPSC-derived neurons, the interacting proteins of alpha-synuclein (α -syn) monomer and α -syn amyloid fibrils, respectively. Although there are some overlaps, different sets of interacting proteins were detected interacting with α -syn monomer and amyloid fibrils. The results were further verified by

- 1) GO enrichment and network enrichment analysis of the detected α -syn interacting proteins.
- 2) comparing to the results of previously reported proteomics studies.
- 3) comparing to proteins that co-precipitation with the α -syn amyloid fibril by Ultracentrifugation.

The studies reported in the manuscript were appropriately conducted. It is worth publishing as a successful application of LiP-MS on identifying interacting proteins of different forms of α -syn. Otherwise, additional data must be provided to support several general statements made about the methodology.

1. The authors stated that LiP-MS can be applied to identify protein-protein interactions specific to protein structural states accurately. A single example was shown using a comparison between α -syn monomer and amyloid fibrils. The difference in structure between these two forms of the protein is significantly greater than the typical variation in structural states of most proteins and protein complexes. Therefore, it is necessary to conduct additional tests using bait proteins with varying degrees of structural differences to validate the general applicability of LiP-MS to identify PPIs for distinct structural states of proteins.

2. The authors promoted the use of detergent-free crude membrane prep to enable the detection of PPIs to membrane proteins. The interaction between calmodulin and CA8 has been identified using crude membrane prep. Would it be possible to identify this interaction using the native lysate? It was shown in the figure that crude membrane prep improved the overall sequence coverage of membrane proteins. It would be good also to show such improvement in individual membrane proteins. In addition, it needed to be clarified if the crude membrane prep increased the number of membrane proteins that can be identified using LiP-MS.

Additional points

1. The title of the manuscript is unclear and currently sounds overstated (as discussed above).
2. It was stated in the abstract, "Here, we adapted limited proteolysis-mass spectrometry (LiP-MS) to systematically identify putative structure-specific PPIs by probing protein structural alterations within cellular extracts upon treatment with specific structural states of a given protein." The changes made to the previously reported method that enabled the study here should be clearly pointed out in the manuscript.
3. When spiking 1 μ g RSVF into 100 μ g HEK239 cell lysate, what is the position of RSVF in the index of the abundance of all detected proteins, aka, was RSVF a high, medium or low abundance protein in the complex protein mixture? I wonder if a dilution series would make sense to demonstrate an exemplary sensitivity limit.
4. The authors wrote that the putative interacting proteins of α -syn fibril detected by both LiP-MS and the ultracentrifugation experiment likely have a higher affinity to α -syn fibril. As supporting evidence, these proteins "have a steeper dose-response curve compared to those identified only by LiP-MS". This is a very vague description. The data should be elucidated, for example, using a figure.
5. The procedures for "Preparation of crude membranes for MS analysis" read very confusing in the methods. It was written,

"The remaining pellets were resuspended in 15 mL LiP buffer and further homogenised using a dounce homogeniser with 20 strokes.", which pellets were used here, the pellets form at the first centrifugation step after lysis (1,000 x g at 4 {degree sign}C for 10 min.), the pellets formed in the ultracentrifugation step or all of them? Was each pellet resuspended in 15 mL LiP buffer, or was in total 15 mL LiP buffer used for all pellets?

Reviewer #2:

The manuscript by A. Holfeld et al. entitled "Systematic identification of structure-specific protein-protein interactions" describes an innovative approach to study protein-protein interactions, which is also applicable to systems where the different structures of a protein of interest can be generated in vitro so that they can be added to the protein solution with the potential binders (see also below: the presence of different structures is one use example, but is not a requirement). After allowing the interaction to take place, the samples are treated with Proteinase K for limited proteolysis prior to tryptic digestion and conventional LC/MS. Different concentrations of the protein (structures) of interest are spiked into the solution of potential binders, so that all peptides have to be quantified across the different concentrations to identify those that show sigmoidal dose-response curves - those are the peptides of interest.

The authors provide two proofs of concepts (although one could argue that neither of them is a good example for the story the authors try to tell): firstly, the binding of 5 different antibodies to the pre- and postfusion state of respiratory syncytial virus F (RSVF) glycoprotein. Secondly, the interaction of calmodulin to the regulatory subunit AC8-C2b of adenylyl cyclase type 8 (AC8).

After those proof of concepts, the authors proceed with the most relevant dataset: identifying the interaction partners of alpha-synuclein monomers and alpha-synuclein fibrils. Their work nicely shows commonalities, but also some clear differences between the PPI of those two forms of alpha-synuclein.

The manuscript in its current form describes an interesting body of work, but there are several limitations, which have to be addressed before this manuscript can be considered for publication in MSB.

The biggest issue of the current manuscript is the misleading title and paradigm: "systematic identification of structure-specific PPIs". This fuzziness becomes apparent when taking a closer look at the three presented case studies:

In the case of the RSVF glycoprotein, the two different structures of the same protein are incubated with different antibodies. The LiP-MS method is then used akin to good old footprinting methods to determine where the antibodies bind along the two structures of the same bait protein, i.e., this is more LiP-MS-based structural biology and confirms the known specificity of particular antibodies for either of the two structures of RSVF glycoprotein.

In the case of CaM-binding to AC8-C2b, there is nothing about structural diversity. The authors simply show that different concentrations of CaM result in different levels of CaM-binding, which can be recapitulated when (again) using LiP-MS-based footprinting, i.e., monitoring AC8-C2b peptides. In addition, peptides from other proteins with CaM-binding motifs are identified. It is not clear where any structure-specificity is covered by this example.

In short, those two examples have little to do with identifying structure-specific PPIs and have more to do with identifying binding domains using limited proteolysis (instead of e.g. HDX).

Finally, the third example shows nicely that their approach can detect PPI specific to alpha-synuclein monomers or fibrils. However, the fact that alpha-synuclein has two different quaternary structures is secondary for the method at hand. Instead, their approach can be used for any protein for which different proteoforms can be generated and isolated - whether the different proteoforms are now different quaternary structures, or different phosphorylation states, or whether is truncated, is irrelevant for the described method.

The fuzziness of the concept structure is furthered by the fact that the authors refer to the different structures of the bait protein (i.e., alpha-synuclein monomer vs. fibril), and on top they keep referring to structures altered upon addition of alpha-synuclein. However, whether this different protease K accessibility is due to interaction-induced structural changes or simply by protection of the interaction surface cannot be assessed. While I am favoring the latter, the author seem to favor the former without providing any proof - their preference is apparent based on their statements such as "among other proteins structurally altered upon addition of aSyn fibrils..." (throughout pages 10 and 11, pp). If the authors refer to alterations in quaternary structure, then these statements are correct. However, they cannot draw any conclusions regarding the alterations in secondary or tertiary structures.

In short, if the authors were to take off their 'structural proteomics hat' and thought a bit broader, the paper would be more convincing. Or in other words: the authors have to streamline the message and the examples, and focus on the most salient point for their story: LiP-MS can be used to identify protein interactors; the structure aspect is secondary/irrelevant for the method itself. At the moment, it is a bit of a hotchpotch of things, that don't completely go together.

Figure 1: The relevant domains are hard to see. Please work on the contrast and make them more apparent.

Figure 1 and 2: It does not happen too often these days, but in this case, I am asking for some nice representation of MS1 and MS2 data, which will be more informative than showing 6 times the pre- and 6 times the postfusion versions of RSVF glycoprotein.

Figure 2: "Of the 14 peptides identified in the single-dose experiment, the intensity responses of eight peptides were proportional

to the amount of motavizumab with high correlation ($r > 0.85$; Figure 2a)". I presume the author mean anti-proportional changes (DEcreasing peptide abundances with INcreasing antibody amounts). The fact that the authors see most prominently decreasing peptide abundances upon increasing amounts of binders supports the notion of conventional footprinting and not interaction-induced structural alterations as it seems to be implied throughout the manuscript.

In the AC8/CaM section, the author emphasize the applicability to IMPs. Later they mention that 56 proteins/85 peptides are predicted to contain CaM-binding motifs - how many of these 56 proteins are IMPs and how many are soluble/cytosolic proteins. Please clarify.

One of the limitations of the method is the need for dose-response curves, which significantly increases the experimental and instrument time burden. However, this need for numerous experiments is not mentioned once, but is an important consideration and thus should be discussed at the latest in the discussion section.

Page 14: The LiP-MS story is interesting enough, so that the conventional co-precipitation experiment is confusing especially as it shows rather little overlap with the LiP-MS experiments. As such, I would argue that this experiment does not add, but confuses.

Discussion section: the author talk about pinpointing the EXACT location of the three antigenic sites. Given the numerous peptides over a wide surface area of the RSVF glycoprotein, I would strongly suggest to omit the word 'exact'.

Discussion (see also above): "Notably, we have identified novel aSyn monomer-specific structural alterations on proteins involved in RNA binding and protein binding." Unless the authors refer to alterations in the quaternary structure, I strongly disagree with this statement. Instead, I argue, they simply see changes as the PPI protects a certain region in the protein.

Figures in general: please provide the amino acid numbers when providing a peptide sequence. In the case of Figure 2a, please indicate whether the listed peptide is from the 'red' or the 'green' domain.

Reviewer #3:

The authors describe the application of Limited Proteolysis-MS (LiP-MS) to map protein-protein interactions (PPIs) on several target proteins. These include known, well-characterized drug-target interactions and novel PPIs relevant for several neurobiological diseases. The manuscript thus contains a proof of concept as well as an application for a disease-related topic. The data are solid and analyzed thoroughly, the manuscript is well-written and the results and conclusions are of interest for a broad scientific audience. As such, the manuscript fits the scope of Molecular Systems Biology, and I recommend acceptance once the points listed below have been addressed.

Major points:

The data are clear and convincing, but independent, second generation experiments to validate the findings would strengthen the manuscript significantly. While quantitative proteomics experiments were conducted, these rely on a similar experimental approach. I'd suggest to use e.g. immunofluorescent imaging experiments, showing that the interactors identified using LiP-MS also co-localize with aSyn monomers or fibrils. Alternatively, reverse co-immunoprecipitation experiments showing that aSyn monomers or fibrils can be bound by the novel identified interaction partners could be used.

Minor points:

RSVF glycoprotein section: One question about the terminology for "peptides showing changes" - did you observe both increase and decrease in susceptibility to proteolysis in different stretches of the RSVF glycoprotein or were all changes "one-directional"? In comparison, for example for HDX-MS, you typically see both decreases in deuterium uptake in the segments directly involved in binding, but also increases (and decreases) in deuterium uptake in other segments undergoing conformational changes during ligand binding. If I remember correctly this was examined in previous LiP-MS publications.

PostRSVF interaction with motavizumab in complex extract and other affected proteins: There are apparently other proteins which also came up in the first experiment. The dose-dependence shown in Figure 2 is a clear proof of the target PPI, but is there a way to discriminate between off-targets and the real target already in the initial experiment? I.e. were the effects less pronounced for the other proteins? The raw peptide numbers already give a good indication (8 of 14 peptides were found for the known target), but any additional metric would be valuable.

AC8 - CaM interaction: The data are convincing and well presented, however the "other proteins" could be analyzed in more details. Some of these proteins contain CaM-binding motifs, but what about the rest? Are they functionally related and are we looking at secondary effects? A more detailed analysis using e.g. string-db or other bioinformatics analyses would be helpful. With the high number of detected peptides in a lysate, it's not surprising to see some background. However it would be great to have a visual readout or quality check, indicating which proteins are the most likely direct interaction partners of the tested protein.

aSyn monomers vs. fibrils: The data for the known interaction partners are clear, but the "grading" of the other identified PPIs could be explained more clearly. Is there a way to create a list sorted by probability or strength of interaction, i.e. in such a way that most likely direct interactors are on top and proteins with less pronounced effects at the bottom? Also, could the authors comment on how "complete" the aSyn interactome was mapped? I.e. how many interactions are listed in STRING (with which stringency settings used?) and what's the percentage of these PPIs detected by LiP-MS here?

A final minor comment for the intro: The authors may consider mentioning "Thermal Proteome Profiling" approaches, which can also be used to identify drug-target interactions (with the limitation that TPP does not provide interaction sites).

Reviewer #4:

In this article, Holfeld et al use LiP to probe PPIs. More specifically, they spike recombinant proteins in cell extracts and employ LiP to detect variations in proteolytic signatures that are caused by alteration of binding sites and/or conformation changes. This manuscript presents an interesting series of results that demonstrate how LiP can also be applied to probe PPIs at a global scale. It also provides novel interesting insights related to alpha-synuclein interactomes. I recommend this article for publication providing the authors can address the minors points raised below.

It seems that different cut off were used: \log_2 FC >1 in page 4, then 0.75 on page 7. Unless there is a specific rationale, the same cut off should be used across the study.

In principle a reduction of tryptic peptide signature should be caused when the majority of the protein within a sample is affected. In contrast, an increase of a semi-tryptic peptide could be detected even when sub-stoichiometric fraction is affected. Using that scheme, can the authors differentiate between stable vs more transient PPIs.

The authors should include a table to summarize which peptides were significantly more or less detected in presence of the different antibodies and their positions. Reading the text, it is not always obvious how the peptide signature was altered more precisely. In principal, tryptic peptides and semi tryptic peptides that cover the same binding motif/region should show opposite trends.

Instead of showing the semi-tryptic peptides on the structure, the position of PK cleavage site (e.g. P1 site) would more precisely show which residues are more (or less) accessible upon antibody binding.

How do the authors know which conformation of RSVF is present in their samples (pre or post)? Unless I missed that information, more clarifications should be provided.

Figure 2a and S1. Specify n and indicate whether the peptides are semi-tryptic or not

About 1/3 of LIP peptide detected upon the addition of CaM are located CaM binding motifs. What about the other 2/3. Do these proteins also have CaM motifs but mapped elsewhere? The authors should add some clarifications.

Page 12. Line 4. Are the p-values and FC related to this study or Xia et al 2008. If the former, the threshold should be adjusted to be the same across the study

Page 12 Figure S1d should probably be S4d

Page 12 last sentence: add citation(s).

Page 14. The section related to GWAS analysis is underdeveloped and should be removed if not appropriately substantiated. What is the meaning of the analysis? Do the authors think that SNPs in some of these GWAS could weaken some of the PPIs?

MSB-2023-11552

Systematic identification of structure-specific protein-protein interactions

Summary of revisions

We thank all referees for their insightful comments and summarize here the main changes to the revised manuscript. Details are below in our point-by-point response. Please note that line numbers refer to the manuscript with embedded figures, provided for the referee's convenience.

1. We have added new data on the identification of conformation-specific interactions. Here, we used our approach to compare candidate interactors of two conformational states of Rab GTPase proteins: the GTP- and GDP-bound forms of Rab2A and Rab5A (lines 455-492, Revised Figure 5). Briefly, we identified a number of significantly changing peptides and proteins upon addition of increasing amounts of Rab2A-GTP, Rab2A-GDP, Rab5A-GTP and Rab5A-GDP to HEK293T cellular extracts (Revised Figure 5A, B). The set of proteins that changed was different for the GTP and GDP bound forms of both Rabs (Revised Figure 5C, D) thus showing that we can detect differences in candidate interactors even of proteins that are similar in structure. Our candidate interactors for each of these proteins were significantly enriched in previously identified hits from a proximity labeling approach (MitolD, based on BioID proximity labeling of mitochondrially-targeted proteins; Gillingham et al, 2019) (Revised Figure 5E-H). As expected, we also identified different sets of candidate interactors for the two Rabs (Revised Figure 5I-K).
2. We have revised the manuscript to better separate the initial methods development, in which we demonstrate the approach on well-characterized protein-protein interactions, and the application to discovery of conformation-specific interactions. As mentioned above, we have now added another example of using the method for identifying conformation specific interactors.
3. We have revised the manuscript to make clearer that our method measures protease susceptibility changes which could either indicate protection of the interaction interface upon protein-protein interaction (i.e., a footprinting scenario) or binding-related allosteric changes in a region of the protein other than the interface (lines 152-158, lines 219-232). Our language in the initially submitted version may have caused some misunderstanding on this point and we hope that the manuscript is now clearer.
4. Related to point 3, we have revised the manuscript to provide a more nuanced picture of how changes in peptide intensity may be interpreted. For the experiments on RSVF-antibody binding, we have annotated all data to indicate whether a given RSVF peptide is semi-tryptic (ST) or fully tryptic (FT) (i.e., with one or two tryptic ends, respectively) as well as the directionality of the intensity change upon antibody addition. For those peptides at known RSVF antigenic sites we show that, as expected, ST and FT peptides show opposite intensity changes and that these

indicate protection of the interaction site in the presence of antibody (lines 152-158, lines 219-232).

Reviewer 1

In the manuscript titled "Systematic identification of structure-specific protein-protein interactions", the authors applied LiP-MS to identify potential interactors to a given target (bait) protein by probing changes in the digestibility of proteins upon the addition of the bait protein. This is similar to fishing interaction partners using recombinant bait proteins, albeit with the marked difference that in the case of LiP-MS the interaction does not have to survive subsequent biochemical enrichment. Also, in the case of LiP-MS there would be no background from unspecific binding of proteins to beads used during the biochemical enrichment. In consequence, LiP-MS should give much cleaner data. I wonder if the authors might be well advised to demonstrate this by help of a comparison, using some tagged recombinant protein in a classical pulldown and using LiP-MS.

We thank the referee for their comments and note that we have addressed their specific points below. Please see the summary for all referees (above) for an overview of the new data added to the revised manuscript.

While we agree that in principle a direct comparison to AP-MS might be useful, the results would depend substantially on the affinity of the antibody used for AP-MS and the properties of the bait protein, and would thus not be generalizable. Given (i) that we have validated the approach using known antibody-target and other known protein-protein interactions, (ii) that the problem of background in AP-MS is widely appreciated, and (iii) that our approach has the unique strength of being applicable to different protein conformations without the need for conformation-specific antibodies, we have not added a direct comparison to AP-MS to the revised manuscript.

As a benchmark, the authors demonstrated, using purified RSVF and its site-specific antibodies, that LiP-MS could detect interactions between RSVF and antibodies accurately. The structurally altered peptides are located at and near the expected binding sites of respective antibodies. Such interactions were also detected when spiking the RSVF in the background of HEK293 cell extract, where a dose-response experiment was necessary to distinguish true and false positive hits. A second proof of principle analysis was conducted to show that LiP-MS could detect protein-protein interaction to a membrane protein. Here, LiP-MS was applied to detect a known interaction between calmodulin and a bovine membrane-integral protein CA8 which is over-expressed in the HEK239 cells. Structural changes were also identified for additional proteins from the HEK239 cell extract, a subset of them were predicted to contain calmodulin-binding motifs. In the end, authors applied LiP-MS to detect, in the extract of SNCA-KO iPSC-derived neurons, the interacting proteins of α -synuclein (α -syn) monomer and α -syn amyloid fibrils, respectively. Although there are some overlaps, different sets of interacting proteins were detected interacting with α -syn monomer and amyloid fibrils. The results were further verified by 1) GO enrichment and network enrichment analysis of the detected α -syn interacting

proteins.

2) comparing to the results of previously reported proteomics studies.

3) comparing to proteins that co-precipitation with the α -syn amyloid fibril by ultracentrifugation.

The studies reported in the manuscript were appropriately conducted. It is worth publishing as a successful application of LiP-MS on identifying interacting proteins of different forms of α -syn. Otherwise, additional data must be provided to support several general statements made about the methodology.

1. The authors stated that LiP-MS can be applied to identify protein-protein interactions specific to protein structural states accurately. A single example was shown using a comparison between α -syn monomer and amyloid fibrils. The difference in structure between these two forms of the protein is significantly greater than the typical variation in structural states of most proteins and protein complexes. Therefore, it is necessary to conduct additional tests using bait proteins with varying degrees of structural differences to validate the general applicability of LiP-MS to identify PPIs for distinct structural states of proteins.

We have now added data using LiP-MS to compare candidate interactors of two conformational states of Rab GTPase proteins: the GTP- and GDP-bound forms of Rab2A and Rab5A (lines 455-492). Briefly, we identified a number of significantly changing peptides and proteins upon addition of increasing amounts of Rab2A-GTP, Rab2A-GDP, Rab5A-GTP and Rab5A-GDP to HEK293T cellular extracts (Revised Figure 5A, B; reproduced below). The set of proteins that changed was different for the GTP and GDP bound forms of both Rabs (Revised Figure 5 C, D) thus showing that we can detect differences in candidate interactors even of proteins that are similar in structure.

We further analyzed the set of candidate interactors in light of the literature. Most known effectors of Rab2A-GTP and Rab5A-GTP were not detected in the mass spectrometer, most probably because of their low abundance. Only 5 known effectors were MS detectable but our LiP-MS screen did not identify them as interactors, possibly because of low sequence coverage (average of 36.6%).

We also compared our candidate interactors to the sets of proteins previously identified in a proximity labeling approach (MitoID, based on BioID proximity labeling of mitochondrially-targeted proteins) (Gillingham et al, 2019), for each form of each Rab. In each case, our LiP-MS candidate interactors were significantly enriched in MitoID hits (Revised Figure 5E-H). However, our candidate interactors are of low specificity based on a score assessing the MitoID hits for a given Rab relative to those for multiple (n=11) Rabs. This is because proteins that were identified as specific interactors based on this criterion tend to be at low abundance and are therefore not detectable in the mass spectrometer. Despite this, the set of candidate interactors that overlapped with MitoID hits differed between the two Rabs (Revised Figure 5I-K).

Revised Figure 5: (A-B) Barplot with numbers of significantly changing peptides (blue) and corresponding proteins (green) for Rab5A (A) or Rab2A (B) in their GTP- (left) or GDP-bound (right) forms. (C-d) Venn diagrams with the overlap of proteins identified as significantly changing upon addition of GTP and GDP bound forms of Rab5A (C) or Rab2A (D). (E-F) The plots show the fraction of MitolD hits in the set of LiP-MS candidate interactors (right) versus all detected proteins (left) upon spike-in of the indicated proteins into a HEK293T cellular extract. The p-value assessing enrichment (Fisher's exact test) is shown. (I-K) Venn diagram showing interactors identified by both MitolD and LiP-MS, compared between both forms of Rab5A and Rab2A (I), GTP-bound forms (J), and GDP-bound forms (K) of the two proteins.

2. The authors promoted the use of detergent-free crude membrane prep to enable the detection of PPIs to membrane proteins. The interaction between calmodulin and CA8 has been identified using crude membrane prep. Would it be possible to identify this interaction using the native lysate

The peptides of AC8 that changed upon calmodulin addition in the membrane prep were not all detected in the native lysate. In addition, the overall sequence coverage of AC8 is lower in a native lysate (45.2%) compared to that in membrane suspensions (56.4%). This result itself indicates that the membrane preparation improves the detection of AC8 peptides. We therefore did not repeat the calmodulin titration series in native lysate as this would be unlikely to detect an effect. We show the peptide-level coverage data in Figure 3C and 3D in the revised test manuscript, reproduced below, and describe this in lines 280-282.

N-terminal AC8-CaMBD
 LSGSEELYTIHPTPPAGDSGSGSRPQRLLWQTAVRHITQRFI¹HHG
 LSGSEELYTIHPTPPAGDSGSGSRPQR
 SGSEELYTIHPTPPAGDSGSGSRPQR
 TIHPTPPAGDSGSGSRPQR
 IHPTPPAGDSGSGSRPQR

C-terminal AC8-CaMBD
 RLPGQYSLA¹AVVLLGLVQSLNRQRQKQLLNENNNTGI¹IK
 RLPGQYSLA¹AVVLLGLVQSLNR LNENNNTGI¹IK
 RLPGQYSLA¹AVVLLGLVQSLNR NENNNTGI¹IK
 AVVLLGLVQSLNR
 VVLLGLVQSLNR
 GLVQSLNR

Revised Figure 3: LiP-MS detects interactors of integral membrane proteins in crude membranes. (A) Schematic of AC8 with the CaMBD in the N-terminus, transmembrane domains 1-6 and 7-12 (TM1-6 and TM7-12), and catalytic domains C1a, C1b, C2a, and C2b indicated. (B) Distribution of protein coverage for membrane-annotated proteins identified in crude membrane preparations of HEK293F GnTI- cells (blue) and in HEK293T cellular extracts (green). Blue and green vertical lines indicate calculated median coverages of 29.6% and 17.6%, respectively. (C, D) Protein sequence coverage of bovine AC8-YFP in LiP-MS in crude membranes (C) and in cellular extracts (D) is visualized. The barcodes depict peptides along the AC8-YFP sequence. Gray represents detected peptides, white represents non-detected regions, and red represents peptides that were significantly altered upon CaM addition ($r > 0.85$, $|\log_2 FC| > 1$, moderated t-test, $qvalue < 0.01$). (E) AlphaFold2-predicted

(Varadi et al, 2022; Jumper et al, 2021) structure of AC8 (including the tag domain) with peptides altered upon CaM addition highlighted in red. Amino acid sequences comprising the CaM-binding motifs of AC8 are depicted in black. Hydrophobic residues of the CaM-binding motif are underlined. The significantly altered peptides upon CaM addition are shown in red.

It was shown in the figure that crude membrane prep improved the overall sequence coverage of membrane proteins. It would be good also to show such improvement in individual membrane proteins.

Protein sequence coverage in the two preparations is now shown for 1478 individual membrane-annotated proteins in Dataset EV 3.

In addition, it needed to be clarified if the crude membrane prep increased the number of membrane proteins that can be identified using LiP-MS.

We identified 3037 membrane-annotated proteins in the crude membrane prep and 1506 membrane proteins in the full lysate; this is now reported in the revised manuscript (lines 256-257).

Additional points

1. The title of the manuscript is unclear and currently sounds overstated (as discussed above).

We hope that our additional example of candidate structure-specific interactions addresses the perceived overstatement.

2. It was stated in the abstract, "Here, we adapted limited proteolysis-mass spectrometry (LiP-MS) to systematically identify putative structure-specific PPIs by probing protein structural alterations within cellular extracts upon treatment with specific structural states of a given protein." The changes made to the previously reported method that enabled the study here should be clearly pointed out in the manuscript.

We have removed the word "adapted" from the abstract and replaced it simply with "used", which is also accurate and less distracting. We had already described the two adaptations we meant, namely applying LiP-MS in a protein spike-in context and using the concept of dose-response curves to identify peptides more likely to report direct true positive protein interactors. We now include a paragraph in the discussion (lines 504-518) that clearly distinguishes the multi-dose approach we use here for PPIs from a similar but not identical approach (LiP-Quant) that we had previously described for the study of protein-small molecule interactions.

3. When spiking 1ug RSVF into 100 ug HEK239 cell lysate, what is the position of RSVF in the index of the abundance of all detected proteins, aka, was RSVF a high, medium or low abundance protein in the complex protein mixture? I wonder if a dilution series would make sense to demonstrate an exemplary sensitivity limit.

We compared the signal intensity of spiked-in RSVF to that of proteins in our lysate and find that RSVF is the most abundant protein compared to all other proteins. This is now reported in the revised manuscript (line 210) and shown in Revised Appendix Figure S1A, reproduced below.

Revised Appendix Figure S1A. The plot shows the ranked abundance of all the identified proteins (gray). The spiked-in RSVF protein is shown in blue.

While we could in principle carry out a dilution series with RSVF to define sensitivity limits as the referee suggests, this will not be generalizable to other proteins since the results will strongly depend on features of the protein being probed and of its interactors (i.e., difference in proteolytic accessibility of the bound and unbound states, sequence coverage, coverage of interaction sites, expression levels of interactors, behavior of peptides in the mass spectrometer). Since our goal is not to develop a specific assay for RSVF or RSVF antibodies here, we have therefore not done this experiment.

4. The authors wrote that the putative interacting proteins of α -syn fibril detected by both LiP-MS and the ultracentrifugation experiment likely have a higher affinity to α -syn fibril. As supporting evidence, these proteins "have a steeper dose-response curve compared to those identified only by LiP-MS". This is a very vague description. The data should be elucidated, for example, using a figure.

We now write this more precisely (lines 440-443). We used the LiP-MS-based dose response curves of candidate interactor peptides upon spike-in of increasing amounts of aSyn fibrils to estimate a median half-maximal response concentration for the 53 hits that were detected by both LiP-MS and AP-MS, compared to those detected by LiP-MS alone. We observed a half-maximal response concentration of 1.7 μ g for hits detected with both techniques and of 2.7 μ g for hits detected only with LiP-MS; this is consistent with our hypothesis that the former may represent higher affinity interactions. This is now written more clearly and is shown in Revised Appendix Figure S5F, reproduced below. Note that we have shown in the past that the slope of the LiP-MS based dose response curves report on affinity for protein-small molecule interactions (Piazza et al, 2020). While this is likely also

for protein-protein interactions, it remains to be rigorously tested, as we had already noted in the discussion section (lines 510-515 in revised manuscript).

Revised Appendix Figure S5F. The box plots compare the distribution of half-maximal response concentrations for proteins identified as aSyn fibril candidate interactors by LiP-MS alone versus proteins identified by both LiP-MS and ultracentrifugation. The median (line within box), interquartile range (box), potential outliers (individual data points), and p-value (Wilcoxon test) are shown.

5. The procedures for "Preparation of crude membranes for MS analysis" read very confusing in the methods. It was written, "The remaining pellets were resuspended in 15 mL LiP buffer and further homogenised using a dounce homogeniser with 20 strokes.", which pellets were used here, the pellets form at the first centrifugation step after lysis (1,000 x g at 4 {degree sign}C for 10 min.), the pellets formed in the ultracentrifugation step or all of them? Was each pellet resuspended in 15 mL LiP buffer, or was in total 15 mL LiP buffer used for all pellets?

We have clarified the methods. The text is now "The remaining (membrane enriched) pellets from the second centrifugation step were combined, resuspended in a total of 15 mL LiP buffer and further homogenized using a dounce homogenizer with 20 strokes. The total protein concentration of crude membranes was determined with a Pierce BCA Protein Assay Kit (cat #23225) according to the manufacturer's instructions. Crude membranes were stored at -80 °C prior to further use." (lines 796-800).

Reviewer 2

The manuscript by A. Holfeld et al. entitled "Systematic identification of structure-specific protein-protein interactions" describes an innovative approach to study protein-protein interactions, which is also applicable to systems where the different structures of a protein of interest can be generated in vitro so that they can be added to the protein solution with the potential binders (see also below: the presence of different structures is one use example, but is not a requirement). After allowing the interaction to take place, the samples are treated with Proteinase K for limited proteolysis prior to tryptic digestion and conventional LC/MS. Different concentrations of the protein (structures) of interest are spiked into the solution of potential binders, so that all peptides have to be quantified across the different concentrations to identify those that show sigmoidal dose-response curves - those are the peptides of interest. The authors provide two proofs of concepts (although one could argue that neither of them is a good example for the story the authors try to tell): firstly, the binding of 5 different antibodies to the pre- and postfusion state of respiratory syncytial virus F (RSVF) glycoprotein. Secondly, the interaction of calmodulin to the regulatory subunit AC8-C2b of adenylyl cyclase type 8 (AC8). After those proof of concepts, the authors proceed with the most relevant dataset: identifying the interaction partners of alpha-synuclein monomers and alpha-synuclein fibrils. Their work nicely shows commonalities, but also some clear differences between the PPI of those two forms of alpha-synuclein. The manuscript in its current form describes an interesting body of work, but there are several limitations, which have to be addressed before this manuscript can be considered for publication in MSB.

The biggest issue of the current manuscript is the misleading title and paradigm: "systematic identification of structure-specific PPIs". This fuzziness becomes apparent when taking a closer look at the three presented case studies: In the case of the RSVF glycoprotein, the two different structures of the same protein are incubated with different antibodies. The LiP-MS method is then used akin to good old footprinting methods to determine where the antibodies bind along the two structures of the same bait protein, i.e., this is more LiP-MS-based structural biology and confirms the known specificity of particular antibodies for either of the two structures of RSVF glycoprotein.

The experiments testing antibody-target (RSVF) interaction and AC8-calmodulin interaction were a part of our methods development. Our goal in this part of the study was to analyse whether LiP-MS can detect protein-protein interactions in vitro and in a complex background. We needed well studied interactions to assess this, for which we chose antibody-target (RSVF) interactions. The experiment with AC8-calmodulin was done to test whether the approach could extend to membrane proteins, a particularly interesting class of proteins. This said, some of the RSVF experiments do have a structure-specific component, since they show that LiP-MS identifies the expected structure-specific interactions of the D25 and 5C4 antibodies with the pre-fusion but not the post-fusion conformation of RSVF.

Having established that LiP-MS can detect PPIs, and can do so in a complex cellular context, we have then applied it to look for differential interactors of aSyn monomers and fibrils, and newly of Rab GTPases in two conformational states (GTP and GDP bound forms of both Rab5A and Rab2A). These applications are meant as an example of the unique aspect that the approach can be used for structure-specific studies. We have revised the manuscript to make clearer the methods development and application aspects (see lines 180-182, 247-249, 292-293).

In the case of CaM-binding to AC8-C2b, there is nothing about structural diversity. The authors simply show that different concentrations of CaM result in different levels of CaM-binding, which can be recapitulated when (again) using LiP-MS-based footprinting, i.e., monitoring AC8-C2b peptides. In addition, peptides from other proteins with CaM-binding motifs are identified. It not clear where any structure-specificity is covered by this example.

Please see our comments above. This experiment indeed does not probe structure-specific interactions, and we have made it clearer in the revised manuscript that we are here testing the applicability of the basic method (i.e., using LiP to detect PPIs) for the interesting class of membrane proteins, which are hard to study using existing methods (lines 247-249). Note however that our data overall suggest that in principle we should be able to probe structure specific interactors of membrane proteins, as long as the interactions happen within the soluble portions of these proteins.

In short, those two examples have little to do with identifying structure-specific PPIs and have more to do with identifying binding domains using limited proteolysis (instead of e.g. HDX).

As discussed above, these experiments were mainly part of methods development, although some of the experiments with anti-RSVF antibodies do show structure-specific interactions.

HDX could in fact not be used to detect interactions in medium or high-complexity samples such as lysates or membrane preps as it typically requires purified proteins, but we show here that this can be achieved with LiP-MS.

Finally, the third example shows nicely that their approach can detect PPI specific to alpha-synuclein monomers or fibrils. However, the fact that alpha-synuclein has two different quaternary structures is secondary for the method at hand. Instead, their approach can be used for any protein for which different proteoforms can be generated and isolated - whether the different proteoforms are now different quaternary structures, or different phosphorylation states, or whether is truncated, is irrelevant for the described method.

Yes, that is exactly right. The approach could be used for any two stable proteoforms that can be spiked into a lysate, as we mention in the abstract (lines 41-43) and discussion (lines 495-497).

We have now added a second example of this type of application, in which we tested two forms of a protein that are expected to differ much more subtly in their conformation. We tested GDP- and GTP-bound versions of two Rab GTPases, Rab5a and Rab2a, which should be much more similar in structure than the aSyn monomer and fibril. We identified differential candidate interactors, both when comparing GTP and GDP bound forms of each protein, and when comparing the two proteins to each other. These data are now shown in Revised Figure 5 (above, see response to referee 1) and described in lines 455-492.

The fuzziness of the concept structure is furthered by the fact that the authors refer to the different structures of the bait protein (i.e., alpha-synuclein monomer vs. fibril), and on top they keep referring to structures altered upon addition of alpha-synuclein.

However, whether this different protease K accessibility is due to interaction-induced structural changes or simply by protection of the interaction surface cannot be assessed.

While I am favoring the latter, the author seem to favor the former without providing any proof - their preference is apparent based on their statements such as "among other proteins structurally altered upon addition of aSyn fibrils..." (throughout pages 10 and 11, pp). If the authors refer to alterations in quaternary structure, then these statements are correct. However, they cannot draw any conclusions regarding the alterations in secondary or tertiary structures.

We think this comment is the result of a misunderstanding. We accept that we did not describe our data and interpretations well enough.

To clarify our intentions for the referee: our readout is peptide intensity changes between conditions (i.e., with and without spike-in). This reports on changes in protease susceptibility between these conditions. Some of these changes in protease susceptibility will be due to a direct protection of interaction sites (these are the ones we are interested in here). Others may be due to an interaction-induced conformational change such as an allosteric change. In addition, there may be secondary events that occur due to an interaction (i.e., our spiked-in protein could interact with X, which could in turn change its interaction with Y). The referee is correct that we cannot distinguish between these types of events based on just a LiP-MS experiment, nor do we claim to. We certainly do not wish to imply that we favor any particular interpretation of the protease susceptibility changes. We simply wish to be clear that one must be circumspect about any such changes that occur upon spike-in and that one cannot assume based on these data alone that they reflect direct PPI with the spiked-in protein.

We had used the term "structural alteration" as an umbrella term to reflect both potential quaternary structural changes (i.e., PPIs) and other potential structural changes upon spike-in. Based on the referee's comments, we now describe the changes that we see as "protease susceptibility changes" and refer to all hits as candidate interactors.

In short, if the authors were to take off their 'structural proteomics hat' and thought a bit broader, the paper would be more convincing. Or in other words: the authors have to

streamline the message and the examples, and focus on the most salient point for their story: LiP-MS can be used to identify protein interactors; the structure aspect is secondary/irrelevant for the method itself. At the moment, it is a bit of a hotchpotch of things, that don't completely go together.

The manuscript shows that LiP-MS can detect known PPIs in a complex lysate, and goes on to demonstrate that it can do so for different structural forms of a protein. We do not think this is irrelevant or secondary at all, but rather a unique and interesting use case. We have now added textual clarification to better separate the methods development and the application to identifying structure-specific interactors and have added an example comparing structural forms of two different Rab GTPases which are expected to be much more similar to each other than aSyn monomer and fibrils. We hope that this makes our line of argument clearer and more convincing.

Figure 1: The relevant domains are hard to see. Please work on the contrast and make them more apparent.

We have improved the contrast as requested. The figure is reproduced below.

Revised Figure 1: LiP-MS detects protein-protein interactions in purified systems. (A) Schematic of LiP-MS workflow. Proteins are extracted from an experimental model, such as tissues, human cells, bacteria, yeast, viruses, or biofluids, under native-like conditions. The extract is then exposed to a protein of interest (treated) or not exposed (control) and subjected to limited proteolysis with proteinase K. Under LiP conditions, proteinase K cleaves solvent-exposed, accessible, and flexible regions thus generating protein fragments that may differ between the treated and control samples for an interactor of the spiked-in protein. These protein fragments are digested by trypsin under denaturing conditions to produce

peptides that are measurable by bottom-up proteomics. By comparing differential peptides between the treated and control sample, interactors of the protein of interest can be identified. (B) Structures of preRSVF (left, PDB: 4JHW) (McLellan et al, 2013) and postRSVF (right, PDB: 3RRR) (McLellan et al, 2011). Known antigenic sites are shown both on the protein structure and in isolation (middle). Blue indicates antigenic site Ø, targeted by antibodies D25 and 5C4. Red indicates antigenic site II, targeted by palivizumab and motavizumab. Orange indicates antigenic site IV, targeted by 101F. (C) Visualization of structurally altered peptides ($|\log_2 FC| > 1$, moderated t-test, q-value < 0.01) in green, on one of the subunit of trimeric preRSVF (upper panel) and postRSVF (lower panel) protein structures upon addition of the indicated antibodies. Antigenic sites are colored as in panel (b).

Figure 1 and 2: It does not happen too often these days, but in this case, I am asking for some nice representation of MS1 and MS2 data, which will be more informative than showing 6 times the pre- and 6 times the postfusion versions of RSVF glycoprotein.

We now provide examples of extracted ion chromatogram (XIC) plots for two peptides of RSVF upon addition of motavizumab, which binds to antigenic site II (Revised Appendix Figure S1B-E, reproduced below). The first peptide INDMPITNDQK 2+ is a semi-tryptic peptide in antigenic site II. This peptide shows significant decreases in abundance with increasing concentrations of motavizumab (panel B), indicating reduced accessibility due to antibody-target binding. Similarly, treating purified RSVF with motavizumab causes a reduced abundance of this peptide versus treatment with a non-specific control antibody (panel C). The second peptide QLLPIVNK +1, which is not within antigenic site II, remains relatively constant upon motavizumab addition in both experimental setups (panels D and E).

B INDMPITNDQK (2+) (251-261)**C** INDMPITNDQK (2+) (251-261)**D** QLLPIVNK (+1) (192-199)**E** QLLPIVNK(+1) (192-199)
Revised Appendix Figure S1B-E. Extracted ion chromatogram (XIC) plots of the indicated peptides: INDMPITNDQK 2+; 251-261 (**B**, **C**) and QLLPIVNK +1; 192-199 (**D**, **E**). Plots depict the XIC for MS2 (upper row) and the MS1 Isotope Envelope (lower row) across all five conditions (**B**, **D**) (T0: untreated sample; T1-T5: treated samples with increasing concentrations of motavizumab) in cellular lysate ($n=3$ replicates each), and (**C**, **E**) in Motavizumab-treated versus IgG control-treated purified RSVF samples ($n=4$ replicates each).

"Of the 14 peptides identified in the single-dose experiment, the intensity responses of eight peptides were proportional to the amount of motavizumab with high correlation ($r > 0.85$; Figure 2A)". I presume the author mean anti-proportional changes (DEcreasing peptide

abundances with INcreasing antibody amounts). The fact that the authors see most prominently decreasing peptide abundances upon increasing amounts of binders supports the notion of conventional footprinting and not interaction-induced structural alterations as it seems to be implied throughout the manuscript.

Yes, we did mean this. We now say “inversely proportional” (line 216).

The general expectation is that semi-tryptic (ST) peptides (i.e, those with a single tryptic end) at a binding site will decrease in intensity when the antibody is added, since binding reduces accessibility to proteinase K. In contrast, fully tryptic (FT) peptides (i.e., those with two tryptic ends) at binding sites, and that encompass the PK cleavage site, would be expected to increase in intensity when the binder is added. This is indeed a classical footprinting scenario as the referee says, but with the added twist of ST and FT peptides which are in principle expected to behave in opposite ways.

The 8 peptides referred to here are all ST and are at (4 peptides) or near (4 peptides) the known motavizumab epitope, and this anti-correlation is therefore indeed consistent with antibody-target binding as the referee points out; this is now mentioned in the revised manuscript (lines 219-220). Our replacement of the term “structural alterations” with “altered protease susceptibility”, as discussed earlier, will hopefully resolve the referee’s general concern.

The referee’s comments suggest that a more detailed discussion in the manuscript of the behavior of ST versus FT peptides would be useful; this has now been added to the revised manuscript (lines 152-158). For the experiments on RSVF-antibody interactions, for which we have ground truth in the form of the known epitopes, we have now annotated Datasets EV1 and EV2 with this information.

We also analyzed the behavior of RSVF peptides that map to known epitope sequences, upon antibody spike-in. In all cases (D25, 5C4, Motavizumab, Palivizumab, 101F), peptides at epitopes indicate increased proteolytic protection upon addition of the antibody. Specifically, there are 3 FT peptides that go up and 8 ST peptides that go down in intensity upon antibody addition. This corroborates that our experiment is capturing the RSVF-antibody interaction in these cases.

In contrast, analysis of all peptides with altered intensity upon antibody addition (these include peptides at the epitope as before, but also peptides near the epitope in the 3D structure and those that are more distant) shows a more complex picture. Out of the 57 such peptides we detect in all experiments, 63% (36 peptides: 6 FT and 30 ST) are indicative of increased protection and 37% (21 peptides: 11 ST and 10 FT) are indicative of decreased protection. Note that, in analyses of proteolytic susceptibility, FT peptides are more reliable than ST peptides since the intensity of the latter can also be artifactually reduced even in conditions of increased cleavage of a given protein region, due to secondary cleavage events within the sequence of ST peptides (i.e., additional cleavage of a previously generated ST peptide). By contrast, increased cleavage within the sequence of a FT peptide will always result in a decrease of intensity of the FT peptide, and vice versa.

In any event, this analysis shows that some of the protease susceptibility changes we detect are unlikely to represent a footprinting effect, but may be due to other structural changes that occur as a secondary consequence of binding. We have now added a discussion of this analysis to the revised manuscript (lines 221-232).

In the AC8/CaM section, the author emphasizes the applicability to IMPs. Later they mention that 56 proteins/85 peptides are predicted to contain CaM-binding motifs - how many of these 56 proteins are IMPs and how many are soluble/cytosolic proteins. Please clarify.

37 of the 56 proteins are annotated as membrane proteins. We have now added this information to the revised manuscript (line 286).

One of the limitations of the method is the need for dose-response curves, which significantly increases the experimental and instrument time burden. However, this need for numerous experiments is not mentioned once, but is an important consideration and thus should be discussed at the latest in the discussion section.

We have mentioned this point in the discussion, where we also include a longer paragraph about the multi- versus single-dose approach (lines 504-518).

We note that dose-response curves are not strictly needed, and there may indeed be cases where a single-dose experiment is the better one (e.g., fewer resources or smaller sample amount, higher tolerance for false positives, availability of secondary screens, or a concern that the need for a sigmoidal profile may increase false negatives).

Page 14: The LiP-MS story is interesting enough, so that the conventional co-precipitation experiment is confusing especially as it shows rather little overlap with the LiP-MS experiments. As such, I would argue that this experiment does not add, but confuses.

We understand the reasoning, but decided to keep the data in the revised manuscript since it could help others interested in following up on the aSyn fibril hits to prioritize their efforts.

Discussion section: the author talk about pinpointing the EXACT location of the three antigenic sites. Given the numerous peptides over a wide surface area of the RSVF glycoprotein, I would strongly suggest to omit the word 'exact'.

We have removed the word exact.

Discussion (see also above): "Notably, we have identified novel aSyn monomer-specific structural alterations on proteins involved in RNA binding and protein binding." Unless the authors refer to alterations in the quaternary structure, I strongly disagree with this statement. Instead, I argue, they simply see changes as the PPI protects a certain region in the protein.

Again, this is the result of a misunderstanding of what we mean with “structural alterations” (see earlier discussion of this point). We have now revised to call these proteins “candidate interactors”. The sentence is now “Notably, we have identified novel aSyn monomer-specific candidate interactor proteins involved in RNA binding and protein binding.”

Figures in general: please provide the amino acid numbers when providing a peptide sequence. In the case of Figure 2A, please indicate whether the listed peptide is from the 'red' or the 'green' domain.

We have added amino acid numbers to Figures 2, 3 and Appendix Figures 1, 2.

For Figure 2A, all of the peptides are from the “green” domain (i.e., altered peptides, this is now indicated in the figure legend).

We note that the starting and ending aa are indicated for all peptides in Datasets EV1-5.

Reviewer #3:

The authors describe the application of Limited Proteolysis-MS (LiP-MS) to map protein-protein interactions (PPIs) on several target proteins. These include known, well-characterized drug-target interactions and novel PPIs relevant for several neurobiological diseases. The manuscript thus contains a proof of concept as well as an application for a disease-related topic. The data are solid and analyzed thoroughly, the manuscript is well-written and the results and conclusions are of interest for a broad scientific audience. As such, the manuscript fits the scope of Molecular Systems Biology, and I recommend acceptance once the points listed below have been addressed.

Major points:

The data are clear and convincing, but independent, second generation experiments to validate the findings would strengthen the manuscript significantly. While quantitative proteomics experiments were conducted, these rely on a similar experimental approach. I'd suggest to use e.g. immunofluorescent imaging experiments, showing that the interactors identified using LiP-MS also co-localize with aSyn monomers or fibrils. Alternatively, reverse co-immunoprecipitation experiments showing that aSyn monomers or fibrils can be bound by the novel identified interaction partners could be used.

We agree that an analysis of co-localization by imaging would in principle be interesting. However, the structural state of aSyn is unknown in the systems in which we are working (cortical neurons differentiated from human iPSCs). aSyn may be present as a mixture of structures, and there is limited evidence that it forms amyloid fibrils in cellular overexpression systems (Zamboni et al, 2019; Tofaris et al, 2001). In addition, the aSyn staining pattern is partially diffuse in these neurons (Reviewer Figure 1). A co-localization experiment would as a result be both difficult to do and to interpret as indicating interaction with aSyn monomers or fibrils. We therefore did not add any imaging data to the revised manuscript.

Day56cortical neurons, fixed 05.04.2023

Reviewer Figure 1. Immunofluorescence imaging for the indicated markers of cortical neurons after 56 days of differentiation from patient-derived induced pluripotent stem cells. Wild type neurons are in the upper row, aSyn knockout neurons are in the lower row. Images were taken with an SP8 confocal microscope, 63x.

In the initial submission, we had shown that aSyn fibril co-precipitation enriched for candidate fibril interactors identified by LiP-MS. We have now added a co-immunoprecipitation of aSyn monomer in a different cell line (SH-SY5Y). This identified seven proteins (EEF1A1, VIM, PCBP1, PAF1, TRAP1, GTF3C3, and RPL3) that overlap with the 64 candidate monomer interactions identified by LiP-MS. This is now mentioned in the revised manuscript (lines 442-444).

Minor points:

RSVF glycoprotein section: One question about the terminology for "peptides showing changes" - did you observe both increase and decrease in susceptibility to proteolysis in different stretches of the RSVF glycoprotein or were all changes "one-directional"? In comparison, for example for HDX-MS, you typically see both decreases in deuterium uptake

in the segments directly involved in binding, but also increases (and decreases) in deuterium uptake in other segments undergoing conformational changes during ligand binding. If I remember correctly this was examined in previous LiP-MS publications.

We have now annotated the Extended View Data files with information on whether a peptide is fully tryptic (FT) or semi-tryptic (ST) (i.e., having either two or one tryptic ends), and with the directionality of the change in peptide intensity (see lines 152-158).

For the RSVF-antibody experiments, the known epitopes provide a ground truth to assess peptide intensity changes in what is expected to be a classical footprinting scenario, with the added twist that ST and FT peptides are expected to behave differently. The general expectation is that ST peptides at an epitope will decrease in intensity when the antibody is added, since binding reduces accessibility to proteinase K (PK). In contrast, FT peptides at epitopes, and that encompass the PK cleavage site, would be expected to increase in intensity when the binder is added.

We analyzed the behavior of RSVF peptides that map to known epitope sequences, upon antibody spike-in. In all cases (D25, 5C4, Motavizumab, Palivizumab, 101F), peptides at epitopes indicate increased proteolytic protection upon addition of the antibody. Specifically, there are 3 FT peptides that go up and 8 ST peptides that go down in intensity upon antibody addition. This indicates that our experiment is capturing the RSVF-antibody interaction in these cases.

In contrast, analysis of all peptides with altered intensity upon antibody addition (these include peptides at the epitope as before, but also peptides near the epitope in the 3D structure and those that are more distant) shows a more complex picture. Out of the 57 such peptides we detect in all experiments, 63% (36 peptides: 6 FT and 30 ST) are indicative of increased protection and 37% (21 peptides: 11 ST and 10 FT) are indicative of decreased protection. Note that, in analyses of proteolytic susceptibility, FT peptides are more reliable than ST peptides since the intensity of the latter can also be artifactually reduced even in conditions of increased cleavage of a given protein region, due to secondary cleavage events within the sequence of ST peptides (i.e., additional cleavage of a previously generated ST peptide). By contrast, increased cleavage within the sequence of a FT peptide will always result in a decrease of intensity of the FT peptide, and vice versa.

In any event, this analysis shows that some of the protease susceptibility changes we detect are unlikely to represent a footprinting effect, but may be due to other structural changes that occur as a secondary consequence of binding. We have now added a discussion of this analysis to the revised manuscript (lines 219-230).

PostRSVF interaction with motavizumab in complex extract and other affected proteins:
There are apparently other proteins which also came up in the first experiment. The dose-dependence shown in Figure 2 is a clear proof of the target PPI, but is there a way to discriminate between off-targets and the real target already in the initial experiment? I.e. were the effects less pronounced for the other proteins? The raw peptide numbers already give a good indication (8 of 14 peptides were found for the known target), but any additional metric would be valuable.

The reason we introduced the dose-response analysis is precisely because, based both on first principles and on our previous observations with protein-small molecule interactions, it is a good way to distinguish between peptide changes that reflect a binding event versus those that do not. We note as well that, in general (i.e., outside of the case of a therapeutic antibody like motavizumab), defining an off-target PPI is not so easy.

We have however followed the referee's suggestion and assessed the strength of the effects for RSVF versus unexpected targets in the case of motavizumab. We computed a score for each peptide that shows significant changes upon addition of antibody, by dividing the absolute $\log_2(\text{fold change})$ of peptide intensity by the adjusted p-value. We reasoned that this score should yield a measure of the effect strength. Indeed we observed that RSVF peptides that change upon motavizumab addition have a higher score (median of 3651) relative to all peptides (median of 141), reflecting both their higher abundance change and statistical significance. These scores are shown in Dataset EV2 and the analysis is mentioned in the revised manuscript (lines 204-206). We note however that this approach will penalize low-abundance proteins.

AC8 - CaM interaction: The data are convincing and well presented, however the "other proteins" could be analyzed in more details. Some of these proteins contain CaM-binding motifs, but what about the rest? Are they functionally related and are we looking at secondary effects? A more detailed analysis using e.g. string-db or other bioinformatics analyses would be helpful. With the high number of detected peptides in a lysate, it's not surprising to see some background. However it would be great to have a visual readout or quality check, indicating which proteins are the most likely direct interaction partners of the tested protein.

As suggested, we have analyzed the candidate interactors of calmodulin using STRING but we unfortunately found no enrichment for known calmodulin interactors. This could be because the database is incomplete and/or because of secondary effects as the referee suggests. This is now reported in the revised manuscript (lines 287-288).

All proteins in the candidate interactor set have CaM-binding motifs. For 56 of them, as we had already reported, these map to the changing peptide. Proteins in the candidate calmodulin interactor set (n=162 proteins) have a median of 88 CaM-binding domains in their sequence, in comparison to 77 CaM-binding domains in all detected proteins minus the candidate interactor set (n=4972). The number of CaM binding domains per protein has now been added to Dataset EV3.

aSyn monomers vs. fibrils: The data for the known interaction partners are clear, but the "grading" of the other identified PPIs could be explained more clearly. Is there a way to create a list sorted by probability or strength of interaction, i.e., in such a way that most likely direct interactors are on top and proteins with less pronounced effects at the bottom?

As we had already mentioned in the discussion and shown for the aSyn fibril interactors, we can in principle use the LiP-MS dose-response curves to derive half-maximal response concentrations for any peptide. These should be a measure of relative affinity of an

interaction reported by that peptide and could be used to rank candidate interactors. We have previously tested this approach for protein-small molecule interactions (Piazza et al, 2020) but it remains to be tested against ground-truth datasets for protein-protein interactions. We have now described this in more detail in the discussion (lines 510-515) .

We also considered using the scores described above, i.e., absolute $\log_2(\text{fold change})$ of peptide intensity divided by the adjusted p-value, to grade other candidate interactors, but did not do so since the score is likely to penalize low abundance proteins.

Also, could the authors comment on how "complete" the aSyn interactome was mapped? I.e., how many interactions are listed in STRING (with which stringency settings used?) and what's the percentage of these PPIs detected by LiP-MS here?

We detected a total of 5565 proteins in the LiP-MS experiments probing for aSyn interactions. Out of these, 235/237 proteins are considered aSyn fibril/monomer interactors based on the STRING database (we used a permissive threshold, interaction score >150). Out of these, 25 proteins (10.63%) significantly changed upon addition of aSyn fibrils and 10 changed upon addition of aSyn monomer (4.2%). This information is now more clearly described in the revised manuscript (lines 324-325, lines 342-346) .

A final minor comment for the intro: The authors may consider mentioning "Thermal Proteome Profiling" approaches, which can also be used to identify drug-target interactions (with the limitation that TPP does not provide interaction sites).

We have now mentioned in the intro that TPP has been used to profile protein complex dynamics (lines 90-92).

Reviewer #4:

In this article, Holfeld et al use LiP to probe PPIs. More specifically, they spike recombinant proteins in cell extracts and employ LiP to detect variations in proteolytic signatures that are caused by alteration of binding sites and/or conformation changes. This manuscript presents an interesting series of results that demonstrate how LiP can also be applied to probe PPIs at a global scale. It also provides novel interesting insights related to alpha-synuclein interactomes. I recommend this article for publication providing the authors can address the minors points raised below.

It seems that different cutoff were used: $\log_2 \text{FC} > 1$ in page 4, then 0.75 on page 7. Unless there is a specific rationale, the same cut off should be used across the study.

We used $\log_2 \text{FC} > 1$ for single-dose experiments with purified samples and $\log_2 \text{FC} > 0.75$ for all experiments in lysates, which in most cases were multi-dose experiments. In the latter case, meaningful changes are not only detected based on the fold change, but also on how well the dose-response data fit a sigmoidal profile; the analysis can thus tolerate a slightly less stringent fold change threshold, and we use this to capture changes that are subtle and

yet likely to be biologically meaningful. We have now made this clear in the methods section (lines 958-962).

In principle a reduction of tryptic peptide signature should be caused when the majority of the protein within a sample is affected. In contrast, an increase of a semi-tryptic peptide could be detected even when sub-stoichiometric fraction is affected. Using that scheme, can the authors differentiate between stable vs more transient PPIs.

We do not understand the rationale. We think that when a mixture of bound and unbound states exists, both fully- and half-tryptic peptides will report an average.

While we do not see a way to distinguish between stable and transient interactions based on the strategy proposed, we now provide the requested table (see next point) showing all changing peptides, annotated by whether they are fully or half-tryptic, upon spike in of RSVF and anti-RSVF antibodies into lysates.

The authors should include a table to summarize which peptides were significantly more or less detected in presence of the different antibodies and their positions. Reading the text, it is not always obvious how the peptide signature was altered more precisely. In principal, tryptic peptides and semi tryptic peptides that cover the same binding motif/region should show opposite trends.

We have now annotated the EV Datasets with information on whether a peptide is fully tryptic (FT) or semi-tryptic (ST) (i.e., having either two or one tryptic ends), and with the directionality of the change in peptide intensity.

For the RSVF-antibody experiments, the known epitopes provide a ground truth to assess peptide intensity changes in what is expected to be a classical footprinting scenario, with the added twist that ST and FT peptides are expected to behave differently. As the referee says, the general expectation is that ST peptides at an epitope will decrease in intensity when the antibody is added, since binding reduces accessibility to proteinase K (PK). In contrast, FT peptides at epitopes, and that encompass the PK cleavage site, would be expected to increase in intensity when the binder is added. We have now added further discussion of these peptide types to the manuscript (lines 152-158).

We analyzed the behavior of RSVF peptides that map to known epitope sequences, upon antibody spike-in. In all cases (D25, 5C4, Motavizumab, Palivizumab, 101F), peptides at epitopes indicate increased proteolytic protection upon addition of the antibody. Specifically, there are 3 FT peptides that go up and 8 ST peptides that go down in intensity upon antibody addition. This indicates that our experiment is capturing the RSVF-antibody interaction in these cases.

In contrast, analysis of all peptides with altered intensity upon antibody addition (these include peptides at the epitope as before, but also peptides near the epitope in the 3D structure and those that are more distant) shows a more complex picture. Out of the 57 such peptides we detect in all experiments, 63% (36 peptides: 6 FT and 30 ST) are indicative of increased protection and 37% (21 peptides: 11 ST and 10 FT) are indicative of decreased

protection. Note that, in analyses of proteolytic susceptibility, FT peptides are more reliable than ST peptides since the intensity of the latter can also be artifactually reduced even in conditions of increased cleavage of a given protein region, due to secondary cleavage events within the sequence of ST peptides (i.e., additional cleavage of a previously generated ST peptide). By contrast, increased cleavage within the sequence of a FT peptide will always result in a decrease of intensity of the FT peptide, and vice versa.

In any event, this analysis shows that some of the protease susceptibility changes we detect are unlikely to represent a footprinting effect, but may be due to other structural changes that occur as a secondary consequence of binding. We have now added a discussion of this analysis to the revised manuscript (lines 221-232).

Instead of showing the semi-tryptic peptides on the structure, the position of PK cleavage site (e.g. P1 site) would more precisely show which residues are more (or less) accessible upon antibody binding.

Since our readout is the peptides, we think depicting them is the more conservative and accurate representation of the data. However, we have added the position of the PK cleavage site to all EV Datasets.

How do the authors know which conformation of RSVF is present in their samples (pre or post)? Unless I missed that information, more clarifications should be provided.

These experiments were done with spiked-in RSVF, in specific conformations. Note that these conformations are known to be stable (Sesterhenn et al, 2020; McLellan et al, 2013; McLellan et al, 2011).

Figure 2A and S2. Specify n and indicate whether the peptides are semi-tryptic or not

We have added this information to the figures and legends.

Revised Figure 2: LiP-MS detects protein-protein interactions in complex proteomes. (A) Dose-response curves of eight LiP peptides with indicated amino acid positions originating from postRSVF show relative peptide intensities proportional to the amount of motavizumab spiked into HEK293T cellular extracts ($n = 3$ replicates each). Pearson's coefficient (r) to a sigmoidal trend of the peptide-intensity response profile is indicated. These peptides correspond to the altered peptides (green) in panels B and C. (B) The structure of postRSVF (PDB: 3RRR) (McLellan et al, 2011) with peptides altered in the dose-response analysis ($r > 0.85$; $|\log_2 FC| > 0.75$; moderated t -test, q -value < 0.01) indicated in green and antigenic site II in red. (C) Zoom of the altered peptides on the structure of postRSVF (PDB: 3RRR) (McLellan et al, 2011) with colors as in panel (B).

About 1/3 of LIP peptide detected upon the addition of CaM are located CaM binding motifs. What about the other 2/3. Do these proteins also have CaM motifs but mapped elsewhere? The authors should add some clarifications.

Yes, all proteins in the candidate interactor set have CaM-binding motifs. For 56 of them, as we had already reported, these map to the changing peptide. Proteins in the candidate calmodulin interactor set (n=162) have a median of 88 CaM-binding domains in their sequence, in comparison to 77 CaM-binding domains in all detected proteins (n=4972). This information has now been added to Dataset EV3.

Also, in response to referee 3, we analyzed the candidate interactors of calmodulin using STRING but we unfortunately found no enrichment for known calmodulin interactors. This could be because the database is incomplete and/or because of secondary effects. We now mention this in the revised manuscript (lines 287-288).

Page 12. Line 4. Are the p-values and FC related to this study or Xia et al 2008. If the former, the threshold should be adjusted to be the same across the study

The p-values relate to the study of Xia et al, 2008.

Page 12 Figure S1D should probably be S4D

We have corrected this.

Page 12 last sentence: add citation(s).

We have added citations.

Page 14. The section related to GWAS analysis is underdeveloped and should be removed if not appropriately substantiated. What is the meaning of the analysis? Do the authors think that SNPs in some of these GWAS could weaken some of the PPIs?

We realize that this analysis could have been better motivated.

Our objective was to study if the proteins that are structurally altered in the presence of aSyn are functionally linked with PD. It is well documented that proteins that interact tend to be functionally related and it is also well known that GWAS will miss many genes that are important for a given disease simply due to lack of association power. Since our LiP-MS experiment did not find known PD disease genes among the proteins that are structurally altered in the presence of aSyn, we asked if these proteins are part of pathways and/or complexes that contain PD disease genes. This is what we indeed observed, in the analysis of genes that are linked to PD via GWAS and other genetic studies.

In short, we do not imply that the GWAS SNPs act to modulate the interaction we found; rather, the analysis shows that the structurally altered proteins are functionally related to cell biology that is relevant to PD. We have now revised the relevant section to clarify the motivation and interpretation of this analysis (lines 407-412, lines 424-426).

REFERENCES

Gillingham AK, Bertram J, Begum F & Munro S (2019) In vivo identification of GTPase interactors by mitochondrial relocalization and proximity biotinylation. *Elife* 8

McLellan JS, Yang Y, Graham BS & Kwong PD (2011) Structure of Respiratory Syncytial Virus Fusion Glycoprotein in the Postfusion Conformation Reveals Preservation of Neutralizing Epitopes. *J Virol* 85: 7788–7796

McLellan JS, Chen M, Leung S, Graepel KW, Du X, Yang Y, Zhou T, Baxa U, Yasuda E, Beaumont T, *et al* (2013) Structure of RSV fusion glycoprotein trimer bound to a prefusion-specific neutralizing antibody. *Science (1979)* 340: 1113–1117

Piazza I, Beaton N, Bruderer R, Knobloch T, Barbisan C, Chandat L, Sudau A, Siepe I, Rinner O, de Souza N, *et al* (2020) A machine learning-based chemoproteomic approach to identify drug targets and binding sites in complex proteomes. *Nat Commun* 11

Sesterhenn F, Yang C, Bonet J, Cramer JT, Wen X, Wang Y, Chiang CI, Abriata LA, Kucharska I, Castoro G, *et al* (2020) De novo protein design enables the precise induction of RSV-neutralizing antibodies. *Science (1979)* 368.

Tofaris GK. *et al* (2001). α -Synuclein metabolism and aggregation is linked to ubiquitin-independent degradation by the proteasome. *FEBS Letters* 509, 1873-3468.

Zambon F. *et al* (2019). Cellular α -synuclein pathology is associated with bioenergetic dysfunction in Parkinson's iPSC-derived dopamine neurons *Human Molecular Genetics*, 28, 12, 2001–2013.

12th Mar 2024

Manuscript Number: MSB-2023-11552R

Title: Systematic identification of structure-specific protein-protein interactions

Dear Natalie and Paola,

Thank you for sending us your revised manuscript. We have now heard back from the two reviewers who agreed to evaluate your revised study. As you will see below, the reviewers think that the study has improved as a result of the performed revisions. However, reviewer #1 raises some concerns regarding analyses identifying proteins interacting with the GDP and GTP-bound states of Rab2A. Reviewer #1 has included a figure with their comments, which I have attached below. We think that a more transparent presentation and balanced discussion/interpretation of the related findings can address the remaining concerns of the reviewer without the need to perform additional experiments. Taken together, we would ask you to address the remaining concerns in a last round of revision. We do not think changing the title of the manuscript as recommended by reviewer #1 is required. We would also ask you to address some remaining editorial issues listed below.

- The title of the "Conflict of Interest" statement needs renaming to "Disclosure And Competing Interests Statement".
- Please remove the 'Authors Contributions' from the manuscript. The 'Author Contributions' section is replaced by the CRediT contributor roles taxonomy to specify the contributions of each author in the journal submission system. Please use the free text box in the 'author information' section of the online submission system to provide more detailed descriptions if needed (e.g., 'X provided intracellular Ca⁺⁺ measurements in fig Y').
- Please include callouts for Fig. 3E, 4F.
- The callouts to Appendix Table S1 need to be updated to Appendix Table S1, (instead of Supplementary Table 1.)
- The section order should be corrected as follows: title page with complete author information, abstract, keywords, introduction, results, discussion, materials & methods, data availability, acknowledgements, disclosure and competing interests statement, references, main figure legends, tables, expanded figure legends.

Please resubmit your revised manuscript online, with a covering letter listing amendments and responses to each point raised by the referees. Please resubmit the paper ****within one month**** and ideally as soon as possible. If we do not receive the revised manuscript within this time period, the file might be closed and any subsequent resubmission would be treated as a new manuscript. Please use the Manuscript Number (above) in all correspondence.

Click on the link below to submit your revised paper.

Kind regards,

Maria

Maria Polychronidou, PhD
Senior Editor
Molecular Systems Biology

If you do choose to resubmit, please click on the link below to submit the revision online before 11th Apr 2024.

IMPORTANT: Please note that corresponding authors are required to supply an ORCID ID for their name upon submission of a revised manuscript (EMBO Press signed a joint statement to encourage ORCID adoption).
(<https://www.embopress.org/page/journal/17444292/authorguide#editorialprocess>)

Currently, our records indicate that the ORCID for your account is 0000-0003-4286-8951.

Link Not Available

*** PLEASE NOTE *** As part of the EMBO Press transparent editorial process initiative (see our Editorial at <https://dx.doi.org/10.1038/msb.2010.72> , Molecular Systems Biology will publish online a Review Process File to accompany accepted manuscripts. When preparing your letter of response, please be aware that in the event of acceptance, your cover letter/point-by-point document will be included as part of this File, which will be available to the scientific community. More information about this initiative is available in our Instructions to Authors. If you have any questions about this initiative, please contact the editorial office (msb@embo.org).

Reviewer #1:

In the revised manuscript titled "Systematic identification of structure-specific protein-protein interactions," the authors have diligently addressed many of the comments raised by the reviewers, resulting in significant enhancements to the manuscript's clarity. However, while commendable progress has been made, the current work remains insufficient to unequivocally verify the absolute assertion of "Systematic identification of structure-specific protein-protein interactions."

This concern was a significant focal point in the previous version of the manuscript. The authors have undertaken an additional experiment in the revised manuscript to bolster their claim. Specifically, they employed LiP-MS to identify proteins interacting with the GDP and GTP-bound states of Rab 2A and Rab 5A, respectively. The resulting data revealed different groups of proteins with altered protease susceptibility when probing with either the GDP or GTP-bound protein forms. While this experiment provides valuable insights, the manuscript still lacks evidence to conclusively demonstrate that the identified candidates exclusively interact with one bait protein form while remaining uninvolved with its counterpart. While enrichment might suffice, the biological significance of the results remains hidden for now. The display of the results is not satisfactory. Typically, volcano plots are presented and enriched proteins are labelled. Surely the authors know this, and it looks as if they did not quite see a meaningful story in the data. The statistical comparison to the results of another screen (proximity labelling) is insufficient when wanting to assess if the approach is throwing up biologically valuable candidates. As known structure-specific interactors failed to show up due to sensitivity limits, I see little alternative but listing the top hits and discussing or better experimentally proving their biological meaning in the context of the screen. This would allow other researchers to assess if the proposed approach is worth pursuing by them and what to expect in terms of hit rate and candidate types.

Considering this absence, it may be prudent for the authors to moderate their statement to "Towards systematic identification of structure-specific protein-protein interactions" or "Preliminary work on systematic identification of structure-specific protein-protein interactions" to better reflect the current status of their technical advancements.

Additionally, here are some suggestions regarding the presentation of the Rab GTPase data:

1. A figure illustrating the structural differences between the GDP and GTP-bound states of Rab 2A/Rab 5A would enhance comprehension.

2. In Figure 5, employing a proportional Venn diagram would more accurately depict the proportions of overlapped and unique protein candidates.

Furthermore, the manuscript states, "Since our goal was to systematically identify PPIs within a native cellular environment, we further analyzed the 186 interactions between postRSVF and motavizumab in a complex extract of HEK293T cells." However, it's noteworthy that "spiked-in RSVF was the most abundant protein in these lysates."

Insert figure here

Given this disproportionate ratio of spiked-in RSVF to HEK293 lysate, the latter fails to offer an analytical background resembling a true native cellular environment. Analogously, spotting a tree amidst a forest differs significantly from spotting it among the grass on a football field.

These considerations should be addressed to refine the manuscript further before it is published in MSB.

Reviewer #3:

The authors addressed the points I raised in the first round of revisions adequately by adding data from new experiments and rewriting the manuscript. A minor remaining point is that no reverse 2nd generation experiments were done - for example, a reverse IPs with interaction partners to fish for aSyn was not included (an IP was conducted using an anti-aSyn antibody, but not using an anti-interaction-partner antibody and then looking for aSyn = a true "reverse experiment"). This would be a nice confirmation of the interactors postulated in the manuscript.

Still, a very nice piece of work that is suitable for MolSysBiol.

We thank the referees for their time and effort.

Reviewer

#1:

In the revised manuscript titled "Systematic identification of structure-specific protein-protein interactions," the authors have diligently addressed many of the comments raised by the reviewers, resulting in significant enhancements to the manuscript's clarity. However, while commendable progress has been made, the current work remains insufficient to unequivocally verify the absolute assertion of "Systematic identification of structure-specific protein-protein interactions."

We intend this approach to be a screen and have made very clear throughout the manuscript that what we identify here are candidate interactors that have to be validated with orthogonal techniques. We have now adjusted the abstract, introduction and discussion to further emphasize this.

This concern was a significant focal point in the previous version of the manuscript. The authors have undertaken an additional experiment in the revised manuscript to bolster their claim. Specifically, they employed LiP-MS to identify proteins interacting with the GDP and GTP-bound states of Rab 2A and Rab 5A, respectively. The resulting data revealed different groups of proteins with altered protease susceptibility when probing with either the GDP or GTP-bound protein forms. While this experiment provides valuable insights, the manuscript still lacks evidence to conclusively demonstrate that the identified candidates exclusively interact with one bait protein form while remaining uninvolved with its counterpart. While enrichment might suffice, the biological significance of the results remains hidden for now. The display of the results is not satisfactory. Typically, volcano plots are presented and enriched proteins are labelled. Surely the authors know this, and it looks as if they did not quite see a meaningful story in the data. The statistical comparison to the results of another screen (proximity labelling) is insufficient when wanting to assess if the approach is throwing up biologically valuable candidates. As known structure-specific interactors failed to show up due to sensitivity limits, I see little alternative but listing the top hits and discussing or better experimentally proving their biological meaning in the context of the screen. This would allow other researchers to assess if the proposed approach is worth pursuing by them and what to expect in terms of hit rate and candidate types.

We cannot show volcano plots since the screens for interactors of the different Rab proteins was done using a multiple dose setup. i.e., the bait protein of interest was spiked into the lysate in several different doses. As we have previously shown for protein-small molecule interactions, using such a setup and filtering for proteins with a good dose-response increases the specificity of the identified interactors.

We have however added Figure 5F to the revised manuscript, which shows the top 50 hits for each nucleotide-bound form of each Rab, in a manner that also gives a visually informative picture of the specificity and relative strengths of these top interactors.

Considering this absence, it may be prudent for the authors to moderate their statement to "Towards systematic identification of structure-specific protein-protein interactions" or "Preliminary work on systematic identification of structure-specific protein-protein interactions" to better reflect the current status of their technical advancements.

As suggested by the editor, we have retained the current title. If needed, we suggest changing it to "Systematic identification of candidate structure-specific protein-protein interactions:

Additionally, here are some suggestions regarding the presentation of the Rab GTPase data:
1. A figure illustrating the structural differences between the GDP and GTP-bound states of Rab 2A/Rab 5A would enhance comprehension.

We have added this in Figure 5A, as requested, for Rab5A. There are no experimental structures available for Rab2A.

2. In Figure 5, employing a proportional Venn diagram would more accurately depict the proportions of overlapped and unique protein candidates.

The Venn diagrams are now proportional, as requested.

Furthermore, the manuscript states, "Since our goal was to systematically identify PPIs within a native cellular environment, we further analyzed the 186 interactions between postRSVF and motavizumab in a complex extract of HEK293T cells." However, it's noteworthy that "spiked-in RSVF was the most abundant protein in these lysates."

Given this disproportionate ratio of spiked-in RSVF to HEK293 lysate, the latter fails to offer an analytical background resembling a true native cellular environment. Analogously, spotting a tree amidst a forest differs significantly from spotting it among the grass on a football field.

We have moved the comment about RSVF abundance up to further emphasize it. While it is correct that the RSVF system is a best-case scenario (not only in terms of the amount of RSVF present but also in terms of the high affinity and specificity of the antibodies used), we have also demonstrated that the method identifies known interactions of AC8 and of aSyn monomer and fibril. Further, the method is meant as an initial screen to identify candidate interactors that must be verified with orthogonal methods. This is now further emphasized in the abstract, introduction and discussion of the manuscript.

These considerations should be addressed to refine the manuscript further before it is published in MSB.

Reviewer #3:

The authors addressed the points I raised in the first round of revisions adequately by adding data from new experiments and rewriting the manuscript. A minor remaining point is that no reverse 2nd generation experiments were done - for example, a reverse IPs with interaction partners to fish for aSyn was not included (an IP was conducted using an anti-aSyn antibody, but not using an anti-interaction-partner antibody and then looking for aSyn = a true "reverse experiment"). This would be a nice confirmation of the interactors postulated in the manuscript. Still, a very nice piece of work that is suitable for MolSysBiol.

We thank the referee for their support.

15th Apr 2024

Manuscript number: MSB-2023-11552RR

Title: Systematic identification of structure-specific protein-protein interactions

Dear Paola,

Thank you again for sending us your revised manuscript. We are now satisfied with the modifications made and I am pleased to inform you that your paper has been accepted for publication.

Kind regards,

Maria

Maria Polychronidou, PhD
Senior Editor
Molecular Systems Biology
